# SHIKI: Self-Supervised Heuristic for Improving MLPs' Knowledge by Integrating GNNs

## Abstract

Graph Neural Networks (GNNs) are widely recognized as leading architectures for addressing classification problems involving graphical data. In this paper, we formally define the challenge of effectively constructing edges within a dataset and training a GNN over this graph and introduce SHIKI - a novel method to tackle this task. We provide a comprehensive theoretical analysis demonstrating how graph convolutions can improve expected performance by leveraging edges. Our study focuses on the node classification problem within a non-linearly separable Gaussian mixture model, combined with a stochastic block model, and we visually demonstrate its applicability to real-world datasets. Specifically, we show that a single graph convolution in the second layer can reduce the expected loss when applying a heuristic for edge creation. We validate our findings through extensive experiments on both synthetic and real-world datasets, including those related to the entity matching problem and textual review classification. For the synthetic data, we conduct experiments based on the dataset's difficulty and various hyperparameters in our method, drawing connections between the two. Additionally, we perform an ablation study by systematically removing components of our method and testing the resulting degraded approach, which highlights the necessity of our full method. We employ several GNN architectures in the experiments, including GCN, GraphSAGE, and GAT.

## 1 Introduction

Graph Neural Networks (GNNs) have emerged as a powerful tool for learning from graph-structured data, with applications spanning social networks Ding et al. (2019), molecular biology Gaudelet et al. (2021), recommender systems Wu et al. (2018), and more. Aside from tasks where graph-structure lends itself well to the domain, GNNs were also shown to be useful in tasks where the data is not inherently structured, *e.g.*, entity matching (Genossar et al., 2023b).

Graph convolutional models Kipf & Welling (2017) are among the most popular approaches for learning on relational data, leveraging the idea of aggregating the features of a node's neighbors rather than just its own. While numerous empirical studies on GCN variants Chen et al. (2020) have demonstrated that graph convolutions can outperform traditional classification methods like multi-layer perceptrons (MLPs), there has been little theoretical progress in explaining how graph convolutions enhance node classification in multi-layer networks, especially on non-graphical data.

Baranwal *et al.* recently showed, both theoretically and empirically, that even for applications without inherent graph structure, synthetically created edges can boost performance Baranwal et al. (2022). Specifically, they demonstrate an improvement in performance where the data poses a training challenge for a simple multi-layer perceptron (MLP). In their work, edges are created according to prior knowledge of the sample label, and no method for incorporating edge creation into the learning pipeline was proposed. We aim to bridge this gap, proposing a heuristic to create useful edges on top of an MLP that models non-structural data in a self-supervised manner, followed by training a GNN model over it using the generated edges. Our contribution is threefold:

- We propose and formulate a novel method for adding edges to a non-graphical data.

- We show the effectiveness of our method in terms of expected loss.

- We empirically verify the formal results of our method using basic and known GNN architecture such as GCN Kipf & Welling (2017), GraphSAGE Hamilton et al. (2018), GAT Veličković et al. (2018), showing improvement over MLP training.

An open source anonymous access to SHIKI implementation is available here.

The rest of the paper is organized as follows. In Section 2, we provide a description of the data model. In Section 3 we state our objective and provide a problem definition. In Section 4 we describe our proposed solution. We also provide our notion of improvement and show results on the expected improvement. In Section 5 we detail our experiments on synthetic and real-world data, including an ablation analysis, to demonstrate the proposed method performance quality. We present relevant related work in Section 6 and conclude with some directions for future work in Section 7.

## 2 PRELIMINARIES

In this work, we use the XOR-GMM model Baranwal et al. (2022) to generate synthetic data. The model serves as a basis to our formal and empirical analysis.

Let $n$ and $d$ be positive integers, where $n$ represents the number of data points (sample size) and $d$ denotes the dimension of features. Let $\epsilon_1, \ldots, \epsilon_n \sim \text{Ber}\left(\frac{1}{2}\right)$ and $\eta_1, \ldots, \eta_n \sim \text{Ber}\left(\frac{1}{2}\right)$ be Bernoulli random variables. Also, let $C_b = \{i \in [n] \mid \epsilon_i = b\}$ for $b \in \{0, 1\}$ be two classes. Let $\mu$ and $\nu$ be fixed vectors in $\mathbb{R}^d$, such that $\|\mu\|_2 = \|\nu\|_2$ and $\langle \mu, \nu \rangle = 0$. Let $X \in \mathbb{R}^{n \times d}$ be the data matrix where each row-vector $X_i \in \mathbb{R}^d$ is an independent Gaussian random vector with distribution

$$X_i \sim \mathcal{N}\left((2\eta_i - 1)((1 - \epsilon_i)\mu + \epsilon_i\nu), \sigma^2 I_d\right) \tag{1}$$

We use the notation $X \sim XOR\text{-}GMM(n, d, \mu, \nu, \sigma^2)$ to denote data sampled from this model.

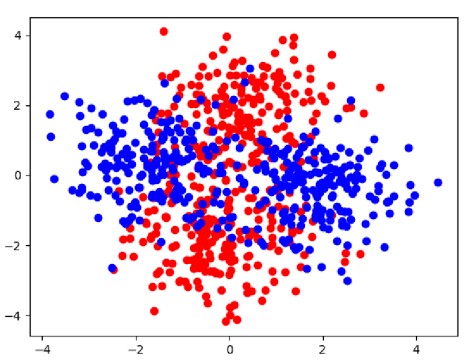
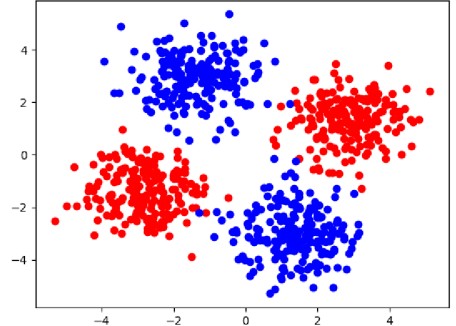

(a) Example of a difficult dataset can be created by the model, with distance between centers of $\|\mu - \nu\| = 2.5$, and standard deviation $\sigma = 0.7$.

(b) Example of an easy dataset can be created by the model, with distance between centers of $\|\mu - \nu\| = 4.5$, and standard deviation $\sigma = 0.7$.

Figure 1: Different data characteristic possible with the synthetic model

**Example 1.** *Through the use of specific parameters of the model, including the distance between centers $\|\mu - \nu\|$, the variance $\sigma^2$, and number of points $n$, we can control the difficulty of classification models to achieve their goal, when trained with the data. We illustrate this difference using Figure 1. Figure 1a illustrates a more challenging classification setting than Figure 1b due to the shorter distance between the cluster centers. The mix between the blue and the red instances makes it harder to train a classification model.*

We use the XOR-GMM model to support our formal analysis. Despite its synthetic nature, we observe that multiple real-world datasets exhibit behavior that can be captured by this model. For illustration, we present next two well-known tasks, namely review classification on Amazon dataset[1] and entity matching on the Walmart-Amazon dataset.[2]

---

[1] https://www.kaggle.com/datasets/drshoaib/amazon-videogames-reviews
[2] https://github.com/anhaidgroup/deepmatcher/blob/master/Datasets.md

The Amazon videogames reviews dataset contains users' reviews for video games on Amazon. For each review, details are given about the reviewer, the product, the review text, and the overall rating ranging in $[1, 5]$. The learning task involves predicting the overall rating given the other details.

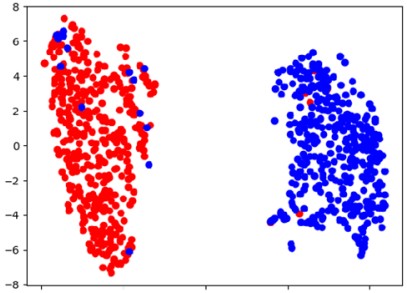
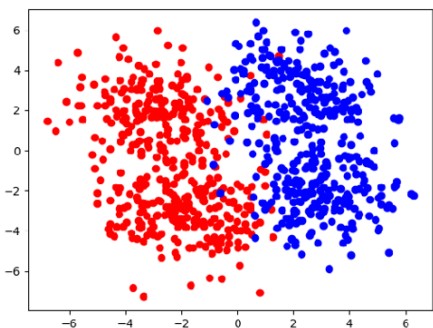

(a) Amazon dataset embeddings after fine-tuning bert and dimension reduction using TSNE.

(b) Synthetically generated data according to the model's version modelling the Amazon dataset.

Figure 2: Visual comparison between real-world data and synthetically generated data for the Amazon review dataset

Figure 2 illustrates the vector space of data points from the Amazon videogames review dataset (left) and a simulation of the data using a variation of the XOR-GMM model, keeping $\epsilon_i, \eta_i, C_b$ unchanged. We note that, unlike the theoretical model where centers of the same class are on opposites sides, in this dataset, the centers of each class are on the same side, creating the elongated shapes. Therefore, the GMM distributions becomes $X_i \sim \mathcal{N}\left((2\epsilon_i - 1)((1 - \eta_i)\mu + \eta_i\nu), \sigma^2 I_d\right)$. We swap the roles of $\eta_i$ and $\epsilon_i$, meaning, points from class 1 will have the centers $\mu, \nu$, and points from class 0 will have the centers $-\mu, -\nu$. The result is given in Figure 2b.

The Walmart-Amazon dataset is taken from the Magellan data repository (Konda et al., 2016). It is a well-known dataset for evaluation of entity matching solutions. This dataset contains product data from Walmart and Amazon. The original dataset contained two tables, and a golden standard match.

To better understand the matching task, we briefly present the entity matching problem. Let $D = \{r_1, r_2, ..., r_n\}$ be a set of data records (*dataset*) and $E = \{e_1, e_2, ..., e_m\}$ a set of real-world entities ($m \leq n$). Each record is associated with an entity in $E$ using an *entity mapping* (mapping for short) $\theta : D \to E$. Whenever $\theta$ is unknown, for example, due to the absence of unique keys to identify entities, entity resolution solutions aim to pair records in $D$ such that if $\{r_i, r_j\} \subseteq D$ are paired together then $\theta(r_i) = \theta(r_j)$. $D$ is usually characterized by a set of attributes $A = \{a_1, a_2, ..., a_k\}$, such that a record $r_i = \langle r_i.a_1, r_i.a_2, ..., r_i.a_k \rangle$ is assigned with values to all attributes (some of which may be null values).

For the Walmart-Amazon entity matching dataset, Figure 3a provides a two dimensional illustration of representative vectors (with dimension of 768) of a fully trained models. We observe that positive pairs tend to gather together, surrounded by a background of negative pairs. Unlike the theoretical model where each class has two centers, this dataset has class imbalance, and the classes are represented by a single positive center and three negative centers instead of balanced two centers for each class. To capture imbalance, one center has less points than each center from the other class. Thus, we need to model further imbalance for the center. We model this by giving a lower probability for a point to be in this center.

To achieve this setting with the XOR-GMM model, we define $w_0$ to be the probability of a node being in class 0, and $w_1$ in class 1. Obviously $0 < w_0 = 1 - w_1 < 1$, in order to achieve the phenomena of a small center, we use $0 < w_1 < \frac{1}{2}$. The definitions for $\epsilon_i, X_i$ stay unchanged, while we set $\eta_i = Ber\left(\frac{1}{2} + \epsilon_i\left(w_1 - \frac{1}{2}\right)\right)$, and set $C_b$ to be $C_0 = \{i \in [n] \mid \epsilon_i = 0 \vee \eta_i = 0\}$, $C_1 = \{i \in [n] \mid \epsilon_i = 1 \wedge \eta_i = 1\}$. Data points from class 1 have a single center $\nu$, and points from class 0 have three centers $\mu, -\mu, -\nu$. The result is given in Figure 3b.

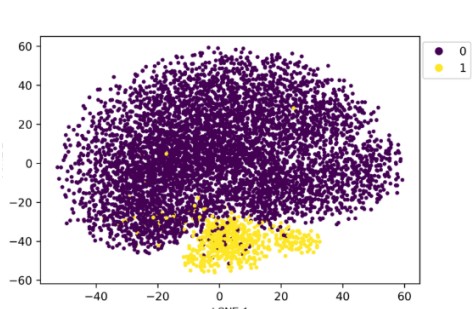 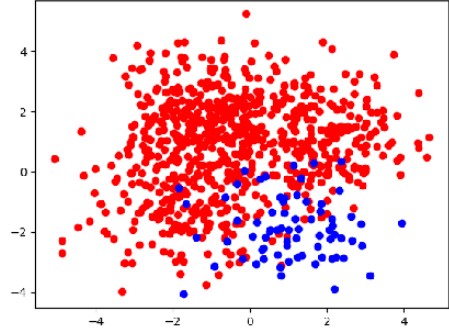

(a) Visualization of pairs distribution by t-SNE, partitioned into match and non-match pairs. Taken from (Genossar et al., 2023a).

(b) Synthetically generated data according to the model's version modelling the Walmart-Amazon Entity Matching dataset

Figure 3: Visual comparison between real world data and Synthetically generated data for the Walmart-Amazon entity matching dataset

These models deviate from the original one by either class imbalance, or shifted centers. Due to the high similarities of these models, in Appendix B we demonstrate that given the original model, each such deviation retains the nice theoretical properties of the original XOR-GMM model, and in Appendices C and D we formally prove these nice theoretical properties.

We conclude the section with a description of the process of creating the graph over a XOR-GMM generated dataset, following (Baranwal et al., 2022). Although we do not use this process, its description assists in defining our proposed method. The graph is represented as an adjacency matrix, $A = (a_{ij}), i, j \in [n]$, which corresponds to an undirected graph including self-loops, and is sampled as follows. $a_{ij} \sim Ber(p)$ if $\epsilon_i = \epsilon_j$ and $a_{ij} \sim Ber(q)$ if $\epsilon_i \neq \epsilon_j$. Therefore, for any two nodes, if they share a class, we create an edge with probability $p$, otherwise we create an edge with probability $q$. We call it XOR-CSBM, and denote $(A, X) \sim XOR\text{-}CSBM(n, d, \mu, \nu, \sigma^2, p, q)$.

## 3 PROBLEM STATEMENT

In Section 2 we have demonstrated, using two real-world datasets, an interesting spatial effect. Using MLP, we can construct an embedded vector space in which data items from the same class tend to cluster together. Such a phenomenon provides us with a good starting point when constructing a graph structure that serves in training a GNN to improve the outcome of a classification problem. Our goal in this paper, is therefore, to enhance MLP usage of node features by connecting similar nodes of the same class and use GNN's message-propagation to improve the generated embedded space. We focus our attention on a careful selection of edges to connect nodes of the same class. In this work we offer a comparative analysis of artificially created graph convolutions with those of a traditional MLP that does not incorporate graphical information. In particular, we are interested in answers to the following two questions. First, is it possible to create edges from a non-graphical data in a way that takes advantage of the performance improvement GNN provides in a graphical data? Then, we are interested in identifying provable improvements.

Let $X \sim XOR\text{-}GMM(n, d, \mu, \nu, \sigma^2)$ be as defined above. Our goal is to design $f_{GNN} = G(X, E)$, a function that takes as a parameter the data $X$, and outputs a graph from $X$ in a way that supports our overall goal of improved training. The nodes of $G$ are the data points, $V = X$ and $E \subseteq V \times V$. We define improvement in terms of expectation over the normal distribution of Eq. 1. We treat $n, d, \mu, \nu$ as constants. Therefore,

$$\mathbb{E}_X(f(X)) \equiv \mathbb{E}_X(f(X)|n, d, \mu, \nu) \tag{2}$$

The MLP and the GNN share most of the characteristics, as can be seen in Table 1. They differ only in the node computation $f^{(l)}(X)$. $k_l$ denotes the number of graph convolutions placed in layer $l$. The learnable parameters are $\theta(W_{(l)}, b_{(l)})_{l \in [L]}$. For the loss we use a standard cross-entropy loss

Table 1: MLP and GCN Characteristics

| Characteristic | MLP | GCN | Comments |
|---|---|---|---|
| $H^{(0)} =$ | $X$ | $X$ | |
| $f^{(l)}(X) =$ | $H^{(l-1)}W^{(l)} + b^{(l)}$ | $(D^{-1}A)^{k_l}H^{(l-1)}W^{(l)} + b^{(l)}$ | for $l \in [L]$ |
| $H^{(l)} =$ | $ReLU(f^{(l)}(X))$ | $ReLU(f^{(l)}(X))$ | for $l \in [L]$ |
| $\hat{y} =$ | $\varphi(f^{(L)}(X))$ | $\varphi(f^{(L)}(X))$ | |

defines as $L(X/G) = -\frac{1}{n}\sum_{i\in[n]} y_i \log(\hat{y}_i) + (1-y_i)\log(1-\hat{y}_i)$. Problem 1 summarizes our goal, as follows.

**Problem 1.** *Let $X \sim XOR\text{-}GMM(n, d, \mu, \nu, \sigma^2)$. Design $f_{GNN}(X)$ s.t.*

$$f_{GNN}(X) = G(X, E) \text{ s.t. } \mathbb{E}_X(L_{GNN}(G)) < \mathbb{E}_X(L_{MLP}(X)) \tag{3}$$

Problem 1 is defined in a way that closely follows the results of Baranwal et al. (2022), in that it seeks a graph that offers a provable improvement over the MLP performance, by expectation. We observe that this problem can be extended to a more general optimization problem of finding an optimal graph, as follows.

**Problem 2.** *Let $X \sim XOR\text{-}GMM(n, d, \mu, \nu, \sigma^2)$. Find $G(X, E)$ s.t.*

$$G = argmin_{G' \in \mathcal{G}}\mathbb{E}_X(L_{GNN}(G)) \tag{4}$$

Following Baranwal et al. (2022), we focus on solving Problem 1, leaving Problem 2 for future work.

## 4 SHIKI: A HURISTIC APPROACH

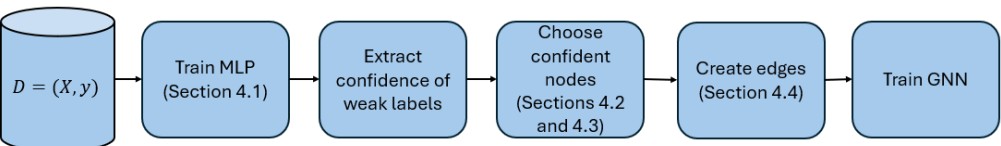

Figure 4: An Illustration of the SHIKI pipeline

Solving Problem 1, we present SHIKI: a Self-supervised Heuristic approach for Improving MLPs' Knowledge by Integrating GNNs. We use a heuristic similar to the graph creation described in Section 2. We observe that the GNN loss in Baranwal et al. (2022) depends upon $\left|\frac{p-q}{p+q}\right|$ (see Section 2), and for a GNN to be effective, we want $p$ and $q$ to be as different as possible. Had we known in advance the ground truth labels, we could directly control $p$ and $q$. However, we do not have a direct knowledge on the ground truth during test time. Therefore, we need to resort to approximating them.

Figure 4 illustrates a pipeline, in which a training dataset $D$ is effectively train by combining some MLP and through an effective selection of edges for a graph over the data items, trains a GNN. The pipeline contains five processing steps, to be detailed next. We conclude this section with some results on the improvement that can be gained by using SHIKI.

### 4.1 MLP TRAINING

The first step in the pipeline involves training and MLP on the data. The training yields three outcomes that are useful for us. First, it yields a label for each trained dataset. Second, it provides a confidence in the classification task. Finally, it generates a latent vector space.

The latent vector space offers a notion of a distance. In general, such a distance measure does not have to rely entirely on the outcome of the MLP. For tabular data, we can use the columns as

dimensions in a vector space. For textual data, we can create embeddings using an LLM, capturing the latent vector space of the last layer. Finally, for vector data, we can use the vectors themselves, or alternatively use an MLP's hidden-layer embeddings.

### 4.2 Confidence of Weak Labels

In this step we get the confidence score from the trained MLP, thus using *weak labels*. Weak labels are noisy and uncertain labels that may differ from the true labels. Using weak labels runs the risk of predicting the wrong labels to nodes in the graph. Wrong labels hurt the training in general and in particular will harm the process of edge creation. In particular, weak labels, when used in edge creation are likely to increase $q$, the probability of connecting mismatched nodes, which leads to deteriorated performance.

For each instance $x$, the MLP first calculates a score $|MLP(x)| \in (-\infty, \infty)$, on which we apply the sigmoid function to generate a distribution in $[0, 1]$. Nodes with high absolute values are assumed to be nodes the model are confident in their prediction, and we call them *confident nodes*. We assume (and later prove) that more confident nodes tend to produce more accurate predictions.

### 4.3 Confident Nodes Selection

In this step we determine the set of confident nodes to be labeled and participate in edge creation with respect to these weak labels, using the method in (Baranwal et al., 2022). We consider two possible ways to select confident nodes, as follows.

**Top percentile:** We order the nodes according to the scores the MLP assigns with them, and define any node $x$ that is within the top $percent$ of the scores, where $percent$ is some hyperparameter to be a confident node.

**Confidence threshold:** Given a threshold $\tau > 0$, a node $x$ is defined to be a confident node if $|MLP(x)| > \tau$.

We note that highly parameterized models, including transformer-based pre-trained language models, often generate uncalibrated confidence scores (Guo et al., 2017; Jiang et al., 2021; Genossar et al., 2023a). These scores tend to be mostly dichotomous, clustering near 0 or 1, making them unreliable. Thus, one needs to be careful directly using a LLM (instead of a simple MLP) as part of our model.

### 4.4 Edge Creation

Edge creation is done in the same manner as described in Section 2, taking into account only confident nodes. This method, which we refer to as *main strategy*, is based solely on node labels.

We observe that in real-world graphs, edges tend to connect nodes that are in close proximity. Thus, we propose a strategy that takes into account also node distances. We propose two strategies, as follows.

**Proximity Strategy:** Following the observation above, closer nodes will be more likely to be connected.

**Diversity Strategy:** Nodes that are further away are assigned with higher probability to be connected. This strategy aims at diversity to increase the information gain.

### 4.5 Measuring improvement

We conclude with the definition of an evaluation measure, with which we measure the performance of the variations of the SHIKI approach. We measure performance in terms of improvement over a basic MLP. For that we calculate expectation of each loss (eqs. 5 and 6) and compare (Eq. 7).

$$\mathbb{E}_{X \sim XOR\text{-}GMM(n,d,\mu,\nu,\sigma^2)}(L_{MLP}) \tag{5}$$

$$\mathbb{E}_{X \sim XOR\text{-}GMM(n,d,\mu,\nu,\sigma^2)}(L_{SHIKI}) \tag{6}$$

$$imp \equiv \mathbb{E}_{X \sim XOR\text{-}GMM(n,d,\mu,\nu,\sigma^2)}(L_{MLP}\text{-}L_{SHIKI})$$
$$= \mathbb{E}_{X \sim XOR\text{-}GMM(n,d,\mu,\nu,\sigma^2)}(L_{MLP}) - \mathbb{E}_{X \sim XOR\text{-}GMM(n,d,\mu,\nu,\sigma^2)} \tag{7}$$

Intuitively speaking, when dealing with an easy data, the task is easy enough for MLP to succeed on its own, and adding GNN does not affect the performance. Also, when the data is too difficult, creating the edges using self-supervision is not effective, and the GNN may even worsen the performance as compared to simply applying MLP. Thus, we seek to identify the region of improvement, where the data is difficult enough for a plain MLP to perform quite poorly, yet sufficiently easy for a GNN to perform well and boost performance. This is controlled by three parameters, namely $n, \gamma, \sigma^2$.

We also expect the improvement to depend on SHIKI's hyper-parameters, namely $\tau$ or $percent$. In what follows we shall discuss only the impact of $\tau$. When $\tau$ is too small, we wind up considering all nodes, which we expect to lead to poor performance. On the other hand, when $\tau$ too large, we barely choose any node, limiting the impact of the GNN.

When analyzing the losses, we use two measure. $\alpha_\tau$ is the probability of getting the right prediction given a confident node and $\beta_\tau$ represents the probability of a node being confident.

**Theorem 1.** *Let $X \sim XOR\text{-}GMM(n, d, \mu, \nu, \sigma^2)$, with the edge creation created as above we have:*

1. $\mathbb{E}_X(L_{MLP}) = 2\sqrt{2}\sigma^2\phi(\frac{\gamma'}{\sqrt{2}\sigma}) - 2\sigma^2\phi(\frac{\gamma'}{\sigma})$

2. $\mathbb{E}_X(L_{SHIKI}) = P(confident) \cdot \mathbb{E}_X(L_{GNN}) + \mathbb{E}_X(L_{MLP}|not\ confident)$
   $$\mathbb{E}_X(L_{SHIKI}) \approx \beta_\tau \cdot exp\left(-\frac{p-q}{p+q}\frac{2\gamma'^2}{\sigma}(2\alpha_\tau - 1)^2\right) +$$
   $$\left(2\sigma^2\phi(\frac{\gamma'}{\sigma})(2(1 - \Phi(\frac{\tau}{\sigma})) - 1) +\right.$$
   $$\left. 2\sqrt{2}\sigma^2\left(\phi(\frac{\gamma'}{\sqrt{2}\sigma}) - \phi(\frac{-\gamma'+\tau}{\sqrt{2}\sigma})\Phi(\frac{-\gamma'-\tau}{\sqrt{2}\sigma}) - \phi(\frac{\gamma'+\tau}{\sqrt{2}\sigma})\Phi(\frac{\gamma'-\tau}{\sqrt{2}\sigma})\right)\right)$$

Where $\phi(x)$ and $\Phi(x)$ denote the pdf and cdf of a standard Gaussian.

**Proof sketch** To calculate the expected MLP loss, we use the plain definition of probability times value, calculating the probability followed by the expectation. To calculate our method's loss, we first separate the loss to the MLP part and the GNN part. For the MLP, we calculate the expectation similarly to the first part. For the GNN part, we know from Baranwal et al. (2022) its value. Then all is left is combining theses in the right way. □

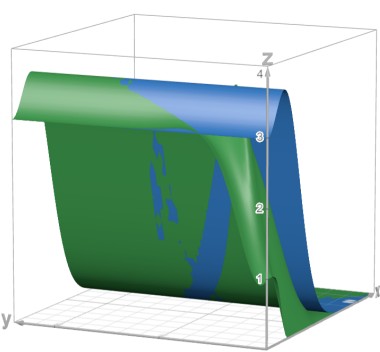

Figure 5: MLP vs SHIKI loss, blue - MLP, green - SHIKI

We next visualize in Figure 5 the losses' behavior by plotting the loss of each model, as a function of x-axis = $\frac{\gamma'}{\sigma}$, y-axis = $\frac{\tau}{\sigma}$, with $\sigma = 5$. As observed, SHIKI's expected loss is generally lower than that of the MLP, with the exception of instances where the y-axis values become significantly large. Even in these cases, it suggests a stronger integration of the GNN is necessary. Additionally, this may be due to the fact that the losses are only approximated.

## 5 EXPERIMENTS

In this section, we present empirical evidence to support our claims that the SHIKI model outperforms the standard MLP.

**Datasets:** We verify our method's result for both the real-world datasets, tailored for node classification tasks and the synthetic data model. Both intuitively and formally, we wish to choose a large $p$ value and a small $q$ value, to connect more nodes of the same class. Thus, in our experiments, we focus on this case, and choose $p \in \{0.7, 0.8\}, q \in \{0.1, 0.2\}$. Furthermore, we use with $\tau \in \{0.7, 0.8, 0.9\}, percent \in \{0.1, 0.2\}$.

In terms of real-world datasets, we tested the SHIKI model on the Amazon reviews and the Walmart-Amazon datasets (see Section 2). The structure of the data cannot be controlled to conform in full to our proposed model and the controlled parameters are $p, q, \tau$ and $percent$.

For the Walmart-Amazon data, we use DITTO Li et al. (2020), a state-of-the-art tool for entity matching, using RoBERTa Liu et al. (2019). The extracted pair embeddings serve us in the training of the MLP and the GNN.

By controlling the distance between the means, as demonstrated in Section 2, we separate the experiments with the synthetic data into two regimes, namely *hard* and *easy*. In the hard case, the distance spans from .3 to 1.5, with jumps of .15. In the easy case, the distance spans from 1.5 to 5, with jumps of .3. For the hard case, we use f-score, to prevent the learner from simply classifying all the data points as the same class. For the easy case, it is suffice to check for accuracy.

For set splits, we used train/test splits with the bigger subset used for training.

**Baseline:** As a baseline, we compare SHIKI to a popular graph creation heuristic, namely KNN (k-nearest-neighbors), where we connect each node to its k closest nodes.

**Evaluation measures:** For evaluation, we use three evaluation measures. The mean number of edges constructed by the method, the mean improvement over the MLP and standard deviation of the improvement. For the improvement, we applied all of SHIKI's strategies described in 4.4, and chose the best one.

**System Details:** All experiments use PyTorch Geometric (Fey & Lenssen, 2019) and were performed on a server with 2 NVIDIA GeForce GTX 1080 Ti and a Rocky Linux release 9.4 (Blue Onyx) operating system. Networks were implemented using PyTorch Paszke et al. (2019) and PyTorch Geometric (Fey & Lenssen, 2019).

### 5.1 RESULTS

We present next partial results of our. Due to space limitations, we present the results for the Walmart-Amazon dataset and results for the hard $XOR\text{-}GMM$ model case only. The Amazon reviews dataset results the analysis of the easy case are given in Appendix E.

| | SHIKI | knn | No confident nodes | No labels | No confident nodes and labels |
|---|---|---|---|---|---|
| **GCN2** | 1179941.125, 0.541 + 0.032 $\pm$ 0.058 | 31355.0, 0.557 + -0.05 $\pm$ 0.057 | 1813669.875, 0.537 + -0.005 $\pm$ 0.044 | 921473.4, 0.528 + **0.05 $\pm$ 0.058** | 1491614.5, 0.53 + -0.015 $\pm$ 0.021 |
| **GraphSAGE2** | 898606.35, 0.554 + **0.122 $\pm$ 0.051** | 31355.0, 0.59 + -0.018 $\pm$ 0.033 | 1479780.25, 0.608 + 0.071 $\pm$ 0.062 | 704871.5, 0.571 + 0.104 $\pm$ 0.077 | 918501.0, 0.58 + 0.09 $\pm$ 0.042 |
| **GAT2** | 1288287.275, 0.538 + **0.057 $\pm$ 0.071** | 31355.0, 0.545 + 0.035 $\pm$ 0.059 | 2711898.75, 0.519 + **0.106 $\pm$ 0.016** | 926080.4, 0.53 + 0.078 $\pm$ 0.068 | 1455389.5, 0.535 + **0.08 $\pm$ 0.071** |

Table 2: Improvement of the SHIKI method with different strategies on multiple GNN types on the Walmart-Amazon dataset.

|  | SHIKI | knn | No confident nodes | No labels | No confident nodes and labels |
|---|---|---|---|---|---|
| GCN2 | 37179.719, 0.319 + **0.213 ± 0.087** | 9137.0, 0.323 + **0.23 ± 0.071** | 143112.812, 0.302 + 0.137 ± 0.071 | 17521.375, 0.342 + 0.189 ± 0.077 | 104540.125, 0.363 + 0.109 ± 0.025 |
| GraphSAGE2 | 54268.725, 0.395 + **0.181 ± 0.218** | 9137.0, 0.516 + -0.086 ± 0.193 | 116882.531, 0.32 + **0.243 ± 0.277** | 44714.9, 0.387 + **0.182 ± 0.216** | 73274.5, 0.323 + **0.243 ± 0.299** |
| GAT2 | 43145.069, 0.363 + **0.225 ± 0.211** | 9137.0, 0.385 + -0.129 ± 0.287 | 192782.938, 0.41 + 0.204 ± 0.3 | 35336.525, 0.387 + 0.204 ± 0.204 | 142004.0, 0.409 + 0.211 ± 0.322 |

Table 3: Improvement of the SHIKI method with different strategies on multiple GNN types on the hard $XOR\text{-}GMM$ synthetic model.

Tables 2 and 3 present the results for the Walmart-Amazon and $XOR\text{-}GMM$ synthetic model data sets, respectively. Each row represents a different GNN architecture (GCN, GraphSAGE, GAT), where the GNN layer is only available at the second layer. Consider for now the first two columns in the table, representing our full SHIKI model and the KNN baseline. Each cell corresponds to a certain GNN architecture and a specific edge creation method. In each cell we present the mean number of edges constructed with the method, the mean MLP accuracy, and the mean and standard deviation of the improvement in the following format: $\#edges, mean\ MLP\ accuracy + mean\ improvement \pm improvement\ standard\ deviation$.

Best performing algorithm, in terms of accuracy improvement is marked in bold. It is evident that SHIKI consistently outperforms both MLP and KNN.

Additionally, SHIKI demonstrates consistent performance across all GNN architectures, with only slight variations in their results. A more detailed discussion is provided in the Appendix E.

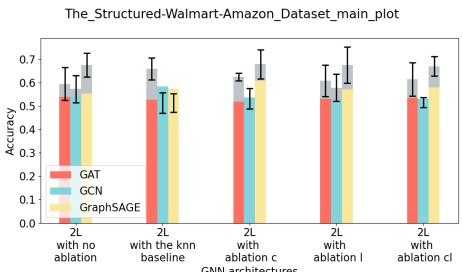 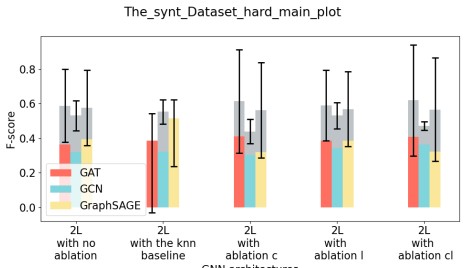

(a) Plot of the SHIKI model improvement across all ablations and baselines on a real-world dataset.

(b) Plot of the SHIKI model improvement across all ablations and baselines on a synthetic dataset.

Figure 6: Plots of real-world and synthetic data comparing SHIKI to multiple ablations and baselines.

The results are also presented visually (Figure 6) for ease of understandings. Similarly to the tables, the first bar group is our full SHIKI model and the second is the KNN baseline. Each bar group consists of the three GNN architectures. The colored bar represents the mean MLP accuracy with its corresponding GNN architecture. The gray bar above represents the mean improvement (no such bar means no improvement) with the standard deviation as the black line. The visual representation provides a clear illustration of SHIKI's superior performance over the baseline.

We also provide an ablation study, using three different variation, as follows.

**No confident nodes:** Instead of taking only the most confident nodes, we take all of the nodes, and apply edge creation considering all of the nodes.

**No labels:** Instead of considering the weak labels of the most confident nodes, for each pair of nodes, we create an edge between them with probability of 0.5. This is done by setting $p = q = 0.5$.

**No labels, No confident nodes:** We take all nodes and for each pair of node, we create an edge between them with probability of 0.5, rendering the weak-labels useless. Note that we randomly create edges between nodes.

The three right-most columns of tables 2 and 3 and three right-most sets of bars of each of the graphs in Figure 6 provide the ablation study analysis. Clearly, SHIKI outperforms its subsets, justifying the use of confident nodes and applying spatial consideration when generating edges.

## 6 RELATED WORK

There is a significant body of theoretical work on unsupervised learning for random graph models where node features are absent, and only relational information is available (Decelle et al., 2011; Massoulié, 2014; Mossel et al., 2018; 2015; Abbe & Sandon, 2015; Abbe et al., 2015; Bordenave et al., 2015; Deshpande et al., 2015; Montanari & Sen, 2016; Banks et al., 2016; Abbe & Sandon, 2018; Li et al., 2019; Kloumann et al., 2017; Gaudio et al., 2022). In contrast, for data models that include both node features and relational information, numerous studies have addressed the semi-supervised node classification problem, such as (Scarselli et al., 2009; Cheng et al., 2011; Gilbert et al., 2012; Dang & Viennet, 2012; Günnemann et al., 2013; Yang et al., 2013; Jin et al., 2019; Mehta et al., 2019; Chien et al., 2022; Yan et al., 2021). These works offer valuable empirical insights into the benefits of incorporating graph structure. Our study addresses a slightly different settings, where node features are available, yet relational information is missing.

In Deshpande et al. (2018); Lu & Sen (2020), the authors investigate the fundamental thresholds for classifying a significant portion of nodes with linear sample complexity and large, but finite, degrees. In Fountoulakis et al. (2022), the authors present a theoretical analysis of the graph attention mechanism (GAT), identifying the conditions under which the attention mechanism is effective (or not) for node classification tasks. Our research, however, focuses on graph convolutions rather than attention-based methods. While several studies examine the expressive power, extrapolation, and the oversmoothing phenomenon in GNNs (see, e.g., Balcilar et al. (2021); Xu et al. (2021); Oono & Suzuki (2020); Li et al. (2018)), we aim to compare the strengths and limitations of graph convolutions with those of traditional MLPs when both do not leverage built-in relational information.

In Li et al. (2024); Chen et al. (2023), the authors also utilize artificial edges using standard k-nearest-neighbors procedure in their node-classification process. However, their setting still requires an existing built-in graph, while we focus on constructing the graph.

## 7 CONCLUSION AND OPEN QUESTIONS

In this work, we defined the challenge of effectively constructing edges within a dataset for improved training using GNNs and introduced a novel method to tackle this task. We formally shown and empirically demonstrated how graph convolutions could improve expected performance by leveraging these created edges. The results were empirically confirmed through extensive experiments on both synthetic and real-world datasets, including those involving the entity matching problem and text prediction.

Our analysis is limited to the SHIKI heuristic. Other heuristics will require a new analysis. Furthermore, we did not solve or claimed to solve the optimality problem (Problem 2). Thus, in future work we intend to investigate effective solutions to this problem by either finding the optimal graph, or showing a way to optimize the task directly.

Finally, our analysis is limited to graph convolution. A possible future direction is to test whether our theoretical insights also apply to graph attention networks (GAT).

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

## A    CALCULATIONS FOR THE MAIN RESULTS

### A.1    ASSUMPTIONS AND NOTATION

**Assumption 1.** *We use similar assumptions and notation as in (Baranwal et al., 2022). For all the variations of the $XOR$-$GMM$ data model variations, the means of the Gaussian mixture are such that $\langle \mu, \nu \rangle = 0$ and $\|\mu\|_2 = \|\nu\|_2$.*

We denote $[x]_+ = RELU(x)$ and $\varphi(x) = \text{sigmoid}(x) = \frac{1}{1+e^{-x}}$, applied element-wise on the inputs. For any vector $\boldsymbol{v}$, $\hat{v} = \frac{v}{\|v\|_2}$ denotes the normalized $\boldsymbol{v}$. We use $\gamma = \|\mu - \nu\|_2$ to denote the distance between the means of the inter-class components of the mixture model, and $\gamma'$ to denote the norm of the means, $\gamma' = \frac{\gamma}{\sqrt{2}} = \|\mu\|_2 = \|\nu\|_2$.

We present the calculations for the algorithm's version with the threshold $\tau$, noting the case with the top percentile is more complicated, and serves no additional purpose.

### A.2    DATA DIFFICULTY

#### A.2.1    PERFECT CLASSIFIER

Assuming $\gamma' = \Omega(\sigma(\log n)^{\frac{1}{2}+\epsilon})$, then, according to Baranwal et al. (2022), both MLP and GNN perfectly classifies the data w.h.p. Thus, rendering this scenario not interesting since our model won't be able to improve the loss and performance of an MLP.

#### A.2.2    PARTIALLY RIGHT

In order for our method to succeed in improving the performance, we need the MLP to be wrong a meaningful percentage of the times. Thus, we would look at the case where $\gamma' = \Omega(K\sigma)$. In this case, the MLP is bound to make mistakes. And for the GNN we have $\Omega\left(\frac{\sigma\sqrt{\log n}}{\sqrt[4]{n(p+q)}}\right) \leq \Omega(\sigma K)$ when $p, q = \Omega(\frac{\log^2 n}{n})$. Thus, the GNN is expected to classify the data well. Meaning, we are most likely to improve in this area.

### A.3    MLP LOSS

First, let's exactly calculate the expected MLP loss.
Define $z_i = |\langle x_i, \hat{\mu}\rangle| - |\langle x_i, \hat{\nu}\rangle|$, the expected MLP loss will be:

$$\mathbb{E}(L_{MLP}) = \int_{-\infty}^{\infty} p(z = t)L(t)dt$$

We also have:

$$\mathbb{E}(L_{MLP}|x \sim |\mu|) = \int_{-\infty}^{\infty} p(z = t|x \sim |\mu|)L(t|x \sim |\mu|)dt$$

Notice that when $(z|x \sim |\mu|) > 0$, we are right in our prediction, subsequently, the loss approaches 0, thus we will ignore this case. Also, note that for $(z|x \sim |\mu|) < 0$, since we use the cross-entropy loss, we have $L(z|x \sim |\mu|) = \ln(1 + e^{-z}) \approx -z$.
Finally, due to symmetry we have:

$$\mathbb{E}(L_{MLP}) = \mathbb{E}(L_{MLP}|x \sim |\mu|) \approx \int_{-\infty}^{0} p(z = t|x \sim |\mu|) \cdot (-t)dt$$

We will need some more auxiliary calculations to help us in the way.

### A.4    LEMMAS

Here we prove some basic lemmas to help us calculate the loss.
Let's define $A = \frac{-\gamma'}{\sigma}, B = \frac{t}{\sigma}, C = A - B = \frac{-\gamma'}{\sigma} - \frac{t}{\sigma}, C' = -A - B = \frac{\gamma'}{\sigma} - \frac{t}{\sigma}, b' = max(-B, 0)$.

**Lemma 1.**

$$P(z_i = t | z_i \sim |\mu|) = \sqrt{2}\phi(\frac{-\gamma'}{\sigma\sqrt{2}} - \frac{t}{\sigma\sqrt{2}})\Phi(\frac{-\gamma'}{\sigma\sqrt{2}} + \frac{t}{\sigma\sqrt{2}}) +$$

$$\sqrt{2}\phi(\frac{\gamma'}{\sigma\sqrt{2}} - \frac{t}{\sigma}\sqrt{2})\Phi(\frac{\gamma'}{\sigma\sqrt{2}} + \frac{t}{\sigma\sqrt{2}})$$

*Proof.* First notice $X_i = \mu + \sigma g_i$. In order to exactly calculate the probability, we will separate it to the cases $\langle X_i, \mu \rangle \geq 0$ and when $\langle X_i, \mu \rangle < 0$.

$$P(z_i = t | z_i \sim |\mu|) = P(|\langle X_i, \mu \rangle| - |\langle X_i, \nu \rangle| = t) = P(|\gamma' + \sigma\langle g_i, \hat{\mu}_i \rangle| - \sigma|\langle g_i, \hat{\nu}_i \rangle| = t) =$$
$$P(\gamma' + \sigma\langle g_i, \hat{\mu}_i \rangle - \sigma|\langle g_i, \hat{\nu}_i \rangle| = t)P(\langle g_i, \hat{\mu}_i \rangle \geq \frac{-\gamma'}{\sigma}) + P(-\gamma' - \sigma\langle g_i, \hat{\mu}_i \rangle - \sigma|\langle g_i, \hat{\nu}_i \rangle| =$$
$$t)P(\langle g_i, \hat{\mu}_i \rangle \leq \frac{-\gamma'}{\sigma}) =$$
$$P(\frac{\gamma'-t}{\sigma} = |\langle g_i, \hat{\nu} \rangle| - \langle g_i, \hat{\mu} \rangle)P(-\langle g_i, \hat{\mu}_i \rangle \leq \frac{\gamma'}{\sigma}) + P(\frac{-\gamma'-t}{\sigma} = |\langle g_i, \hat{\nu} \rangle| + \langle g_i, \hat{\mu} \rangle)P(\langle g_i, \hat{\mu}_i \rangle \leq \frac{-\gamma'}{\sigma})$$

This expression contains 2 sub-expressions within it. We will calculate the first half, and the second half will be very similar.
We now define random variables $Z_1 = \langle g_i, \hat{\nu} \rangle$ and $Z_2 = \langle g_i, \hat{\mu} \rangle$ and note that $Z_1, Z_2 \sim N(0, 1)$ and $\mathbb{E}[Z_1 Z_2] = 0$. We have:

$$P(\frac{\gamma'-t}{\sigma} = |\langle g_i, \hat{\nu} \rangle| - \langle g_i, \hat{\mu} \rangle | \langle g_i, \hat{\mu} \rangle \geq \frac{-\gamma'}{\sigma}) = P(|Z_1| - Z_2 = A - B | Z_2 \geq -A) = 2P(Z_1 - Z_2 =$$
$$A - B, Z_1 \geq 0 | Z_2 \geq -A) = 2\int_{b'}^{\infty} \phi(w)P(-Z_2 = A - B - w | -Z_2 \leq A)dw =$$
$$2\int_{b'}^{\infty} \phi(w)\frac{P(-Z_2 = C - w)}{P(Z_2 \leq A)}dw = \frac{2}{P(Z_2 \leq A)}\int_{b'}^{\infty} \phi(w)P(Z_2 = C - w) =$$
$$\frac{2}{P(Z_2 \leq A)}\int_{-\infty}^{-b'} \phi(w)\phi(C + w)dw = \frac{\sqrt{2}\phi(\frac{C}{\sqrt{2}})}{\Phi(A)}\Phi(-\sqrt{2}b' + \frac{C}{\sqrt{2}})$$

Where $b' = max(-B, 0)$.
Second to last equality is change of parameters, last equality, to evaluate the integral above, we used Owen (1980), Table 1:110.
Similarly for the second expression:

$$P(\frac{-\gamma'-t}{\sigma} = |\langle g_i, \hat{\nu} \rangle| + \langle g_i, \hat{\mu} \rangle | \langle g_i, \hat{\mu} \rangle \leq \frac{-\gamma'}{\sigma}) = P(|Z_1| + Z_2 = -A - B | Z_2 \leq -A) =$$
$$P(|Z_1| - (-Z_2) = A' - B | -Z_2 \geq -A') = \frac{\sqrt{2}\phi(\frac{C'}{\sqrt{2}})}{\Phi(A')}\Phi(-\sqrt{2}b' + \frac{C'}{\sqrt{2}}) = \frac{\sqrt{2}\phi(\frac{C'}{\sqrt{2}})}{\Phi(-A)}\Phi(-\sqrt{2}b' + \frac{C'}{\sqrt{2}})$$

Summing those two expression, we get:

$$p(z_i = t | z_i \sim |\mu|) = P(\frac{\gamma'-t}{\sigma} = |\langle g_i, \hat{\nu} \rangle| - \langle g_i, \hat{\mu} \rangle)P(-\langle g_i, \hat{\mu}_i \rangle \leq \frac{\gamma'}{\sigma}) + P(\frac{-\gamma'-t}{\sigma} =$$
$$|\langle g_i, \hat{\nu} \rangle| + \langle g_i, \hat{\mu} \rangle)P(\langle g_i, \hat{\mu}_i \rangle \leq \frac{-\gamma'}{\sigma}) =$$
$$\frac{\sqrt{2}\phi(\frac{C}{\sqrt{2}})}{\Phi(A)}\Phi(-\sqrt{2}b' + \frac{C}{\sqrt{2}}) \cdot \Phi(A) + \frac{\sqrt{2}\phi(\frac{C'}{\sqrt{2}})}{\Phi(-A)}\Phi(-\sqrt{2}b' + \frac{C'}{\sqrt{2}}) \cdot \Phi(-A) =$$
$$\sqrt{2}\phi(\frac{C}{\sqrt{2}})\Phi(-\sqrt{2}b' + \frac{C}{\sqrt{2}}) + \sqrt{2}\phi(\frac{C'}{\sqrt{2}})\Phi(-\sqrt{2}b' + \frac{C'}{\sqrt{2}}) =$$
$$\sqrt{2}\phi(\frac{\frac{-\gamma'}{\sigma} - \frac{t}{\sigma}}{\sqrt{2}})\Phi(\frac{t}{\sigma}\sqrt{2} + \frac{\frac{-\gamma'}{\sigma} - \frac{t}{\sigma}}{\sqrt{2}}) + \sqrt{2}\phi(\frac{\frac{\gamma'}{\sigma} - \frac{t}{\sigma}}{\sqrt{2}})\Phi(\frac{t}{\sigma}\sqrt{2} + \frac{\frac{\gamma'}{\sigma} - \frac{t}{\sigma}}{\sqrt{2}}) =$$
$$\sqrt{2}\phi(\frac{-\gamma'}{\sigma\sqrt{2}} - \frac{t}{\sigma\sqrt{2}})\Phi(\frac{-\gamma'}{\sigma\sqrt{2}} + \frac{t}{\sigma\sqrt{2}}) + \sqrt{2}\phi(\frac{\gamma'}{\sigma\sqrt{2}} - \frac{t}{\sigma}\sqrt{2})\Phi(\frac{\gamma'}{\sigma\sqrt{2}} + \frac{t}{\sigma\sqrt{2}})$$

$\square$

To give some intuition to how this expression behaves, we will plot it as a function of $t$, where $\gamma' = \sigma = 1$:

**Lemma 2.** *For $t \geq 0$:*

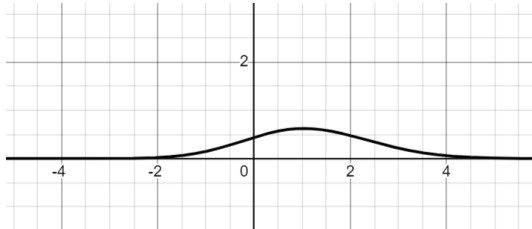

Figure 7: Lemma 1's expression, we can observe that when $t < 0$ or when $t$ is pretty big, this expression approaches 0, suggesting the most likely case is for the expression to be somewhat positive

$$P(z_i > t | z_i \sim |\mu|) = \Phi(\tfrac{A-B}{\sqrt{2}})^2 + \Phi(\tfrac{-A-B}{\sqrt{2}})^2$$

*For $t < 0$:*

$$P(z_i > t | z_i \sim |\mu|) \approx$$
$$2\Phi(-B) - 1 - 2(\tfrac{1}{2} - \Phi(B))(\Phi(A - B) + \Phi(-A - B)) + \Phi(\tfrac{A-B}{\sqrt{2}})^2 + \Phi(\tfrac{-A-B}{\sqrt{2}})^2$$

*Proof.* We will start the proof for both cases similarly to A.4, using the same notations:

$$P(z > t | z \sim |\mu|) = P(|\gamma' + \sigma\langle g_i, \hat{\mu}_i\rangle| \geq \sigma|\langle g_i, \hat{\nu}_i\rangle| + t) = P(\gamma' + \sigma\langle g_i, \hat{\mu}_i\rangle \geq$$
$$\sigma|\langle g_i, \hat{\nu}_i\rangle + t||\langle g_i, \hat{\mu}_i\rangle > \tfrac{-\gamma'}{\sigma})P(\langle g_i, \hat{\mu}_i\rangle > \tfrac{-\gamma'}{\sigma})) + P(-\gamma' - \sigma\langle g_i, \hat{\mu}_i\rangle \geq \sigma|\langle g_i, \hat{\nu}_i\rangle| + t|\langle g_i, \hat{\mu}_i\rangle <$$
$$\tfrac{-\gamma'}{\sigma})P(\langle g_i, \hat{\mu}_i\rangle < \tfrac{-\gamma'}{\sigma})) = P(\gamma' - t \geq \sigma|\langle g_i, \hat{\nu}_i\rangle| - \sigma\langle g_i, \hat{\mu}_i\rangle|\langle g_i, \hat{\mu}_i\rangle > \tfrac{-\gamma'}{\sigma})P(\langle g_i, \hat{\mu}_i\rangle >$$
$$\tfrac{-\gamma'}{\sigma})) + P(-\gamma' - t \geq \sigma|\langle g_i, \hat{\nu}_i\rangle| + \sigma\langle g_i, \hat{\mu}_i\rangle|\langle g_i, \hat{\mu}_i\rangle < \tfrac{-\gamma'}{\sigma})P(\langle g_i, \hat{\mu}_i\rangle < \tfrac{-\gamma'}{\sigma}))$$

We will now separate the calculations depending on the sign of $t$.
For $t \geq 0$, again separating the expression, we get:

$$P(|Z_1| - Z_2 \leq A - B | Z_2 \geq -A) = 2P(Z_1 - Z_2 \leq A - B | Z_2 \geq -A) = 2\int_0^\infty \phi(w)P(-Z_2 \leq$$
$$A - B - w | Z_2 \geq -A)dw = 2\int_0^\infty \phi(w)\frac{P(-Z_2 \leq A-B-w)}{P(-Z_2 \leq A)}dw = \frac{2}{\Phi(A)}\int_0^\infty \phi(w)\Phi(A - B - w)dw =$$
$$\frac{2}{\Phi(A)}\int_{-\infty}^0 \phi(w)\Phi(A - B + w)dw = \frac{2}{\Phi(A)} \cdot \frac{\Phi(\frac{A-B}{\sqrt{2}})^2}{2} = \frac{\Phi(\frac{A-B}{\sqrt{2}})^2}{\Phi(A)}$$

Where we evaluate the integral using Owen (1980) Table 1:10,010.7. For the second sub-expression a very similar calculation can be done. Combining both expressions we get:

$$P(z > \tau | z \sim |\mu|) = \Phi(A) \cdot \frac{\Phi(\frac{A-B}{\sqrt{2}})^2}{\Phi(A)} + \Phi(-A) \cdot \frac{\Phi(\frac{-A-B}{\sqrt{2}})^2}{\Phi(-A)} = \Phi(\tfrac{A-B}{\sqrt{2}})^2 + \Phi(\tfrac{-A-B}{\sqrt{2}})^2$$

For $t < 0$, separating the expression, we get:

$$P(|Z_1| - Z_2 \leq A - B | Z_2 \geq -A) = 2P(Z_1 - Z_2 \leq A - B | Z_2 \geq -A) = 2\int_0^\infty \phi(w)P(-Z_2 \leq$$
$$A - B - w | Z_2 \geq -A)dw = 2\int_0^{-B} \phi(w)P(-Z_2 \leq A - B - w | Z_2 \geq -A)dw +$$
$$2\int_{-B}^\infty \phi(w)P(-Z_2 \leq A - B - w | Z_2 \geq -A)dw = 2\int_0^{-B} \phi(w)1dw + 2\int_{-B}^\infty \phi(w)P(-Z_2 \leq$$
$$A - B - w | Z_2 \geq -A) = 2\left((\Phi(-B) - \tfrac{1}{2}) + \tfrac{1}{\Phi(A)}\int_{-\infty}^B \phi(w)\Phi(A - B + w)dw\right)$$

where we changed the probability to 1 because:

$$P(-Z_2 \leq A - B - w | Z_2 \geq -A) = P(-Z_2 \leq A + (-B) - w | -Z_2 \leq A)$$
$$0 < w < -B \rightarrow B < -w < 0 \rightarrow 0 < -B - w < -B \rightarrow A < A + (-B) - w < A - B \rightarrow$$
$$(-Z_2 \leq A \rightarrow -Z_2 \leq A + (-B) - w)$$

Unfortunately, we can't directly evaluate this integral, since as seen in Owen (1980) Table 1:10,010.4, this expression doesn't have a closed form, so we will result to approximate it.

$$2\left((\Phi(-B) - \tfrac{1}{2}) + \tfrac{1}{\Phi(A)}\int_{-\infty}^{B}\phi(w)\Phi(A-B+w)dw\right) =$$
$$2\left((\Phi(-B) - \tfrac{1}{2}) + \tfrac{1}{\Phi(A)}\left(\int_{-\infty}^{0}\phi(z)\Phi(A-B+w)dw - \int_{B}^{0}\phi(w)\Phi(A-B+w)dw\right)\right) \approx$$
$$2\left((\Phi(-B) - \tfrac{1}{2}) + \tfrac{1}{\Phi(A)}\left(\int_{-\infty}^{0}\phi(w)\Phi(A-B+w)dw - \int_{B}^{0}\phi(w)\Phi(A-B)dw\right)\right) =$$
$$2\left((\Phi(-B) - \tfrac{1}{2}) + \tfrac{1}{\Phi(A)}\left(\tfrac{1}{2}\Phi(\tfrac{A-B}{\sqrt{2}})^2 - \Phi(A-B)(\Phi(0) - \Phi(B))\right)\right) =$$
$$2\left((\Phi(-B) - \tfrac{1}{2}) + \tfrac{1}{\Phi(A)}\left(\tfrac{1}{2}\Phi(\tfrac{A-B}{\sqrt{2}})^2 - \Phi(A-B)(\tfrac{1}{2} - \Phi(B))\right)\right)$$

And the full expression:

$$P(z > \tau | z \sim |\mu|) = 2\left((\Phi(-B) - \tfrac{1}{2}) + \tfrac{1}{\Phi(A)}\left(\tfrac{1}{2}\Phi(\tfrac{A-B}{\sqrt{2}})^2 - \Phi(A-B)(\tfrac{1}{2} - \Phi(B))\right)\right) \cdot$$
$$\Phi(A) + 2\left((\Phi(-B) - \tfrac{1}{2}) + \tfrac{1}{\Phi(-A)}\left(\tfrac{1}{2}\Phi(\tfrac{-A-B}{\sqrt{2}})^2 - \Phi(-A-B)(\tfrac{1}{2} - \Phi(B))\right)\right) \cdot \Phi(-A) =$$
$$2\left(\Phi(A)(\Phi(-B) - \tfrac{1}{2}) + \left(\tfrac{1}{2}\Phi(\tfrac{A-B}{\sqrt{2}})^2 - \Phi(A-B)(\tfrac{1}{2} - \Phi(B))\right)\right) +$$
$$2\left(\Phi(-A)(\Phi(-B) - \tfrac{1}{2}) + \left(\tfrac{1}{2}\Phi(\tfrac{-A-B}{\sqrt{2}})^2 - \Phi(-A-B)(\tfrac{1}{2} - \Phi(B))\right)\right) =$$
$$2\Phi(-B) - 1 - 2(\tfrac{1}{2} - \Phi(B))(\Phi(A-B) + \Phi(-A-B)) + \Phi(\tfrac{A-B}{\sqrt{2}})^2 + \Phi(\tfrac{-A-B}{\sqrt{2}})^2$$

$$\square$$

To give some intuition to how this expressions behaves, we will plot them as a function of $t$, where $\gamma' = \sigma = 1$.

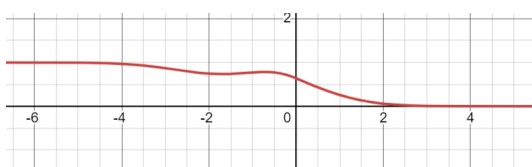

Figure 8: Lemma 2's expression as a combination of the two expressions in the respected cases. We can observe that generally speaking, this function is monotonically decreasing.

**Lemma 3.**

$$\beta_\tau = P(confident) \approx 2\Phi(\frac{A-B}{\sqrt{2}}) + 2\Phi(\frac{-A-B}{\sqrt{2}}) - 2\Phi(B) +$$
$$2(\frac{1}{2} - \Phi(-B))(\Phi(A+B) + \Phi(-A+B))$$

*Proof.* We will first do some intermediate calculations.
In Lemma 2 we calculated $P(z > \tau | z \sim |\mu|)$, now note:

$$P(z > \tau | z \sim |\nu|) = P(z < -\tau | z \sim |\mu|) = 1 - P(z > -\tau | z \sim |\mu|)$$

Now we will also calculate, $P(z > \tau)$:

$$P(z > \tau) = P(z > \tau | z \sim |\mu|)P(z \sim |\mu|) + P(z > \tau | z \sim |\nu|)P(z \sim |\nu|) = \tfrac{1}{2}\left(P(z > \tau | z \sim\right.$$
$$\left.|\mu|) + P(z > \tau | z \sim |\nu|)\right) = \tfrac{1}{2}\left(P(z > \tau | z \sim |\mu|) + 1 - P(z > -\tau | z \sim |\mu|)\right)$$

And now we can calculate $\beta_\tau$ using above expressions.

$$\beta_\tau = P(|z| > \tau) = P(z > \tau) + P(z < -\tau) = P(z > \tau) + (1 - P(z > -\tau)) =$$

$$\frac{1}{2}[P(z > \tau | z \sim |\mu|) + 1 - P(z > -\tau | z \sim |\mu|)] +$$

$$1 - \left(\frac{1}{2}[P(z > -\tau | z \sim |\mu|) + 1 - P(z > \tau | z \sim |\mu|)]\right) =$$

$$1 + P(z > \tau | z \sim |\mu|) - P(z > -\tau | z \sim |\mu|) \approx$$

$$1 + \left(\Phi(\frac{A-B}{\sqrt{2}})^2 + \Phi(\frac{-A-B}{\sqrt{2}})^2 -\right.$$

$$\left.\left(2\Phi(B) - 1 - 2(\frac{1}{2} - \Phi(-B))(\Phi(A+B) + \Phi(-A+B)) + \Phi(\frac{A+B}{\sqrt{2}})^2 + \Phi(\frac{-A+B}{\sqrt{2}})^2\right)\right) =$$

$$1 + \left(\Phi(\frac{A-B}{\sqrt{2}})^2 + \Phi(\frac{-A-B}{\sqrt{2}})^2 - 2\Phi(B) + 1 +\right.$$

$$\left.2(\frac{1}{2} - \Phi(-B))(\Phi(A+B) + \Phi(-A+B)) - \Phi(\frac{A+B}{\sqrt{2}})^2 - \Phi(\frac{-A+B}{\sqrt{2}})^2\right) =$$

$$2 + \left(\Phi(\frac{A-B}{\sqrt{2}})^2 - \Phi(\frac{A+B}{\sqrt{2}})^2 + \Phi(\frac{-A-B}{\sqrt{2}})^2 -\right.$$

$$\left.\Phi(\frac{-A+B}{\sqrt{2}})^2 - 2\Phi(B) + 2(\frac{1}{2} - \Phi(-B))(\Phi(A+B) + \Phi(-A+B))\right) =$$

$$2 + \left(2\Phi(\frac{A-B}{\sqrt{2}}) - 1 + 2\Phi(\frac{-A-B}{\sqrt{2}}) - 1 - 2\Phi(B) + 2(\frac{1}{2} - \Phi(-B))(\Phi(A+B) + \Phi(-A+B))\right) =$$

$$2\Phi(\frac{A-B}{\sqrt{2}}) + 2\Phi(\frac{-A-B}{\sqrt{2}}) - 2\Phi(B) + 2(\frac{1}{2} - \Phi(-B))(\Phi(A+B) + \Phi(-A+B))$$

$\square$

To give some intuition to how this expression behaves, we will plot it as a function of $t$, where $\gamma' = \sigma = 1$.

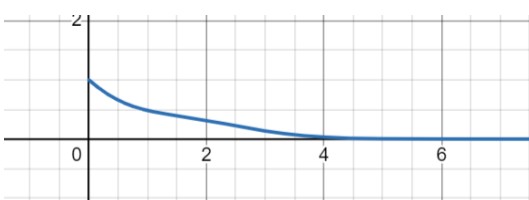

Figure 9: Lemma 3's expression where $\tau \geq 0$. We can observe that generally speaking, this function is monotonically decreasing, and its maximum value is 1 when $\tau = 0$.

**Lemma 4.** $\alpha_\tau = P(right\ classification | confident) \approx$

$$\frac{\Phi(\frac{\gamma'}{\sigma\sqrt{2}} - \frac{\tau}{\sigma\sqrt{2}})^2 + \Phi(-\frac{\gamma'}{\sigma\sqrt{2}} - \frac{\tau}{\sigma\sqrt{2}})^2}{2\Phi(\frac{\frac{\gamma'}{\sigma} - \frac{\tau}{\sigma}}{\sqrt{2}}) + 2\Phi(\frac{-\frac{\gamma'}{\sigma} - \frac{\tau}{\sigma}}{\sqrt{2}}) - 2\Phi(\frac{\tau}{\sigma}) + 2(\frac{1}{2} - \Phi(-\frac{\tau}{\sigma}))(\Phi(\frac{\gamma'}{\sigma} + \frac{\tau}{\sigma}) + \Phi(-\frac{\gamma'}{\sigma} + \frac{\tau}{\sigma}))}$$

*Proof.* Notice that

$$\alpha_\tau = P(right\ classification|confident) = P(right\ classification||z| > \tau) =$$
$$\frac{P(|z|>\tau|right\ classification)P(right\ classification)}{P(|z|>\tau)} =$$
$$\frac{P(|z|>\tau|right\ classification,z\sim|\mu|)P(right\ classification|z\sim|\mu|)}{\beta_\tau} = \frac{P(z>\tau|z>0,z\sim|\mu|)P(z>0|z\sim|\mu|)}{\beta_\tau} =$$
$$\frac{\frac{P(z>\tau,z>0|z\sim|\mu|)}{P(z>0|z\sim|\mu|)}P(z>0|z\sim|\mu|)}{\beta_\tau} = \frac{P(z>\tau|z\sim|\mu|)}{\beta_\tau}$$

Where we switch to $z \sim |\mu|$ due to symmetry, and right classification knowing this implies $z > 0$. But from Lemma 3 we have:

$$\beta_\tau \approx 2\Phi(\frac{\frac{\gamma'}{\sigma}-\frac{\tau}{\sigma}}{\sqrt{2}}) + 2\Phi(\frac{-\frac{\gamma'}{\sigma}-\frac{\tau}{\sigma}}{\sqrt{2}}) - 2\Phi(\frac{\tau}{\sigma}) + 2(\frac{1}{2} - \Phi(-\frac{\tau}{\sigma}))(\Phi(\frac{\gamma'}{\sigma} + \frac{\tau}{\sigma}) + \Phi(-\frac{\gamma'}{\sigma} + \frac{\tau}{\sigma}))$$
$$P(z > \tau|z \sim |\mu|) = \Phi(\frac{\gamma'}{\sigma\sqrt{2}} - \frac{\tau}{\sigma\sqrt{2}})^2 + \Phi(-\frac{\gamma'}{\sigma\sqrt{2}} - \frac{\tau}{\sigma\sqrt{2}})^2$$

And dividing these two expression completes the proof. $\square$

**Lemma 5.** *Given we choose $p, q$ as in Baranwal et al. (2022), we do the edge creation process with the predicted labels instead of the real labels. Thus, we want to calculate the real $rp, rq$ that are actually being used.*

$$rp = P(edge\ between\ two\ inputs\ of\ the\ same\ class) = P(edge|inputs\ of\ the\ same\ class) =$$
$$P(edge|y_i = y_j) = P(\hat{y}_i = \hat{y}_i|y_i = y_j) * P(edge|\hat{y}_i = \hat{y}_j) + P(\hat{y}_i \neq \hat{y}_j|y_i = y_j) * P(edge|\hat{y}_i \neq \hat{y}_i) = p(\alpha_\tau{}^2 + (1 - \alpha_\tau)^2) + q(2\alpha_\tau(1 - \alpha_\tau))$$

*Proof.* Now we calculate $P(\hat{y}_i = \hat{y}_i|y_i = y_j)$ and $P(\hat{y}_i \neq \hat{y}_i|y_i = y_j)$. First:

$$P(\hat{y}_i = \hat{y}_i|y_i = y_j) = \alpha_\tau{}^2 + (1 - \alpha_\tau)^2$$
$$P(\hat{y}_i \neq \hat{y}_i|y_i = y_j) = 2\alpha_\tau(1 - \alpha_\tau)$$

Similarly,

$$rq = P(edge\ between\ two\ inputs\ of\ different\ class) =$$
$$P(edge|inputs\ of\ different\ class) = P(edge|y_i \neq y_j) = P(\hat{y}_i = \hat{y}_i|y_i \neq y_j) * P(edge|\hat{y}_i = \hat{y}_j) + P(\hat{y}_i \neq \hat{y}_j|y_i \neq y_j) * P(edge|\hat{y}_i \neq \hat{y}_i) = q(\alpha_\tau{}^2 + (1 - \alpha_\tau)^2) + p(2\alpha_\tau(1 - \alpha_\tau))$$

Now let's see how they are integrated with the GNN loss.

$$rp = p(\alpha_\tau{}^2 + (1 - \alpha_\tau)^2) + q(2\alpha_\tau(1 - \alpha_\tau)), rq = q(\alpha_\tau{}^2 + (1 - \alpha_\tau)^2) + p(2\alpha_\tau(1 - \alpha_\tau))$$
$$rp - rq = (p - q)\left(\alpha_\tau{}^2 + (1 - \alpha_\tau)^2 - 2\alpha_\tau(1 - \alpha_\tau)\right) = (p - q)\left(4\alpha_\tau{}^2 - 4\alpha_\tau + 1\right) =$$
$$(p - q)\left(2\alpha_\tau - 1\right)^2$$
$$rp + rq = p + q$$
$$\frac{rp-rq}{rp+rq} = \frac{p-q}{p+q}(2\alpha_\tau - 1)^2$$

$\square$

## A.5 LOSSES

Having calculated all the lemmas, we are finally ready to calculate the losses.

**Theorem** (Restatement of part one of Theorem 1). *The expected MLP loss is:*

$$\mathbb{E}_X(L_{MLP}) \approx 2\sqrt{2}\sigma^2\phi(\frac{\gamma'}{\sqrt{2}\sigma}) - 2\sigma^2\phi(\frac{\gamma'}{\sigma})$$

*Proof.* Recall that we have:

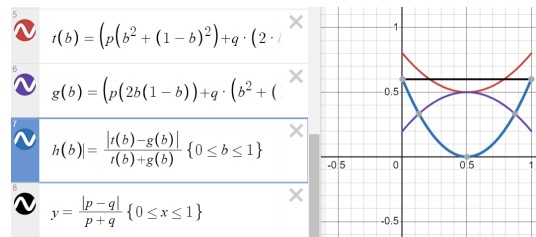

Figure 10: The expression in Lemma 5 as a function of $\alpha_\tau$, for $p = 0.8, q = 0.2$. The Black line is the expression with the original $p, q$, The Blue line is the expression with the real $p, q$. As we can see, as we get farther from $\alpha_\tau = 0.5$ (meaning we are more confident), the real expression gets closer to the original expression.

$$\mathbb{E}(L_{MLP}) \approx -\int_{-\infty}^{0} t \cdot p(z = t | z \sim |\mu|) dt$$

$$P(z = t | z \sim |\mu|) = \sqrt{2}\phi(\frac{-\gamma'}{\sigma\sqrt{2}} - \frac{t}{\sigma\sqrt{2}})\Phi(\frac{-\gamma'}{\sigma\sqrt{2}} + \frac{t}{\sigma\sqrt{2}}) + \sqrt{2}\phi(\frac{\gamma'}{\sigma\sqrt{2}} - \frac{t}{\sigma}\sqrt{2})\Phi(\frac{\gamma'}{\sigma\sqrt{2}} + \frac{t}{\sigma\sqrt{2}})$$

Let's define:

$$P_{.5}(t, \gamma') \equiv \sqrt{2}\phi(\frac{-\gamma'}{\sigma\sqrt{2}} - \frac{t}{\sigma\sqrt{2}})\Phi(\frac{-\gamma'}{\sigma\sqrt{2}} + \frac{t}{\sigma\sqrt{2}})$$

In order to evaluate the integral we will calculate:

$$-\int_{-\infty}^{0} tP_{.5}(t, \gamma') + tP_{.5}(t, -\gamma')dt = \int_{-\infty}^{0} -tP_{.5}(t, \gamma') + \int_{-\infty}^{0} -tP_{.5}(t, -\gamma')$$

Let's calculate:

$$-tP_{.5}(t, \gamma') = -\sqrt{2}t\phi(\frac{-\gamma'}{\sigma\sqrt{2}} - \frac{t}{\sigma\sqrt{2}})\Phi(\frac{-\gamma'}{\sigma\sqrt{2}} + \frac{t}{\sigma\sqrt{2}}) = 2\sigma \cdot \left(\frac{\frac{-\gamma'}{\sigma} - \frac{t}{\sigma}}{\sqrt{2}}\right)\phi(\frac{-\gamma'}{\sigma\sqrt{2}} - \frac{t}{\sigma\sqrt{2}})\Phi(\frac{-\gamma'}{\sigma\sqrt{2}} +$$

$$\frac{t}{\sigma\sqrt{2}}) + \sqrt{2}\gamma'\phi(\frac{-\gamma'}{\sigma\sqrt{2}} - \frac{t}{\sigma\sqrt{2}})\Phi(\frac{-\gamma'}{\sigma\sqrt{2}} + \frac{t}{\sigma\sqrt{2}}) \equiv 2\sigma P_{.5,1}(t, \gamma') + \sqrt{2}\gamma' P_{.5,2}(t, \gamma')$$

We'll calculate each expression separately. In order to calculate the first part, we'll make the following change of parameters:

$$u_t^1 = \frac{\frac{-\gamma'}{\sigma} - \frac{t}{\sigma}}{\sqrt{2}}, u_t^2 = \frac{\frac{\gamma'}{\sigma} - \frac{t}{\sigma}}{\sqrt{2}}$$
$$\alpha_2 = \alpha_1 = -1$$
$$\beta_1 = -\beta_2 = \frac{-2\gamma'}{\sigma\sqrt{2}}$$

Then we can integrate it with Owen (1980) Table 1:10,011.1.

$$\int P_{.5,1}(\gamma', t)dt = \int \left(\frac{\frac{-\gamma'}{\sigma} - \frac{t}{\sigma}}{\sqrt{2}}\right)\phi(\frac{\frac{-\gamma'}{\sigma} - \frac{t}{\sigma}}{\sqrt{2}})\Phi(\sqrt{2}\frac{t}{\sigma} + \frac{\frac{-\gamma'}{\sigma} - \frac{t}{\sigma}}{\sqrt{2}})dt = -\sigma\sqrt{2}\int u_t^1\phi(u_t^1)\Phi(\beta_1 +$$

$$\alpha_1 u_t^1)du = -\sigma\sqrt{2}\left(\frac{\alpha_1}{s}\phi(\frac{\beta_1}{s})\Phi(u_t^1 s + \frac{\alpha_1\beta_1}{s}) - \phi(u_t^1)\Phi(\beta_1 + \alpha_1 u_t^1)\right), s = \sqrt{1 + \alpha_1^2} = \sqrt{2}$$

For the same reasons as before, we can't exactly calculate the second part, so let's approximate it:

$$\int_{-\infty}^{l} P_{.5,2}(\gamma', t)dt \equiv \int_{-\infty}^{l} \phi(\frac{\frac{-\gamma'}{\sigma} - \frac{t}{\sigma}}{\sqrt{2}})\Phi(\sqrt{2}\frac{t}{\sigma} + \frac{\frac{-\gamma'}{\sigma} - \frac{t}{\sigma}}{\sqrt{2}})dt \approx$$

$$\int_{-\infty}^{l} \phi(\frac{\frac{-\gamma'}{\sigma} - \frac{t}{\sigma}}{\sqrt{2}})\Phi(\sqrt{2}\frac{l}{\sigma} + \frac{\frac{-\gamma'}{\sigma} - \frac{l}{\sigma}}{\sqrt{2}})dt = \Phi(\sqrt{2}\frac{l}{\sigma} + \frac{\frac{-\gamma'}{\sigma} - \frac{l}{\sigma}}{\sqrt{2}}) \cdot \int_{-\infty}^{l} \phi(\frac{\frac{-\gamma'}{\sigma} - \frac{t}{\sigma}}{\sqrt{2}}) =$$

$$\Phi(\sqrt{2}\frac{l}{\sigma} + \frac{\frac{-\gamma'}{\sigma} - \frac{l}{\sigma}}{\sqrt{2}}) \cdot \frac{-1}{\sigma\sqrt{2}}\left(\Phi(\frac{\frac{-\gamma'}{\sigma} - \frac{t}{\sigma}}{\sqrt{2}})\Big|_{-\infty}^{l}\right) = \Phi(\sqrt{2}\frac{l}{\sigma} + \frac{\frac{-\gamma'}{\sigma} - \frac{l}{\sigma}}{\sqrt{2}}) \cdot -\sigma\sqrt{2}\left(\Phi(\frac{\frac{-\gamma'}{\sigma} - \frac{l}{\sigma}}{\sqrt{2}}) - 1\right)$$

Combining these two expressions, and the expressions from $P_{.5,1}(t, -\gamma')$ and $P_{.5,2}(t, -\gamma')$, we get:

$$\int_{-\infty}^{l} L(t)P(z = t) \approx \int_{-\infty}^{l} -tp(t) \approx \int_{-\infty}^{l} -t(p_{.5}(\gamma', t) + p_{.5}(-\gamma', t)) =$$
$$\int_{-\infty}^{l} 2\sigma p_{.5,1}(\gamma', t) + \sqrt{2}\gamma' p_{.5,2}(\gamma', t) + 2\sigma p_{.5,1}(-\gamma', t) + -\sqrt{2}\gamma' p_{.5,2}(-\gamma', t) =$$
$$2\sigma \int_{-\infty}^{l} p_{.5,1}(\gamma', t) + \sqrt{2}\gamma' \int_{-\infty}^{l} p_{.5,2}(\gamma', t) + 2\sigma \int_{-\infty}^{l} p_{.5,1}(-\gamma', t) + -\sqrt{2}\gamma'$$

**Applying the integral boundaries**

Having calculated the general form for every upper limit $l$, let's calculate the MLP loss.
For the MLP loss, the boundaries are $0, -\infty$ . The second expressions:

$$\sqrt{2}\gamma' \int_{-\infty}^{0} P_{.5,2}(\gamma', t) = -2\sigma\gamma'\Phi(-\tfrac{\gamma'}{\sigma\sqrt{2}})(\Phi(-\tfrac{\gamma'}{\sigma\sqrt{2}}) - 1)$$
$$-\sqrt{2}\gamma' \int_{-\infty}^{0} P_{.5,2}(-\gamma', t) = -2\sigma\gamma'\Phi(\tfrac{\gamma'}{\sigma\sqrt{2}})(\Phi(\tfrac{\gamma'}{\sigma\sqrt{2}}) - 1)$$

Calculating the first expression, for $P_{.5,1}(\gamma', t)$ we have:

$$2\sigma \int_{-\infty}^{l} P_{.5,1}(\gamma', t) = -2\sqrt{2}\sigma^2 \left( \tfrac{\alpha_1}{s}\phi(\tfrac{\beta_1}{s})\Phi(u_t^1 s + \tfrac{\alpha_1\beta_1}{s}) - \phi(u_t^1)\Phi(\beta_1 + \alpha_1 u_t^1) \right) \Big|_{-\infty}^{0}, s =$$

$$\sqrt{1 + \alpha_1^2}$$

$$-2\sqrt{2}\sigma^2 \left( \tfrac{\alpha_1}{s}\phi(\tfrac{\beta_1}{s})\Phi(u_t^1 s + \tfrac{\alpha_1\beta_1}{s}) - \phi(u_t^1)\Phi(\beta_1 + \alpha_1 u_t^1) \right) \Big|_{-\infty}^{0} = -2\sqrt{2}\sigma^2 \left( \tfrac{\alpha_1}{s}\phi(\tfrac{\beta_1}{s})\Phi(\tfrac{-\gamma'}{\sigma\sqrt{2}}t + \right.$$

$$\tfrac{\alpha_1\beta_1}{s}) - \phi(\tfrac{-\gamma'}{\sigma\sqrt{2}})\Phi(\beta_1 + \alpha_1\tfrac{-\gamma'}{\sigma\sqrt{2}}) - \left( \tfrac{\alpha_1}{s}\phi(\tfrac{\beta_1}{s})\Phi(\infty s + \tfrac{\alpha_1\beta_1}{s}) - \phi(\infty)\Phi(\beta_1 + \alpha_1\infty) \right) \Big) =$$

$$-2\sqrt{2}\sigma^2 \left( \tfrac{\alpha_1}{s}\phi(\tfrac{\beta_1}{s})\Phi(\tfrac{-\gamma'}{\sigma\sqrt{2}}t + \tfrac{\alpha_1\beta_1}{s}) - \phi(\tfrac{-\gamma'}{\sigma\sqrt{2}})\Phi(\beta_1 + \alpha_1\tfrac{-\gamma'}{\sigma\sqrt{2}}) - \tfrac{\alpha_1}{s}\phi(\tfrac{\beta_1}{s}) \right)$$

and similarly for $P_{.5,1}(-\gamma', t)$:

$$2\sigma \int_{-\infty}^{l} P_{.5,1}(-\gamma', t) = -2\sqrt{2}\sigma^2 \left( \tfrac{\alpha_2}{s}\phi(\tfrac{\beta_2}{s})\Phi(u_t^2 s + \tfrac{\alpha_2\beta_2}{s}) - \phi(u_t^2)\Phi(\beta_2 + \alpha_2 u_t^2) \right) \Big|_{-\infty}^{0}, s =$$

$$\sqrt{1 + \alpha_2^2}$$

$$-2\sqrt{2}\sigma^2 \left( \tfrac{\alpha_2}{s}\phi(\tfrac{\beta_2}{s})\Phi(u_t^2 s + \tfrac{\alpha_2\beta_2}{s}) - \phi(u_t^2)\Phi(\beta_2 + \alpha_2 u_t^2) \right) \Big|_{-\infty}^{0} = -2\sqrt{2}\sigma^2 \left( \tfrac{\alpha_2}{s}\phi(\tfrac{\beta_2}{s})\Phi(\tfrac{\gamma'}{\sigma\sqrt{2}}t + \right.$$

$$\tfrac{\alpha_2\beta_2}{s}) - \phi(\tfrac{\gamma'}{\sigma\sqrt{2}})\Phi(\beta_2 + \alpha_2\tfrac{\gamma'}{\sigma\sqrt{2}}) - \left( \tfrac{\alpha_2}{s}\phi(\tfrac{\beta_2}{s})\Phi(\infty s + \tfrac{\alpha_2\beta_2}{s}) - \phi(\infty)\Phi(\beta_2 + \alpha_2\infty) \right) \Big) =$$

$$-2\sqrt{2}\sigma^2 \left( \tfrac{\alpha_2}{s}\phi(\tfrac{\beta_2}{s})\Phi(\tfrac{\gamma'}{\sigma\sqrt{2}}t + \tfrac{\alpha_2\beta_2}{s}) - \phi(\tfrac{\gamma'}{\sigma\sqrt{2}})\Phi(\beta_2 + \alpha_2\tfrac{\gamma'}{\sigma\sqrt{2}}) - \tfrac{\alpha_2}{s}\phi(\tfrac{\beta_2}{s}) \right)$$

Adding all of these four expressions, notice that $P_{.5,2}(t, \gamma')$ and $P_{.5,2}(t, -\gamma')$ sum up to 0. And $P_{.5,1}(t, \gamma')$ and $P_{.5,1}(t, -\gamma')$ sum up to:

$$2\sqrt{2}\sigma^2 \left( \tfrac{\alpha_1}{t}\phi(\tfrac{\beta_1}{t}) + \phi(\tfrac{\gamma'}{\sqrt{2}\sigma}) \right) = 2\sqrt{2}\sigma^2\phi(\tfrac{\gamma'}{\sqrt{2}\sigma}) - 2\sigma^2\phi(\tfrac{\gamma'}{\sigma})$$

$\square$

**Theorem** (Restatement of part two of Theorem 1). *The expected SHIKI loss is:*

$$\mathbb{E}_X(L_{SHIKI}) \approx \left(2\Phi(\frac{\frac{\gamma'}{\sigma}-\frac{\tau}{\sigma}}{\sqrt{2}})+2\Phi(\frac{-\frac{\gamma'}{\sigma}-\frac{\tau}{\sigma}}{\sqrt{2}})-2\Phi(\frac{\tau}{\sigma})+2(\frac{1}{2}-\Phi(-\frac{\tau}{\sigma}))(\Phi(\frac{\gamma'}{\sigma}+\frac{\tau}{\sigma})+\Phi(-\frac{\gamma'}{\sigma}+\frac{\tau}{\sigma}))\right)\cdot$$

$$exp\left(-\frac{p-q}{p+q}\frac{2\gamma'^2}{\sigma}(2\frac{\Phi(\frac{\gamma'}{\sigma\sqrt{2}}-\frac{\tau}{\sigma\sqrt{2}})^2+\Phi(-\frac{\gamma'}{\sigma\sqrt{2}}-\frac{\tau}{\sigma\sqrt{2}})^2}{2\Phi(\frac{\frac{\gamma'}{\sigma}-\frac{\tau}{\sigma}}{\sqrt{2}})+2\Phi(\frac{-\frac{\gamma'}{\sigma}-\frac{\tau}{\sigma}}{\sqrt{2}})-2\Phi(\frac{\tau}{\sigma})+2(\frac{1}{2}-\Phi(-\frac{\tau}{\sigma}))(\Phi(\frac{\gamma'}{\sigma}+\frac{\tau}{\sigma})+\Phi(-\frac{\gamma'}{\sigma}+\frac{\tau}{\sigma}))}-1)^2\right)+$$

$$\left(2\sigma^2\phi(\frac{\gamma'}{\sigma})(2(1-\Phi(\frac{\tau}{\sigma}))-1)+2\sqrt{2}\sigma^2\left(\phi(\frac{\gamma'}{\sqrt{2}\sigma})-\phi(\frac{-\gamma'+\tau}{\sqrt{2}\sigma})\Phi(\frac{-\gamma'-\tau}{\sqrt{2}\sigma})-\phi(\frac{\gamma'+\tau}{\sqrt{2}\sigma})\Phi(\frac{\gamma'-\tau}{\sqrt{2}\sigma})\right)\right)$$

*Proof.*

$$\mathbb{E}_X(L_{SHIKI}) = \mathbb{E}_X(L_{SHIKI}|x\sim|\mu|) = P(confident|x\sim|\mu|)\cdot\mathbb{E}_X(L_{GNN}|confident,x\sim|\mu|)+P(not\,confident|x\sim|\mu|)\cdot\mathbb{E}_X(L_{MLP}|not\,confident,x\sim|\mu|)=$$
$$\mathbb{E}_X(L_{GNN}||x|>\tau,x\sim|\mu|)P(|x|>\tau|x\sim|\mu|)+\mathbb{E}_X(L_{MLP}||x|<\tau,x\sim|\mu|)P(|x|<\tau|x\sim|\mu|)<$$
$$\mathbb{E}_X(L_{GNN})P(|x|>\tau,x\sim|\mu|)+\mathbb{E}_X(L_{MLP}||x|<\tau,x\sim|\mu|)P(|x|<\tau,x\sim|\mu|)$$

We will separate the calculation of our loss for the GNN part and for the MLP part.
First the MLP part.
Say we want to calculate $\mathbb{E}_X(L_{MLP}||x|<\tau,x\sim|\mu|)$:

$$\mathbb{E}_X(L_{MLP}||x|<\tau,x\sim|\mu|) = \int_{-\tau}^{\tau}P(x||x|<\tau,x\sim|\mu|)L(x|x\sim|\mu|)dx =$$
$$\int_{-\tau}^{\tau}\frac{P(x|,x\sim|\mu|)}{P(|x|<\tau,x\sim|\mu|)}L(x)dx = \frac{1}{P(|x|<\tau|x\sim|\mu|)}\cdot\int_{-\tau}^{\tau}P(x|x\sim|\mu|)L(x|x\sim|\mu|)dx$$
$$\downarrow$$
$$\mathbb{E}_X(L_{MLP}||x|<\tau,x\sim|\mu|)P(|x|<\tau|x\sim|\mu|) = \int_{-\tau}^{\tau}P(x|x\sim|\mu|)L(x|x\sim|\mu|)dx$$

Similarly to the case with the regular MLP loss, we we'll ignore the case when we are right. The integral boundaries will become $0$ and $-\tau$.

**Applying the integral boundaries for the MLP part**

We'll calculate $P_{0.5,1}(t,\gamma')$ and $P_{0.5,1}(t,-\gamma')$ with the boundaries $-\tau$ and $-\infty$ in the same way as before.

$$-2\sqrt{2}\sigma^2\left(\frac{\alpha_1}{s}\phi(\frac{\beta_1}{s})\Phi(u_x^1 s+\frac{\alpha_1\beta_1}{s})-\phi(u_x^1)\Phi(\beta_1+\alpha_1 u_x^1)\right)\Big|_{-\infty}^{-\tau} =$$

$$-2\sqrt{2}\sigma^2\left(\frac{\alpha_1}{s}\phi(\frac{\beta_1}{s})\Phi(\frac{-\gamma'+\tau}{\sigma\sqrt{2}}s+\frac{\alpha_1\beta_1}{s})-\phi(\frac{-\gamma'+\tau}{\sigma\sqrt{2}})\Phi(\beta_1+\alpha_1\frac{-\gamma'+\tau}{\sigma\sqrt{2}})-\right.$$

$$\left.\left(\Phi(\infty s+\frac{\alpha_1\beta_1}{s})-\phi(\infty)\Phi(\beta_1+\alpha_1\infty)\right)\right) =$$

$$-2\sqrt{2}\sigma^2\left(\frac{\alpha_1}{s}\phi(\frac{\beta_1}{s})\Phi(\frac{-\gamma'+\tau}{\sigma\sqrt{2}}s+\frac{\alpha_1\beta_1}{s})-\phi(\frac{-\gamma'+\tau}{\sigma\sqrt{2}})\Phi(\beta_1+\alpha_1\frac{-\gamma'+\tau}{\sigma\sqrt{2}})-\frac{\alpha_1}{s}\phi(\frac{\beta_1}{s})\right)$$

$$-2\sqrt{2}\sigma^2\left(\frac{\alpha_2}{s}\phi(\frac{\beta_2}{s})\Phi(u_x^2 s+\frac{\alpha_2\beta_2}{s})-\phi(u_x^2)\Phi(\beta_2+\alpha_2 u_x^2)\right)\Big|_{-\infty}^{-\tau} =$$

$$-2\sqrt{2}\sigma^2\left(\frac{\alpha_2}{s}\phi(\frac{\beta_2}{s})\Phi(\frac{\gamma'+\tau}{\sigma\sqrt{2}}s+\frac{\alpha_2\beta_2}{s})-\phi(\frac{\gamma'+\tau}{\sigma\sqrt{2}})\Phi(\beta_2+\alpha_2\frac{\gamma'+\tau}{\sigma\sqrt{2}})-\right.$$

$$\left.\left(\Phi(\infty s+\frac{\alpha_2\beta_2}{s})-\phi(\infty)\Phi(\beta_2+\alpha_2\infty)\right)\right) =$$

$$-2\sqrt{2}\sigma^2\left(\frac{\alpha_2}{s}\phi(\frac{\beta_2}{s})\Phi(\frac{\gamma'+\tau}{\sigma\sqrt{2}}s+\frac{\alpha_2\beta_2}{s})-\phi(\frac{\gamma'+\tau}{\sigma\sqrt{2}})\Phi(\beta_2+\alpha_2\frac{\gamma'+\tau}{\sigma\sqrt{2}})-\frac{\alpha_2}{s}\phi(\frac{\beta_2}{s})\right)$$

and summing these two expressions we get:

$$-2\sqrt{2}\sigma^2\left[\frac{\alpha_1}{s}\phi(\frac{\beta_1}{s})\left(\Phi(\frac{-\gamma'+\tau}{\sqrt{2}\sigma}s+\frac{\alpha_1\beta_1}{s})+\Phi(\frac{\gamma'+\tau}{\sqrt{2}\sigma}s+\frac{\alpha_2\beta_2}{s})\right)-\phi(\frac{-\gamma'+\tau}{\sqrt{2}\sigma})\Phi(\beta_1+\alpha_1\frac{-\gamma'+\tau}{\sqrt{2}\sigma})-\right.$$

$$\left.\phi(\frac{\gamma'+\tau}{\sqrt{2}\sigma})\Phi(\beta_2+\alpha_2\frac{\gamma'+\tau}{\sqrt{2}\sigma})-2\frac{\alpha_1}{s}\phi(\frac{\beta_1}{s})\right]=$$

$$2\sqrt{2}\sigma^2\left[\frac{\alpha_1}{s}\phi(\frac{\beta_1}{s})\left(2-\left(\Phi(\frac{-\gamma'+\tau}{\sqrt{2}\sigma}s+\frac{\alpha_1\beta_1}{s})+\Phi(\frac{\gamma'+\tau}{\sqrt{2}\sigma}s+\frac{\alpha_2\beta_2}{s})\right)\right)+\phi(\frac{-\gamma'+\tau}{\sqrt{2}\sigma})\Phi(\beta_1+\right.$$

$$\left.\alpha_1\frac{-\gamma'+\tau}{\sqrt{2}\sigma})+\phi(\frac{\gamma'+\tau}{\sqrt{2}\sigma})\Phi(\beta_2+\alpha_2\frac{\gamma'+\tau}{\sqrt{2}\sigma})\right]=$$

$$2\sqrt{2}\sigma^2\left[\frac{-1}{\sqrt{2}}\phi(\frac{\gamma'}{\sigma})\left(2-\left(\Phi(\frac{-\gamma'+\tau}{\sigma}+\frac{\gamma'}{\sigma})+\Phi(\frac{\gamma'+\tau}{\sigma}-\frac{\gamma'}{\sigma})\right)\right)+\phi(\frac{-\gamma'+\tau}{\sqrt{2}\sigma})\Phi(\frac{-2\gamma'}{\sigma\sqrt{2}}-\frac{-\gamma'+\tau}{\sqrt{2}\sigma})+\right.$$

$$\left.\phi(\frac{\gamma'+\tau}{\sqrt{2}\sigma})\Phi(\frac{2\gamma'}{\sigma\sqrt{2}}-\frac{\gamma'+\tau}{\sqrt{2}\sigma})\right]=$$

$$2\sqrt{2}\sigma^2\left[-\sqrt{2}\phi(\frac{\gamma'}{\sigma})\left(1-\Phi(\frac{\tau}{\sigma})\right)+\phi(\frac{-\gamma'+\tau}{\sqrt{2}\sigma})\Phi(\frac{-\gamma'-\tau}{\sqrt{2}\sigma})+\phi(\frac{\gamma'+\tau}{\sqrt{2}\sigma})\Phi(\frac{\gamma'-\tau}{\sqrt{2}\sigma})\right]$$

And calculating these as a part of the MLP loss:

$$-2\sqrt{2}\sigma^2\left(\frac{\alpha_1}{s}\phi(\frac{\beta_1}{s})\Phi(u_x^1 s+\frac{\alpha_1\beta_1}{s})-\phi(u_x^1)\Phi(\beta_1+\alpha_1 u_x^1)\right)\bigg|_{-\tau}^{0}=$$

$$-2\sqrt{2}\sigma^2\left(\frac{\alpha_1}{s}\phi(\frac{\beta_1}{s})\Phi(u_x^1 s+\frac{\alpha_1\beta_1}{s})-\phi(u_x^1)\Phi(\beta_1+\alpha_1 u_x^1)\right)\bigg|_{-\infty}^{0}-$$

$$-2\sqrt{2}\sigma^2\left(\frac{\alpha_1}{s}\phi(\frac{\beta_1}{s})\Phi(u_x^1 s+\frac{\alpha_1\beta_1}{s})-\phi(u_x^1)\Phi(\beta_1+\alpha_1 u_x^1)\right)\bigg|_{-\infty}^{-\tau}=2\sqrt{2}\sigma^2\phi(\frac{\gamma'}{\sqrt{2}\sigma})-2\sigma^2\phi(\frac{\gamma'}{\sigma})-$$

$$\left(2\sqrt{2}\sigma^2\left[-\sqrt{2}\phi(\frac{\gamma'}{\sigma})\left(1-\Phi(\frac{\tau}{\sigma})\right)+\phi(\frac{-\gamma'+\tau}{\sqrt{2}\sigma})\Phi(\frac{-\gamma'-\tau}{\sqrt{2}\sigma})+\phi(\frac{\gamma'+\tau}{\sqrt{2}\sigma})\Phi(\frac{\gamma'-\tau}{\sqrt{2}\sigma})\right]\right)=$$

$$2\sigma^2\phi(\frac{\gamma'}{\sigma})(2(1-\Phi(\frac{\tau}{\sigma}))-1)+2\sqrt{2}\sigma^2\left(\phi(\frac{\gamma'}{\sqrt{2}\sigma})-\phi(\frac{-\gamma'+\tau}{\sqrt{2}\sigma})\Phi(\frac{-\gamma'-\tau}{\sqrt{2}\sigma})-\phi(\frac{\gamma'+\tau}{\sqrt{2}\sigma})\Phi(\frac{\gamma'-\tau}{\sqrt{2}\sigma})\right)$$

Now we'll calculate $P_{0.5,2}(t,\gamma')$ and $P_{0.5,2}(t,-\gamma')$ with the boundaries $-\tau$ and $-\infty$ in the same way as before:

$$\gamma'\int_{-\infty}^{-\tau}p_{.5,2}(\gamma',x)dx+-\gamma'\int_{-\infty}^{-\tau}p_{.5,2}(-\gamma',x)dx=$$

$$-2\sigma\gamma'\Phi(-\sqrt{2}\frac{\tau}{\sigma}+\frac{\frac{-\gamma'}{\sigma}+\frac{\tau}{\sigma}}{\sqrt{2}})\left(\Phi(\frac{\frac{-\gamma'}{\sigma}+\frac{\tau}{\sigma}}{\sqrt{2}})-1\right)+2\sigma\gamma'\Phi(-\sqrt{2}\frac{\tau}{\sigma}+\frac{\frac{\gamma'}{\sigma}+\frac{\tau}{\sigma}}{\sqrt{2}})\left(\Phi(\frac{\frac{\gamma'}{\sigma}+\frac{\tau}{\sigma}}{\sqrt{2}})-1\right)=$$

$$2\sigma\gamma'\Phi(\frac{-\gamma'}{\sqrt{2}\sigma}+\frac{-\tau}{\sqrt{2}})\left(1-\Phi(\frac{\frac{-\gamma'}{\sigma}+\frac{\tau}{\sigma}}{\sqrt{2}})\right)-2\sigma\gamma'\Phi(\frac{\gamma'}{\sqrt{2}\sigma}+\frac{-\tau}{\sqrt{2}})\left(1-\Phi(\frac{\frac{\gamma'}{\sigma}+\frac{\tau}{\sigma}}{\sqrt{2}})\right)=$$

$$2\sigma\gamma'\left(\Phi(\frac{-\gamma'}{\sqrt{2}\sigma}+\frac{-\tau}{\sqrt{2}})\Phi(\frac{\gamma'}{\sqrt{2}\sigma}+\frac{-\tau}{\sqrt{2}})-\Phi(\frac{\gamma'}{\sqrt{2}\sigma}+\frac{-\tau}{\sqrt{2}})\Phi(\frac{-\gamma'}{\sqrt{2}\sigma}+\frac{-\tau}{\sqrt{2}})\right)=0$$

We saw earlier that with the integral boundaries of $0$ and $-\infty$, we also get 0. So when calculating with the boundaries of $0$ and $-\tau$ as it would be in the loss, we will still get 0. And so this final loss is:

$$2\sigma^2\phi(\frac{\gamma'}{\sigma})(2(1-\Phi(\frac{\tau}{\sigma}))-1)+2\sqrt{2}\sigma^2\left(\phi(\frac{\gamma'}{\sqrt{2}\sigma})-\phi(\frac{-\gamma'+\tau}{\sqrt{2}\sigma})\Phi(\frac{-\gamma'-\tau}{\sqrt{2}\sigma})-\phi(\frac{\gamma'+\tau}{\sqrt{2}\sigma})\Phi(\frac{\gamma'-\tau}{\sqrt{2}\sigma})\right)$$

**Calculating the GNN part**
Now let's calculate the GNN part. Using lemma 5, the GNN loss is:

$$e^{-\frac{2\gamma'^2}{\sigma}\frac{rp-rq}{rp+rq}}=e^{-\frac{p-q}{p+q}\frac{2\gamma'^2}{\sigma}(2\alpha_\tau-1)^2}$$

This should be multiplied by $\beta_\tau$. So we finally we get:

$$\mathbb{E}(L_{SHIKI}) = \beta_\tau \cdot exp\left(-\frac{p-q}{p+q}\frac{2\gamma'^2}{\sigma}(2\alpha_\tau - 1)^2\right) + \left(2\sigma^2\phi(\frac{\gamma'}{\sigma})(2(1-\Phi(\frac{\tau}{\sigma})) - 1) +\right.$$

$$\left. 2\sqrt{2}\sigma^2\left(\phi(\frac{\gamma'}{\sqrt{2}\sigma}) - \phi(\frac{-\gamma'+\tau}{\sqrt{2}\sigma})\Phi(\frac{-\gamma'-\tau}{\sqrt{2}\sigma}) - \phi(\frac{\gamma'+\tau}{\sqrt{2}\sigma})\Phi(\frac{\gamma'-\tau}{\sqrt{2}\sigma})\right)\right)$$

And the full expression as a function of $\tau, \gamma', \sigma, p, q$:

$$L_{GNN} = \left(2\Phi(\frac{\frac{\gamma'}{\sigma}-\frac{\tau}{\sigma}}{\sqrt{2}}) + 2\Phi(\frac{-\frac{\gamma'}{\sigma}-\frac{\tau}{\sigma}}{\sqrt{2}}) - 2\Phi(\frac{\tau}{\sigma}) + 2(\frac{1}{2}-\Phi(-\frac{\tau}{\sigma}))(\Phi(\frac{\gamma'}{\sigma}+\frac{\tau}{\sigma}) + \Phi(-\frac{\gamma'}{\sigma}+\frac{\tau}{\sigma}))\right)\cdot$$

$$exp\left(-\frac{p-q}{p+q}\frac{2\gamma'^2}{\sigma}(2\frac{\Phi(\frac{\gamma'}{\sigma\sqrt{2}}-\frac{\tau}{\sigma\sqrt{2}})^2 + \Phi(-\frac{\gamma'}{\sigma\sqrt{2}}-\frac{\tau}{\sigma\sqrt{2}})^2}{2\Phi(\frac{\frac{\gamma'}{\sigma}-\frac{\tau}{\sigma}}{\sqrt{2}}) + 2\Phi(\frac{-\frac{\gamma'}{\sigma}-\frac{\tau}{\sigma}}{\sqrt{2}}) - 2\Phi(\frac{\tau}{\sigma}) + 2(\frac{1}{2}-\Phi(-\frac{\tau}{\sigma}))(\Phi(\frac{\gamma'}{\sigma}+\frac{\tau}{\sigma})+\Phi(-\frac{\gamma'}{\sigma}+\frac{\tau}{\sigma}))} - 1)^2\right) +$$

$$\left(2\sigma^2\phi(\frac{\gamma'}{\sigma})(2(1-\Phi(\frac{\tau}{\sigma})) - 1) + 2\sqrt{2}\sigma^2\left(\phi(\frac{\gamma'}{\sqrt{2}\sigma}) - \phi(\frac{-\gamma'+\tau}{\sqrt{2}\sigma})\Phi(\frac{-\gamma'-\tau}{\sqrt{2}\sigma}) - \phi(\frac{\gamma'+\tau}{\sqrt{2}\sigma})\Phi(\frac{\gamma'-\tau}{\sqrt{2}\sigma})\right)\right)$$

$$\square$$

To better understand the behavior of these expression we refer the reader to Section 4.5 of the main part.

# B    EFFECTIVENESS PROPERTIES OF $XOR$-$GMM$ VARIATIONS

In this Section we show the effectiveness of GNN against MLP where the data is generated from different $XOR$-$GMM$ variations, which are meant to model the real world data described in Section 2. Let $\Phi(\cdot)$ denote the cumulative distribution function of a standard Gaussian, and $\Phi_c(\cdot) = 1 - \Phi(\cdot)$. In what follows, the full proofs are provided in Appendices C and D.

## B.1    SHIFTED CENTERS CASE

We denote the variation described in Section 2 for the Amazon reviews dataset as $XOR$-$GMM$-$SC$. Similarly to the $XOR$-$CSBM$ model in Section 2, we can define edges over the $XOR$-$GMM$-$SC$ models, and denote it $(A, X) \sim XOR$-$CSBM$-$SC(n, d, \mu, \nu, \sigma^2, p, q)$.

### B.1.1    BASELINE

The following theorem provides a complete characterization of the decision boundary for the XOR-GMM-SC data model. This characterization relies on two key factors: the separation between the means in the mixture model and the dataset size, represented by $n$. The theorem is divided into two components. The first component examines the constraints of a perfect classifier regarding its accuracy. And the third component identifies the area in which the optimal MLP achieves perfect classification of the data.

**Theorem 2.** *Let $X \in \mathbb{R}^{n\times d} \sim XOR - GMM - SC(n, d, \mu, \nu, \sigma^2)$. Then we have the following:*

1. *Assume that $\|\mu - \nu\|_2 = K\sigma$ and let $h(x) : \mathbb{R}^d \to \{0, 1\}$ be any binary classifier. Then for any $K > 0$ and any $\epsilon \in (0, 1)$, at least a fraction $\Phi_c(\frac{K}{2}) - O(n^{-\epsilon/2})$ of all data points are misclassified by $h$ with probability at least $1 - \exp(-2n^{1-\epsilon})$.*

2. *For any $\epsilon > 0$, if the distance between the means is $\|\mu - \nu\|_2 = \Omega(\sigma(\log n)^{\frac{1}{2}+\epsilon})$, then for any $c > 0$, with probability at least $1 - O(n^{-c})$, there exists a two-layer that perfectly classify the data, and obtain a cross-entropy loss given by*

$$\ell_\theta(X) = C \exp(-\frac{R}{\sqrt{2}}\|\mu - \nu\|_2(1 \pm \sqrt{c}/(\log n)^\epsilon)),$$

*where $C \in [\frac{1}{2}, 1]$ is an absolute constant.*

### B.1.2 GRAPH CONVOLUTION IMPROVEMENT

We now present the results that illustrate the impact of graph convolutions in multi-layer networks with the specified architecture. We quantify the improvement in the classification threshold based on the separation between the means of the node features.

**Theorem 3.** *Let $(A, X) \sim XOR - CSBM - SC(n, d, \mu, \nu, \sigma^2, p, q)$. Then there exists a two-layer network and a three-layer network with the following properties: If the intra-class and inter-class edge probabilities are $p, q = \Omega(\frac{\log^2 n}{n})$, and the distance between the means is $\|\mu - \nu\|_2 = \Omega(\frac{\sigma \log n}{\sqrt{n(p+q)}})$, then for any $c > 0$, with probability at least $1 - O(n^{-c})$, the networks equipped with a graph convolution in the second or the third layer perfectly classify the data, and obtain the following loss:*

$$\ell_\theta(A, X) = C' \exp\left(-R\|\|\mu - \nu\|\|_2 \left|\frac{p-q}{p+q}\right| \left(1 \pm \sqrt{\frac{c}{\log n}}\right)\right),$$

*where $C > 0$ and $C' \in [\frac{1}{2}, 1]$ are constants.*

### B.2 IMBALANCED CASE

In this section, we prove some basic results similar to Baranwal et al. (2022) on the effectiveness of a GNN against an MLP on an imbalanced synthetic model. We take the original model and add only class imbalance with no shifted centers. This is done to emphasize that while adding imbalance to the model, it retains the nice results from (Baranwal et al., 2022). We achieve the said imbalance by setting $\epsilon_i = Ber(w_1)$ instead of $\epsilon_i = Ber(\frac{1}{2})$ where $w_1 = \Omega(1)$. This is done in order to give class 1 a smaller chance to get picked ($w_1 < \frac{1}{2}$). We follow the same steps as in Baranwal et al. (2022) to achieve similar guarantees. We call this variation $XOR\text{-}GMM\text{-}I$, and similarly to the $XOR\text{-}CSBM$ model in Section 2, we can define edges over the $XOR\text{-}GMM\text{-}I$ models, and denote it $(A, X) \sim XOR\text{-}CSBM\text{-}I(n, d, \mu, \nu, \sigma^2, p, q)$.

To better understand how this model behaves, we show it in Figure 11

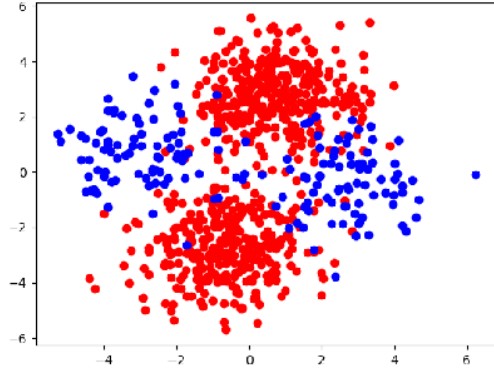

Figure 11: Visual illustration of the $XOR\text{-}GMM\text{-}I$, with distance of 4, and $\sigma = 1.3$.

### B.3 BASELINE

The following theorem provides a complete description of the classification boundary for the XOR-GMM-I data model. This description is based on the distance between the means and the number of data points, $n$. The theorem consists of three parts. The first part explores the limitations of a perfect classifier in terms of its accuracy. The second part explores its limitations in terms of precision/recall/f-score. And finally, the third and last part establishes the region where the best MLP perfectly classifies the data.

**Theorem 4.** *Let $X \in \mathbb{R}^{n \times d} \sim XOR\text{-}GMM\text{-}I(n, d, \mu, \nu, \sigma^2)$. Assume that $\|\mu - \nu\|_2 = K\sigma$, then we have the following:*

1. *Let $h(x) : \mathbb{R}^d \rightarrow \{0,1\}$ be any binary classifier. Then, for $K > 0, K_{|\mu|,i} = \frac{K}{\sqrt{2}} + \sigma^2 \ln(\frac{w_0}{w_1}), K_{|\nu|,i} = \frac{K}{\sqrt{2}} + \sigma^2 \ln(\frac{w_1}{w_0}) = \frac{K}{\sqrt{2}} - \sigma^2 \ln(\frac{w_0}{w_1})$ and any $\epsilon \in (0,1)$, at least a fraction of*

$$
w_0 \cdot \begin{cases} 1 - 2\Phi_c(\frac{K_{|\mu|,i}}{\sqrt{2}})^2 & if \ K_{|\mu|,i} \geq 0 \\ 4\Phi(\frac{K_{|\mu|,i}}{\sqrt{2}}) - 2\Phi(\frac{K_{|\mu|,i}}{\sqrt{2}})^2 + 4\Phi(K_{|\mu|,i})^2 - 4\Phi(K_{|\mu|,i}) & if \ K_{|\mu|,i} < 0 \end{cases}
$$

$$
+
$$

$$
w_1 \cdot \begin{cases} 1 - 2\Phi_c(\frac{K_{|\nu|,i}}{\sqrt{2}})^2 & if \ K_{|\nu|,i} \geq 0 \\ 4\Phi(\frac{K_{|\nu|,i}}{\sqrt{2}}) - 2\Phi(\frac{K_{|\nu|,i}}{\sqrt{2}})^2 + 4\Phi(K_{|\nu|,i})^2 - 4\Phi(K_{|\nu|,i}) & if \ K_{|\nu|,i} < 0 \end{cases}
$$

$$
- O(n^{-\epsilon/2})
$$

*of all data points are misclassified by $h$ with probability of at least $1 - \exp(-2n^{1-\epsilon})$.*

2. *Assume for simplicity's sake that $K_i > 0$. Then, we have:*

$$
accuracy = P(right\ classification) = w_0 \cdot \left( 1 - 2\Phi_c \left( \frac{K}{2} + \frac{\sigma^2 \ln(\frac{w_0}{w_1})}{\sqrt{2}} \right)^2 \right) +
$$

$$
w_1 \cdot \left( 1 - 2\Phi_c \left( \frac{K}{2} - \frac{\sigma^2 \ln(\frac{w_0}{w_1})}{\sqrt{2}} \right)^2 \right) \pm O(n^{-\epsilon/2})
$$

$$
precision = \frac{w_0 \cdot \left( 1 - 2\Phi_c \left( \frac{K}{2} + \frac{\sigma^2 \ln(\frac{w_0}{w_1})}{\sqrt{2}} \right)^2 \right) \pm O(n^{-\epsilon/2})}{w_0 \cdot \left( 1 - 2\Phi_c \left( \frac{K}{2} + \frac{\sigma^2 \ln(\frac{w_0}{w_1})}{\sqrt{2}} \right)^2 \right) + w_1 \cdot \left( 2\Phi_c \left( \frac{K}{2} - \frac{\sigma^2 \ln(\frac{w_0}{w_1})}{\sqrt{2}} \right)^2 \right) \pm O(n^{-\epsilon/2})}
$$

$$
recall = 1 - 2\Phi_c \left( \frac{K}{2} + \frac{\sigma^2 \ln(\frac{w_0}{w_1})}{\sqrt{2}} \right)^2 \pm O(n^{-\epsilon/2})
$$

$$
f\text{-}score =
$$

$$
\frac{2w_0 \cdot \left( 1 - 2\Phi_c \left( \frac{K}{2} + \frac{\sigma^2 \ln(\frac{w_0}{w_1})}{\sqrt{2}} \right)^2 \right)^2 \pm O(n^{-\epsilon/2})}{2w_0 \cdot \left( 1 - 2\Phi_c \left( \frac{K}{2} + \frac{\sigma^2 \ln(\frac{w_0}{w_1})}{\sqrt{2}} \right)^2 \right)^2 + w_1 \cdot \left( 1 - 2\Phi_c \left( \frac{K}{2} + \frac{\sigma^2 \ln(\frac{w_0}{w_1})}{\sqrt{2}} \right)^2 \right) \left( 2\Phi_c \left( \frac{K}{2} - \frac{\sigma^2 \ln(\frac{w_0}{w_1})}{\sqrt{2}} \right)^2 \right) \pm O(n^{-\epsilon/2})}
$$

3. *For any $\epsilon > 0$, if the distance between the means is*

$$
||\mu - \nu||_2 = \Omega(max(\sigma(\log n)^{\frac{1}{2}+\epsilon}, \sigma^2 |logit(w_0)|)
$$

*then for any $c > 0$, with probability of at least $1 - O(n^{-c})$, there exists a two-layer network that perfectly classifies the data, obtaining a cross-entropy loss given by*

$$
\ell_\theta(X) = C \exp(-\frac{R}{\sqrt{2}} ||\mu - \nu||_2 (1 \pm \sqrt{c}/(\log n)^\epsilon)),
$$

*where $C \in [\frac{1}{2}, 1]$ is an absolute constant and $R$ is the optimality constraint from.*

Aside from the basic theorems we prove, we also explicitly show the accuracy/precision/recall/f-score. For any other synthetic model, one can simply show the misclassification rate. However, in our case, we have an imbalance between the classes. In this case, the more informative metrics are the ones that take into account this imbalance, i.e precision/recall/f-score.

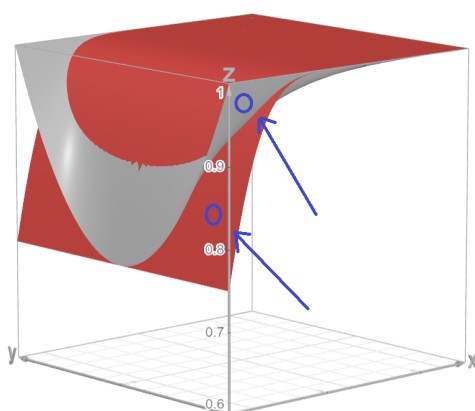

Figure 12: balanced vs imbalanced accuracy, red is the accuracy in the balanced, gray is the accuracy in the balanced. We choose two point, and emphasize that by simply looking at the accuracy, we can achieve far better accuracy than the balanced case when the distance between the means is quite small.

Next, we visually demonstrate the short-coming of looking merely at the accuracy. Let's plot the accuracy (z-axis) for this imbalanced case, and the original balanced case as a function of $\gamma$ (x-axis) and $w_0$ (y-axis). We set $\sigma = 1$.

As we can see, the more the data is unbalanced, the easier the task is, because we are more likely to fall in the bigger class, and just classify it as the bigger class is right most of the times. Instead of looking merely at the accuracy, it's more informative to look at the precision/recall/f-score. Let's plot the other metrics as a function of $\gamma$ (x-axis) and $w_1$ (y-axis), and $\sigma$ as a parameter:

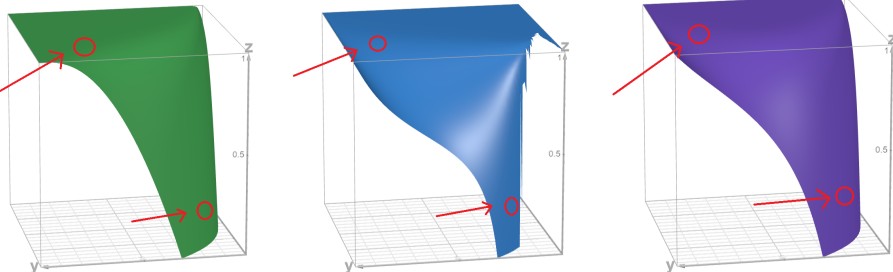

(a) plot of the recall as a function of $\gamma$ (x-axis), $w_1$ (y-axis), $\sigma$ is a parameter equals to 1. x-axis is between 0-4, y-axis is between 0-1, z-axis is between 0-1

(b) plot of the precision as a function of $\gamma$ (x-axis), $w_1$ (y-axis), $\sigma$ is a parameter equals to 1. x-axis is between 0-4, y-axis is between 0-1, z-axis is between 0-1

(c) plot of the f-score as a function of $\gamma$ (x-axis), $w_1$ (y-axis), $\sigma$ is a parameter equals to 1. x-axis is between 0-4, y-axis is between 0-1, z-axis is between 0-1

Figure 13: plots of the informative metrics. We choose two point, and emphasize that by looking at this informative metrics, we get the desired result. Where the data is unbalanced in our favor, we perform quite well even better than unbalanced case. And when the imbalance is against us, we perform very poorly.

Unlike the accuracy which grows bigger as the imbalance grows larger, for the f-score, as the size of class 1 decrease, the f-score decreases as well.

### B.4  GRAPH CONVOLUTION IMPROVEMENT

We now show the effects of graph convolutions in multi-layer networks with the architecture described in Section 3. We characterize the improvement in the classification threshold in terms of the distance between the means of the node features.

**Theorem 5.** *Let $(A, X) \sim XOR\text{-}CSBM\text{-}I(n, d, \mu, \nu, \sigma^2, p, q)$. If the intra-class and inter-class edge probabilities are $p, q = \Omega(\frac{\log^2 n}{n})$, the distance between the means is $\|\mu - \nu\|_2 = max(\Omega(\frac{\sigma \log n}{\sqrt{n(p+q)}}), \sigma^2|logit(w_0)|)$, and $sgn(w_0 p - w_1 q) = sgn(w_1 p - w_0 q)$, then for any $c > 0$, with probability at least $1 - O(n^{-c})$, the networks equipped with a graph convolution in the second layer perfectly classify the data, and obtain the following loss:*

$$\ell_\theta(A, X) \leq C' \exp\left(-R\|\|\mu - \nu\|\|_2 \frac{max(|w_0 p - w_1 q|, |w_1 p - w_0 q|)}{w_0 p + w_1 q}\left(1 \pm \sqrt{\frac{c}{\log n}}\right)\right),$$

*where $C > 0$ and $C' \in [\frac{1}{2}, 1]$ are constants.*

## C CALCULATIONS FOR SHIFTED CENTERS

Here, we prove some basic results similar to Baranwal et al. (2022), for the shifted centers model case described in Section 2 for the Amazon reviews dataset.

**Lemma 6.** *For some fixed $\mu, \nu \in \mathbb{R}^d$ and $\sigma^2 > 0$, the Bayes optimal classifier, $h^*(x) : \mathbb{R}^d \to \{0, 1\}$ for the shifted center data model is given by*

$$h^*(x) = \mathbb{1}(-\langle x, \mu \rangle < \langle x, \nu \rangle)$$

*Proof.* Note that $P(y = 1) = P(y = 0) = \frac{1}{2}$. Let $f(x)$ denote the density function of a continuous random vector $x$. Therefore, for any $b \in \{0, 1\}$,

$$P(y = 1|x) = \frac{f_{x|y}(x|y=1)P(y=1)}{\sum_{c\in\{0,1\}} P[y=c]f_{x|y}(x|y=c)} = \frac{1}{1 + \frac{P(y=0)f_{x|y}(x|y=0)}{P(y=1)f_{x|y}(x|y=1)}} = \frac{1}{1 + \frac{f_{x|y}(x|y=0)}{f_{x|y}(x|y=1)}}$$

$$\frac{f_{x|y}(x|y=0)}{f_{x|y}(x|y=1)} = \frac{e^{\frac{\langle x,\mu \rangle}{\sigma^2}} + e^{\frac{\langle x,\nu \rangle}{\sigma^2}}}{e^{\frac{-\langle x,\nu \rangle}{\sigma^2}} + e^{\frac{-\langle x,\mu \rangle}{\sigma^2}}}$$

For label 0, we require the probability to be less than $\frac{1}{2}$, thus, we need that expression to be less than 1,

$$\frac{e^{\frac{\langle x,\mu \rangle}{\sigma^2}} + e^{\frac{\langle x,\nu \rangle}{\sigma^2}}}{e^{\frac{-\langle x,\nu \rangle}{\sigma^2}} + e^{\frac{-\langle x,\mu \rangle}{\sigma^2}}} < 1$$

$$e^{\frac{\langle x,\mu \rangle}{\sigma^2}} + e^{\frac{\langle x,\nu \rangle}{\sigma^2}} < e^{\frac{-\langle x,\nu \rangle}{\sigma^2}} + e^{\frac{-\langle x,\mu \rangle}{\sigma^2}}$$

$$e^{\frac{\langle x,\mu \rangle}{\sigma^2}} - e^{\frac{-\langle x,\mu \rangle}{\sigma^2}} < e^{\frac{-\langle x,\nu \rangle}{\sigma^2}} - e^{\frac{\langle x,\nu \rangle}{\sigma^2}}$$

$$sinh(\frac{\langle x,\mu \rangle}{\sigma^2}) < sinh(-\frac{\langle x,\nu \rangle}{\sigma^2})$$

$$\frac{\langle x,\mu \rangle}{\sigma^2} < -\frac{\langle x,\nu \rangle}{\sigma^2}$$

$$\langle x, \mu \rangle < -\langle x, \nu \rangle$$

$$\langle x, \nu \rangle < -\langle x, \mu \rangle$$

$\square$

And for label 1 we have:

$$\langle x, \mu \rangle > -\langle x, \nu \rangle$$
$$\langle x, \nu \rangle > -\langle x, \mu \rangle$$

**Fact 1.** *For any $x, y \in \mathbb{R}$:*

$$x + y = \max(-x - y, 0) + \max(y + x, 0)$$
$$x < -y \leftrightarrow \max(-x - y, 0) < \max(y + x, 0)$$

**Proposition 1.** *Consider two-layer networks without biases (i.e., $b^{(l)} = 0$ for all layers l), for parameters $W^{(l)}$ and some $R \in \mathbb{R}^+$ as follows.*

$$W^{(1)} = R \left( \hat{\mu} + \hat{\nu} \quad -\hat{\mu} - \hat{\nu} \quad 0 \quad 0 \right)$$
$$W^{(2)} = \left( 1 \quad -1 \quad 0 \quad 0 \right)^T$$

*Then for any $\sigma > 0$, the defined networks realize the Bayes optimal classifier for the shifted centers data model.*

$$\hat{y}_i = \varphi(R(\langle X_i, \hat{\nu} \rangle + \langle X_i, \hat{\mu} \rangle))$$

*Proof.* Note that the output of the two-layer network is $\varphi([XW^{(1)}]_+ W^{(2)})$, which is interpreted as the probability with which the network believes that the input is in the class with label 1. The final prediction for the class label is thus assigned to be 1 if the output is $\geq 0.5$, and 0 otherwise. For each $i \in [n]$, we have that the output of the network on data point $i$ is

$$\hat{y}_i = \varphi(R([\langle X_i, \hat{\mu} + \hat{\nu} \rangle]_+ - [\langle X_i, -\hat{\mu} - \hat{\nu} \rangle]_+)) =$$
$$\varphi(R([\langle X_i, \hat{\mu} \rangle + \langle X_i, \hat{\nu} \rangle]_+ - [-\langle X_i, \hat{\mu} \rangle - \langle X_i, \hat{\nu} \rangle]_+)) = \varphi(R(\langle X_i, \hat{\mu} \rangle + \langle X_i, \hat{\nu} \rangle))$$

where the last equality is due to Fact 1. $\qquad \square$

## C.1 PROOF OF THEOREM 2 PART ONE

**Lemma 7.** *For some fixed $\mu, \nu \in \mathbb{R}^d$ and $\sigma^2 > 0$, the Bayes optimal classifier and let $h*(x) : \mathbb{R}^d \to \{0, 1\}$ be any binary classifier. For any $\epsilon \in (0, 1)$, If the probability for a point $X_i$ to misclassified is $\tau$, then w.p $1 - exp(-n^{(}1 - \epsilon)$ the fraction of misclassified nodes is*

$$\tau - n^{-\frac{\epsilon}{2}}$$

*Proof.* Define $M(n)$ to be the fraction of misclassified nodes. Define $x_i$ to be the indicator random variable $\mathbb{1}(X_i$ is misclassified). Then $x_i$ are Bernoulli random variables with mean at least $\tau$, and $\mathbb{E}(M(n)) = \frac{2}{n} \sum_{i \in [n]} \mathbb{E}(x_i) \geq \tau$. Using Hoeffding's inequality, we have that for any $t > 0$,

$$P(M(n) \geq \tau - t) \geq \Pr M(n) \geq \mathbb{E}(M(n)) - t \geq 1 - \exp(-nt^2).$$

Choosing $t = n^{-\epsilon/2}$ for any $\epsilon \in (0, 1)$ yields

$$P(M(n) \geq \tau - n^{-\epsilon/2}) \geq 1 - \exp(-n^{1-\epsilon}).$$

$\qquad \square$

**Theorem** (Restatement of Theorem 2 part one). *Let $X \in \mathbb{R}^{n \times d} \sim XOR\text{-}GMM\text{-}SC(n, d, \mu, \nu, \sigma^2)$. Assume that $\|\mu - \nu\|_2 = K\sigma$ and let $h(x) : \mathbb{R}^d \to \{0, 1\}$ be any binary classifier. Then for any $K > 0$ and any $\epsilon \in (0, 1)$, at least a fraction $\Phi_c(\frac{K}{2}) - O(n^{-\epsilon/2})$ of all data points are misclassified by $h$ with probability at least $1 - \exp(-2n^{1-\epsilon})$.*

*Proof.* We will upper bound the probability of the right classification similar to (Baranwal et al., 2022). We consider only class 1, since the analysis for class 0 is similar. For class 1, $i \in \{\mu, \nu\}$, we take a point from the center $\nu$, since the other case is symmetric. We can write $X_i = \nu + \sigma g_i$, where $g_i \sim N(0, I)$, then the probability of right classification:

$$P(right\ classification) = P(-\langle X_i, \mu \rangle < \langle X_i, \nu \rangle) = P(-\sigma \langle g_i, \hat{\mu} \rangle < \gamma' + \sigma \langle g_i, \hat{\nu} \rangle) =$$
$$P(\langle g_i, \hat{\nu} \rangle + \langle g_i, \hat{\mu} \rangle > -\frac{\gamma'}{\sigma}) = P(\langle g_i, \hat{\nu} \rangle + \langle g_i, \hat{\mu} \rangle > -\frac{K}{\sqrt{2}}) = 1 - P(\langle g_i, \hat{\nu} \rangle + \langle g_i, \hat{\mu} \rangle < -\frac{K}{\sqrt{2}})$$

Denote $Z_1 = \langle g_i, \hat{\nu} \rangle, Z_2 = \langle g_i, \hat{\mu} \rangle$

$$P(Z_1 + Z_2 < -K') = \int_{-\infty}^{\infty} \phi(z) \Phi(-K' - z) dz = \int_{-\infty}^{\infty} \phi(z) \Phi(-K' + z) dz = \Phi(-\frac{K}{2}) =$$
$$1 - \Phi(\frac{K}{2})$$

So we have:

$$P(X_i \ is \ misclassified) = \Phi(\tfrac{K}{2})$$

Now, applying Lemma 13 from the previous appendix on the total misclassification rate we get the desired result.

$\square$

### C.2 PROOF OF THEOREM 2 PART TWO

**Theorem** (Restatement of Theorem 2 part two). *Let $X \in \mathbb{R}^{n \times d} \sim XOR\text{-}GMM\text{-}SC(n, d, \mu, \nu, \sigma^2)$. For any $\epsilon > 0$, if the distance between the means is $|\mu - \nu|_2 = \Omega(\sigma(\log n)^{\frac{1}{2}+\epsilon})$, then for any $c > 0$, with probability at least $1 - O(n^{-c})$, there exists a two-layer that perfectly classify the data, and obtain a cross-entropy loss given by*

$$\ell_\theta(X) = C \exp(-\frac{R}{\sqrt{2}}\|\mu - \nu\|_2(1 \pm \sqrt{c}/(\log n)^\epsilon)),$$

*where $C \in [\frac{1}{2}, 1]$ is an absolute constant and $R$ is the optimality constraint from.*

*Proof.* We have $\hat{y}_i = \varphi(R(\langle X_i, \hat{\mu}_i \rangle + \langle X_i, \hat{\nu}_i \rangle))$ and $l_i(X, \theta) = -y_i \log(\hat{y}_i) - (1 - y_i) \log(1 - \hat{y}_i) = \log\left(1 + \exp\left((1 - 2y_i)R(\langle X_i, \hat{\mu}_i \rangle + \langle X_i, \hat{\nu}_i \rangle)\right)\right)$. We can apply the same Gaussian concentration arguments as in (Baranwal et al., 2022). We have with probability at least $1 - \frac{n^{-c}}{\sqrt{\pi(c+1)\log n}}$ that

$$\langle X_i, \hat{m}_c \rangle = \langle \mathbb{E}(X_i), \hat{m}_c \rangle \pm O(\sigma\sqrt{c\log n}). \forall i \in [n] \ for \ m_c \in \{\mu, -\mu, \nu, -\nu\}$$

Let's look at the expression inside the prediction $\hat{y}_i$, namely $\langle X_i, \hat{\mu}_i \rangle + \langle X_i, \hat{\nu}_i \rangle$.

For $X_i \in \{\mu, \nu\}$ i.e in class 1, then, this expression becomes:

$$\gamma'(1 \pm O(\sqrt{\tfrac{c}{\log n}}))$$

For $X_i \in \{-\mu, -\nu\}$ i.e in class 0, then, this expression becomes:

$$-\gamma'(1 \pm O(\sqrt{\tfrac{c}{\log n}}))$$

We obtain for all $i \in [n]$,

$$\ell_i(X, \theta) = \log(1 + \exp(-R\gamma'(1 \pm o_n(1)))),$$

where the error term $o_n(1) = \sqrt{\tfrac{c}{\log n}}$. The total loss is then given by

$$\ell_\theta(X) = \frac{1}{n} \sum \ell_i(X, \theta) = \log(1 + \exp(-R\gamma'(1 + o_n(1)))).$$

Next, Fact 2 implies that for $t < 0$, $\frac{e^t}{2} \leq \log(1 + e^t) \leq e^t$, hence, we have that there exists a constant $C \in [\frac{1}{2}, 1]$ such that

$$\ell_\theta(X) = C \exp(-R\gamma'(1 + o_n(1)))).$$

Note that by scaling the optimality constraint $R$, the loss can go arbitrarily close to 0.

$\square$

**Lemma 8.** *Let $h(x) = \langle x, \hat{\nu} \rangle + \langle x, \hat{\mu} \rangle$. Then, GCN with weights as defined above satisfies:*

$$\hat{y}_i = \varphi(f_i^{(L)}(X)) = \varphi(\tfrac{R sgn(p-q)}{deg(i)} \sum_{j \in [n]} a_{ij} h(X_j))$$

*Proof.* We will prove for the 2-layer networks. Notice that for this case, we apply the convolution at the end, the output of the last layer for data $(A, X)$ is $f_i^{(2)}(X) = D^{-1}A[XW^{(1)}]_+W^{(2)}$. Then we have

$$f_i^{(2)}(X) = \frac{R}{deg(i)} \sum_{j \in [n]} a_{ij}(\langle X_j, \hat{\nu} \rangle + \langle X_j, \hat{\mu} \rangle) = \frac{R}{deg(i)} \sum_{j \in [n]} a_{ij}h(X_j)$$

$\square$

**Lemma 9.** *Let $h(x) = \langle x, \hat{\nu} \rangle + \langle x, \hat{\mu} \rangle$. Then:*

$$\mathbf{E}(h(X_i)) = \mathbf{E}(\langle x, \hat{\nu} \rangle + \langle x, \hat{\mu} \rangle) = \mathbf{E}(\langle x, \hat{\nu} \rangle) + \mathbf{E}(\langle x, \hat{\mu} \rangle) = \langle \mathbf{E}(x), \hat{\nu} \rangle + \langle \mathbf{E}(x), \hat{\mu} \rangle =$$
$$\begin{cases} \gamma' & i \in \{\mu, \nu\} = \{C_1\} \\ -\gamma' & i \in \{-\mu, -\nu\} = \{C_0\} \end{cases}$$

similarly to (Baranwal et al., 2022).

### C.3 PROOF OF THEOREM 3

**Lemma 10.** *Let $h(x) = \langle x, \hat{\nu} \rangle + \langle x, \hat{\mu} \rangle$ for any $x \in \mathbb{R}^d$. Consider the two-layer networks in Proposition 1 where the weight parameter of the first layer, $W^{(1)}$, is scaled by a factor of $\varepsilon = sgn(p - q)$. If a graph convolution is added to these networks in either the second or the third layer then for a sample $(A, X) \sim XOR - CSBM - SC(n, d, \mu, \nu, \sigma^2, p, q)$, the output of the networks for a point $i \in [n]$ is*

$$\hat{y}_i = \varphi(f_i^{(2)}(X)) = \varphi\left(R\varepsilon \frac{1}{deg(i)} \sum_{j \in [n]} a_{ij}h(X_j)\right).$$

*Proof.* The networks with scaled parameters are given as follows. For the two-layer network, when a graph convolution is applied at the second layer of this two-layer MLP, the output of the last layer for data $(A, X)$ is $f_i^{(2)}(X) = D^{-1}A[XW^{(1)}]_+W^{(2)}$. Then we have

$$f_i^{(2)}(X) = \frac{R\varepsilon}{deg(i)} \sum_{j \in [n]} a_{ij}\left(\langle x, \hat{\nu} \rangle + \langle x, \hat{\mu} \rangle\right) = R\varepsilon\left(\frac{1}{deg(i)} \sum_{j \in [n]} a_{ij}h(X_j)\right)$$

$\square$

**Theorem** (Restatement of Theorem 3). *Let $(A, X) \sim XOR\text{-}CSBM\text{-}SC(n, d, \mu, \nu, \sigma^2, p, q)$. Then there exists a two-layer network and a three-layer network with the following properties: If the intra-class and inter-class edge probabilities are $p, q = \Omega(\frac{\log^2 n}{n})$, and the distance between the means is $||\mu - \nu||_2 = \Omega(\frac{\sigma \log n}{\sqrt{n(p+q)}})$, then for any $c > 0$, with probability at least $1 - O(n^{-c})$, the networks equipped with a graph convolution in the second or the third layer perfectly classify the data, and obtain the following loss:*

$$\ell_\theta(A, X) = C' \exp\left(-R|||\mu - \nu|||_2\left|\frac{p - q}{p + q}\right|\left(1 \pm \sqrt{\frac{c}{\log n}}\right)\right),$$

*where $C > 0$ and $C' \in [\frac{1}{2}, 1]$ are constants.*

*Proof.* Let's look at the Bayes optimal classifiers for this model and for original model.

$$h_{orig}^*(x) = |\langle x, \nu \rangle| - |\langle x, \mu \rangle|$$
$$h_{curr}^*(x) = \langle x, \nu \rangle + \langle x, \mu \rangle$$

We have

$$h_{orig}^* \text{ is } \rho - Lipschitz \leftrightarrow h_{curr}^* \text{ is } \rho - Lipschitz$$

Thus, we can reuse from Baranwal et al. (2022) arguments used to characterize $f_i^{(2)}(X)$. Specifically: Gaussian concentration -

$$P(\tfrac{1}{R}|f_i^{(2)}(X) - \mathbb{E}[f_i^{(L)}(X)]| > \delta \mid A) \le 2\exp(-\tfrac{\delta^2 deg(i)}{4\sigma^2})$$

Let $\varepsilon = sgn(p - q)$, $\frac{\varepsilon(p-q)}{p+q} = \frac{|p-q|}{p+q} = \Gamma(p, q)$. Note that the process of creating the edges remains the same between this model and the original model, because it depends solely on the nodes' labels. Thus, we have from Proposition A.1 in (Baranwal et al., 2022):

$$\sum_{j \in C_1} a_{ij} - \sum_{j \in C_0} a_{ij} = (2\epsilon_i - 1)\tfrac{p-q}{p+q}(1 + o_n(1))$$

$$f_i^{(2)}(X) = \mathbb{E}(f_i^{(2)}(X)) \pm O(R\sigma\sqrt{\tfrac{c\log n}{n(p+q)}})$$
$$= \tfrac{R\varepsilon}{deg(i)} \sum_{j \in [n]} a_{ij}\mathbb{E}(h(X_j)) \pm o_n(R\sigma) =$$
$$\tfrac{R\varepsilon\gamma'}{deg(i)}(\sum_{j \in C_1} a_{ij} - \sum_{j \in C_0} a_{ij}) \pm o_n(R\sigma) = \text{(using Lemma 9)}$$
$$(2\epsilon_i - 1)R\Gamma(p, q)\gamma'(1 \pm o_n(1)) \pm o_n(R\sigma).$$

We need $\gamma' = \Omega(o_n(R\sigma)) = Omega\left(\sigma\frac{\log n}{\sqrt{n(p+q)}}\right)$.

So we have for some constant $C > 0$:

$$f_i^{(2)}(X) = (2\epsilon_i - 1)CR\gamma'\Gamma(p, q)(1 \pm o_n(1))$$

Recall that the loss for node $i$ is given by

$$\ell_\theta^{(i)}(A, X) = \log(1 + e^{(1-2\epsilon_i)f_i^{(L)}(X)}) = \log(1 + \exp(-CR\gamma'\Gamma(p, q)(1 \pm o_n(1)))).$$

Next, Fact 2 implies that for any $t < 0$, $\frac{e^t}{2} \le \log(1 + e^t) \le e^t$, hence, we have for some $C' \in [\frac{1}{2}, 1]$ that
$$\ell_\theta^{(i)}(A, X) = C' \exp(-CR\gamma'\Gamma(p, q)(1 \pm o_n(1))).$$

The total loss is given by $\frac{1}{n} \sum_{i \in [n]} \ell_\theta^{(i)}(A, X)$. Thus

$$\ell_\theta(A, X) = C' \exp(-CR\gamma'\Gamma(p, q)(1 \pm o_n(1))).$$

We can observe the loss decreases as $\gamma$ (distance between the means) increases, and increases if $\sigma^2$ (variance of the data) increases. $\qquad\square$

## D    CALCULATIONS FOR THE IMBALANCED CASE

We denote $i \in |\mu| \leftrightarrow i \in \{\mu, -\mu\}$

**Proposition 2.** *For any constant $c > 0$, with probability at least $1 - 2n^{-c}$, we have for all $i \in [n]$ that*

$$deg(i) = n(w_0 p + w_1 q)(1 \pm o_n(1)) \text{ for } i \in |\mu|$$
$$deg(i) = n(w_1 p + w_0 q)(1 \pm o_n(1)) \text{ for } i \in |\nu|$$
$$\tfrac{1}{deg(i)} = \tfrac{1}{n(w_0 p + w_1 q)}(1 \pm o_n(1)) \text{ for } i \in |\mu|$$
$$\tfrac{1}{deg(i)} = \tfrac{1}{n(w_1 p + w_0 q)}(1 \pm o_n(1)) \text{ for } i \in |\nu|$$
$$\tfrac{1}{deg(i)}\left(\sum_{j \in C_1} a_{ij} - \sum_{j \in C_0} a_{ij}\right) = \tfrac{w_1 q - w_0 p}{w_0 p + w_1 q}(1 \pm o_n(1)) \text{ for } i \in |\mu|$$
$$\tfrac{1}{deg(i)}\left(\sum_{j \in C_1} a_{ij} - \sum_{j \in C_0} a_{ij}\right) = \tfrac{w_1 p - w_0 q}{w_1 p + w_0 q}(1 \pm o_n(1)) \text{ for } i \in |\nu|$$

*Proof.* $deg(i)$ is a sum of n Bernoulli random variables. For $i \in |\mu|$, the probability of an edge is:

$$p(egde) = p(edge|same\ class) \cdot p(same\ class) + p(edge|same\ class) \cdot p(same\ class) =$$
$$p \cdot w_0 + q \cdot w_1$$

similarly for $i \in |\nu|$:

$$p(egde) = p(edge|same\ class) \cdot p(same\ class) + p(edge|same\ class) \cdot p(same\ class) =$$
$$p \cdot w_1 + q \cdot w_0$$

By the Chernoff bound we get, w.h.p:

$$P[deg(i) \in [\tfrac{n}{2}(p \cdot w_0 + q \cdot w_1)(1-\delta), \tfrac{n}{2}(p \cdot w_0 + q \cdot w_1)(1+\delta)] \leq 2\exp(-Cn(p \cdot w_0 + q \cdot w_1)\delta_{|\mu|}^2)$$
$$\text{for } i \in |\mu|$$
$$P[deg(i) \in [\tfrac{n}{2}(p \cdot w_1 + q \cdot w_0)(1-\delta), \tfrac{n}{2}(p \cdot w_1 + q \cdot w_0)(1+\delta)] \leq 2\exp(-Cn(p \cdot w_1 + q \cdot w_0)\delta_{|\nu|}^2)$$
$$\text{for } i \in |\nu|$$

for some $C > 0$. Now choose $\delta_{|\mu|} = \sqrt{\frac{(c+1)\log n}{Cn(p \cdot w_0 + q \cdot w_1)}}$ and $\delta_{|\mu|} = \sqrt{\frac{(c+1)\log n}{Cn(p \cdot w_0 + q \cdot w_1)}}$ for a large constant $c > 0$. Note that since $p, q = \Omega(\frac{\log^2 n}{n})$ and $w_1 = \Omega(1)$, we have that $\delta = O(\sqrt{\frac{c}{\log n}}) = o_n(1)$. Then following a union bound over $i \in [n]$, we obtain that with probability at least $1 - 2n^{-c}$,

$$deg(i) = n(w_0 p + w_1 q)(1 \pm o_n(1)) \text{ for } i \in |\mu|$$
$$deg(i) = n(w_1 p + w_0 q)(1 \pm o_n(1)) \text{ for } i \in |\nu|$$
$$\frac{1}{deg(i)} = \frac{1}{n(w_0 p + w_1 q)}(1 \pm o_n(1)) \text{ for } i \in |\mu|$$
$$\frac{1}{deg(i)} = \frac{1}{n(w_1 p + w_0 q)}(1 \pm o_n(1)) \text{ for } i \in |\nu|$$

Note that $|C_b| = w_b n$. Also note that $\sum_{j \in C_b} a_{ij}$ for any $b \in \{0, 1\}$ is a sum of independent Bernoulli random variables. Hence, we have by similar arguments

$$\sum_{j \in C_b} a_{ij} = w_b n p (1 \pm o_n(1)) \text{ for } i \in C_b$$

We can calculate this to each $i \in C_b, j \in C_{b'}$. Combining it all we have that with probability at least $1 - 2n^{-c}$,

$$\frac{1}{deg(i)}\left(\sum_{j \in C_1} a_{ij} - \sum_{j \in C_0} a_{ij}\right) = \frac{w_1 p - w_0 q}{w_1 p + w_0 q}(1 + o_n(1)) \ for\ i \in |\nu|$$

$$\frac{1}{deg(i)}\left(\sum_{j \in C_1} a_{ij} - \sum_{j \in C_0} a_{ij}\right) = \frac{w_1 q - w_0 p}{w_0 p + w_1 q}(1 + o_n(1)) \ for\ i \in |\mu|$$

$\square$

**Lemma 11.** *Assume $x, y \in \mathbb{R}, c > 0$, We can linearly approximate the solution to*

$$cosh(x) < c \cdot cosh(y)$$

*by*

$$|x| < |y| + \ln(c)$$

*Proof.* Let's start with the inequality:

$$cosh(x) \leq c \cdot cosh(y)$$
$$-cosh^{-1}(c \cdot cosh(y)) \leq x \leq cosh^{-1}(c \cdot cosh(y))$$

Notice that:

$$cosh^{-1}(z) = ln(z + \sqrt{z^2 - 1})$$

Thus:

$$cosh^{-1}(c \cdot cosh(y)) = ln(c \cdot cosh(y) + \sqrt{(c \cdot cosh(y))^2 - 1})$$

But:

$$(c \cdot cosh(y))^2 - 1 = c^2 \cdot (cosh(y)^2 - 1) + (c^2 - 1) = c^2 \cdot sinh(y)^2 + (c^2 - 1)$$

Substituting it into the expression:

$$ln(c \cdot cosh(y) + \sqrt{(c \cdot cosh(y))^2 - 1}) = ln(c \cdot cosh(y) + \sqrt{c^2 \cdot sinh(y)^2 + (c^2 - 1)})$$

We want to transform this expression into a linear expression. In order to achieve that, we change the expression to:

$$ln(c \cdot cosh(y) + \sqrt{c^2 \cdot sinh(y)^2 + (c^2 - 1)}) \approx ln(c \cdot cosh(y) + \sqrt{c^2 \cdot sinh(y)^2})$$

And calculating this:

$$ln(c \cdot cosh(y) + \sqrt{c^2 \cdot sinh(y)^2}) = ln(c \cdot cosh(y) + c|sinh(y)|)$$

When $y > 0$, we have $sinh(y) > 0$ and $cosh(y) + |sinh(y)| = e^y = e^{|y|}$. When $y < 0$, we have $sinh(y) < 0$ and $cosh(y) + |sinh(y)| = e^{-y} = e^{|y|}$.
So, all in all we get:

$$ln(c \cdot cosh(y) + c|sinh(y)|) = ln(c \cdot e^{|y|}) = ln(c) + |y|$$

And going back to the original inequality:

$$-cosh^{-1}(c \cdot cosh(y)) \le x \le cosh^{-1}(c \cdot cosh(y))$$
$$-(ln(c) + |y|) \le x \le ln(c) + |y|$$
$$|x| \le ln(c) + |y|$$

$\square$

**Lemma 12.** *For some fixed $\mu, \nu \in \mathbb{R}^d$ and $\sigma^2 > 0$, the Bayes optimal classifier, $h^*(x) : \mathbb{R}^d \to \{0, 1\}$ for the imbalanced data model is approximately:*

$$h^*(x) = \mathbb{1}(|\langle x, \nu \rangle| < |\langle x, \mu \rangle| + \sigma^2 logit(w_0))$$

*Proof.* Note that $P(y = b) = w_b$ for $b \in \{0, 1\}$. Let $f(x)$ denote the density function of a continuous random vector $x$. Therefore, for any $b \in \{0, 1\}$,

$$P(y = b | x) = \frac{P(y=b) f_{x|y}(x|y=b)}{\sum_{c \in \{0,1\}} P(y=c) f_{x|y}(x|y=c)} = \frac{1}{1 + \frac{w_{1-b}}{w_b} \frac{f(x|y=1-b)}{f(x|y=b)}}$$

Computing it for label 0, we need:

$$\frac{w_1}{w_0} \frac{f(x|y=1-b)}{f(x|y=b)} < 1$$
$$\frac{w_1}{w_0} \frac{cosh(\frac{\langle x, \nu \rangle}{\sigma^2})}{cosh(\frac{\langle x, \mu \rangle}{\sigma^2})} exp(\frac{||\mu||^2 - ||\nu||^2}{2\sigma^2}) < 1$$
$$\frac{w_1}{w_0} \frac{cosh(\frac{\langle x, \nu \rangle}{\sigma^2})}{cosh(\frac{\langle x, \mu \rangle}{\sigma^2})} < 1$$
$$cosh(\frac{\langle x, \nu \rangle}{\sigma^2}) < \frac{w_0}{w_1} cosh(\frac{\langle x, \mu \rangle}{\sigma^2})$$
$$|\langle x, \nu \rangle| < |\langle x, \mu \rangle| + \sigma^2 \ln(\frac{w_0}{w_1}) = |\langle x, \mu \rangle| + \sigma^2 logit(w_0) \text{ (By Lemma 11)}$$

where in the second to last inequality, we used $||\mu||=||\nu||$. □

To give some intuition, let's look at the decision boundaries of the real expression and our approximation.

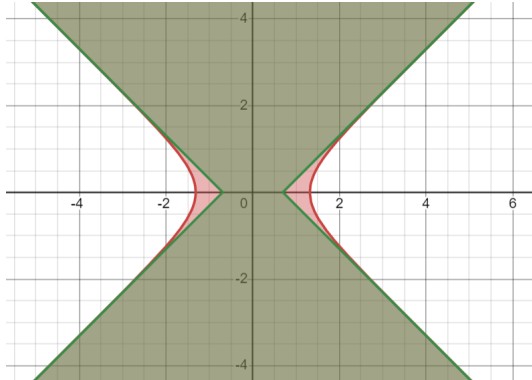

Figure 14: Decision boundaries of the real inequality compared to the approximated inequality where $c = 2$. The red area represents the are where the first inequality holds, and vice versa for the green area. As we can see, the difference is very small, and mainly appears where $|x| \approx 1, y \approx 0$.

**Proposition 3.** *Consider two-layer network of the same form described in Baranwal et al. (2022), for bias in the last layer $b^{(L)} = -R\sigma^2 \ln(\frac{w_0}{w_1})$, and $W^{(l)}$ and some $R \in \mathbb{R}^+$ as follows.*

$$W^{(1)} = R \begin{pmatrix} \mu & -\mu & \nu & -\nu \end{pmatrix}, W^{(2)} = \begin{pmatrix} -1 & -1 & 1 & 1 \end{pmatrix}^T.$$

*Then for any $\sigma > 0$, the defined networks realize the approximate Bayes optimal classifier for the imbalanced data model.*

*Proof.* Notice that the only difference between our parameters and the parameters in Baranwal et al. (2022) is our bias in the last layer. In their case we have:

$$\hat{y}_i = \varphi((R(|\langle X_i, \hat{\nu}\rangle| - |\langle X_i, \hat{\mu}\rangle|))$$

thus, adding the bias in the last layer we get:

$$\hat{y}_i = \varphi((R(|\langle X_i, \hat{\nu}\rangle| - |\langle X_i, \hat{\mu}\rangle| - \sigma^2 \ln(\frac{w_0}{w_1})))$$

□

### D.1 PROOF OF THEOREM 4 PART ONE

**Lemma 13.** *For some fixed $\mu, \nu \in \mathbb{R}^d$ and $\sigma^2 > 0$, the Bayes optimal classifier and let $h^*(x) : \mathbb{R}^d \to \{0, 1\}$ be any binary classifier. For any $\epsilon \in (0, 1)$, If the probability for a point $X_i$ to misclassified is $\tau$, then w.p $1 - exp(-n^{(1-\epsilon)})$ the fraction of misclassified nodes is*

$$\tau - n^{-\frac{\epsilon}{2}}$$

*Proof.* See Lemma 7 □

**Theorem** (Restatement of part one of Theorem 4). *Let $X \in \mathbb{R}^{n \times d} \sim XOR - GMM - I(n, d, \mu, \nu, \sigma^2)$. Assume that $\|\mu - \nu\|_2 = K\sigma$ and let $h(x) : \mathbb{R}^d \to \{0, 1\}$ be any binary classifier. Then for $K > 0, K_{|\mu|,i} = \frac{K}{\sqrt{2}} + \sigma^2 \ln(\frac{w_0}{w_1}), K_{|\nu|,i} = \frac{K}{\sqrt{2}} + \sigma^2 \ln(\frac{w_1}{w_0}) = \frac{K}{\sqrt{2}} - \sigma^2 \ln(\frac{w_0}{w_1})$ and*

*any $\epsilon \in (0,1)$, at least a fraction of*

$$w_0 \cdot \begin{cases} 1 - 2\Phi_c(\frac{K_{|\mu|,i}}{\sqrt{2}})^2 & if \ K_{|\mu|,i} \geq 0 \\ 4\Phi(\frac{K_{|\mu|,i}}{\sqrt{2}}) - 2\Phi(\frac{K_{|\mu|,i}}{\sqrt{2}})^2 + 4\Phi(K_{|\mu|,i})^2 - 4\Phi(K_{|\mu|,i}) & if \ K_{|\mu|,i} < 0 \end{cases}$$

$$+$$

$$w_1 \cdot \begin{cases} 1 - 2\Phi_c(\frac{K_{|\nu|,i}}{\sqrt{2}})^2 & if \ K_{|\nu|,i} \geq 0 \\ 4\Phi(\frac{K_{|\nu|,i}}{\sqrt{2}}) - 2\Phi(\frac{K_{|\nu|,i}}{\sqrt{2}})^2 + 4\Phi(K_{|\nu|,i})^2 - 4\Phi(K_{|\nu|,i}) & if \ K_{|\nu|,i} < 0 \end{cases}$$

$$- O(n^{-\epsilon/2})$$

*of all data points are misclassified by h with probability at least $1 - \exp(-2n^{1-\epsilon})$.*

*Proof.* We will upper bound the probability of the right classification similar to (Baranwal et al., 2022). We consider only class 0, since the analysis for class 1 is similar. Define $c = \sigma^2 \ln(\frac{w_0}{w_1}), \gamma \leq \sigma K, K_{|\mu|,i} = \frac{K}{\sqrt{2}} + c$. For $i \in |\mu|$, we can write $X_i = \mu + \sigma g_i$, where $g_i \sim N(0, I)$, then the probability of right classification:

$$P(|\langle x, \nu \rangle| < |\langle x, \mu \rangle| + \sigma^2 \ln(\frac{w_0}{w_1})) = P(|\langle x, \nu \rangle| < |\langle x, \mu \rangle| + c) \leq P(|\langle g_i, \hat{\nu} \rangle| - |\langle g_i, \hat{\mu} \rangle| \leq \frac{K}{\sqrt{2}} + c) = $$
$$P(|\langle g_i, \hat{\nu} \rangle| - |\langle g_i, \hat{\mu} \rangle| \leq K_{|\mu|,i})$$

Notice that this expression is the same as in Baranwal et al. (2022) in their part one of Theorem 1. Thus applying the same calculations we get:

$$P(|\langle g_i, \hat{\nu} \rangle| - |\langle g_i, \hat{\mu} \rangle| \leq K_{|\mu|,i}) = 1 - 2\Phi_c(\frac{K_{|\mu|,i}}{\sqrt{2}})^2$$

However, for some combination of $\gamma$ and $w_0$, we get $K_{|\mu|,i} < 0$. Thus, we can't calculate the integral in the same way for this case. The integral boundaries become $max(0, -K_{|\mu|,i})$ and $\infty$. But calculating with $-K_{|\mu|,i}$ doesn't have a closed from according to Owen (1980) Table 1:10,010,4, so we will need to estimate it.

**estimating**

Assuming $K_{|\mu|,i} < 0$:

$$P(|Z_1| - |Z_2| \leq K_{|\mu|,i}) = 4P(Z_1 - Z_2 \leq K_{|\mu|,i}, Z_1, Z_2 \geq 0) = $$
$$4\int_{-K_{|\mu|,i}}^{\infty} \phi(w)(\Phi(w + K_{|\mu|,i}) - \frac{1}{2})dw = 4\int_{-K_{|\mu|,i}}^{\infty} \phi(w)\Phi(w + K_{|\mu|,i})dw - 2\int_{-K_{|\mu|,i}}^{\infty} \phi(w)dw = $$
$$4\int_{-K_{|\mu|,i}}^{\infty} \phi(w)\Phi(w + K_{|\mu|,i})dw - 2(1 - \Phi(-K_{|\mu|,i})) = $$
$$4\int_{-K_{|\mu|,i}}^{\infty} \phi(w)\Phi(w + K_{|\mu|,i})dw - 2\Phi(K_{|\mu|,i})$$

$$4\int_{0}^{-K_{|\mu|,i}} \phi(w)\Phi(w + K_{|\mu|,i})dw \approx 4\int_{0}^{-K_{|\mu|,i}} \phi(w)\Phi(K_{|\mu|,i})dw = $$
$$4\Phi(K_{|\mu|,i})\int_{0}^{-K_{|\mu|,i}} \phi(w)dw = 4\Phi(K_{|\mu|,i})\left(\Phi(-K_{|\mu|,i}) - \frac{1}{2}\right) = $$
$$4\Phi(K_{|\mu|,i})\Phi(-K_{|\mu|,i}) - 2\Phi(K_{|\mu|,i}) = 2\Phi(K_{|\mu|,i}) - 4\Phi(K_{|\mu|,i})^2$$

$$4\int_{-K_{|\mu|,i}}^{\infty} \phi(w)\Phi(w + K_{|\mu|,i})dw = 4\int_{0}^{\infty} \phi(w)\Phi(w + K_{|\mu|,i})dw - 4\int_{0}^{-K_{|\mu|,i}} \phi(w)\Phi(w + $$
$$K_{|\mu|,i})dw = 2\Phi(\frac{K_{|\mu|,i}}{\sqrt{2}}) + 2\Phi(\frac{K_{|\mu|,i}}{\sqrt{2}})\Phi_c(\frac{K_{|\mu|,i}}{\sqrt{2}}) - 4\int_{0}^{-K_{|\mu|,i}} \phi(w)\Phi(w + K_{|\mu|,i})dw \approx $$
$$2\Phi(\frac{K_{|\mu|,i}}{\sqrt{2}}) + 2\Phi(\frac{K_{|\mu|,i}}{\sqrt{2}})\Phi_c(\frac{K_{|\mu|,i}}{\sqrt{2}}) + 4\Phi(K_{|\mu|,i})^2 - 2\Phi(K_{|\mu|,i})$$

$$P(|Z_1| - |Z_2| \leq K_{|\mu|,i}) \approx 2\Phi(\frac{K_{|\mu|,i}}{\sqrt{2}}) + 2\Phi(\frac{K_{|\mu|,i}}{\sqrt{2}})\Phi_c(\frac{K_{|\mu|,i}}{\sqrt{2}}) + 4\Phi(K_{|\mu|,i})^2 - 4\Phi(K_{|\mu|,i}) = $$
$$4\Phi(\frac{K_{|\mu|,i}}{\sqrt{2}}) - 2\Phi(\frac{K_{|\mu|,i}}{\sqrt{2}})^2 + 4\Phi(K_{|\mu|,i})^2 - 4\Phi(K_{|\mu|,i})$$

For class 1, define $K_{|\nu|,i} = \frac{K}{\sqrt{2}} + \sigma^2 \log(\frac{w_1}{w_0})$, doing similar calculations, we get:

$$P(right\ classification | class\ 1) =$$

$$\begin{cases} 1 - 2\Phi_c(\frac{K_{|\nu|,i}}{\sqrt{2}})^2 & if\ K_{|\nu|,i} \geq 0 \\ 4\Phi(\frac{K_{|\nu|,i}}{\sqrt{2}}) - 2\Phi(\frac{K_{|\nu|,i}}{\sqrt{2}})^2 + 4\Phi(K_{|\nu|,i})^2 - 4\Phi(K_{|\nu|,i}) & if\ K_{|\nu|,i} < 0 \end{cases}$$

Notice that:

$$P(right\ classification) =$$
$$P(right\ classification | class\ 0)P(class\ 0) + P(right\ classification | class\ 1)P(class\ 1)$$

So overall, we get:

$$P(right\ classification) = w_0 \cdot$$

$$\begin{cases} 1 - 2\Phi_c(\frac{K_{|\mu|,i}}{\sqrt{2}})^2 & if\ K_{|\mu|,i} \geq 0 \\ 4\Phi(\frac{K_{|\mu|,i}}{\sqrt{2}}) - 2\Phi(\frac{K_{|\mu|,i}}{\sqrt{2}})^2 + 4\Phi(K_{|\mu|,i})^2 - 4\Phi(K_{|\mu|,i}) & if\ K_{|\mu|,i} < 0 \end{cases} +$$

$$w_1 \cdot$$

$$\begin{cases} 1 - 2\Phi_c(\frac{K_{|\nu|,i}}{\sqrt{2}})^2 & if\ K_{|\nu|,i} \geq 0 \\ 4\Phi(\frac{K_{|\nu|,i}}{\sqrt{2}}) - 2\Phi(\frac{K_{|\nu|,i}}{\sqrt{2}})^2 + 4\Phi(K_{|\nu|,i})^2 - 4\Phi(K_{|\nu|,i}) & if\ K_{|\nu|,i} < 0 \end{cases}$$

Now, applying Lemma 13 on the total misclassification rate we get the desired result. $\square$

### D.2 PROOF OF THEOREM 4 PART TWO

**Theorem** (Restatement of part two of Theorem 4). *Let $X \in \mathbb{R}^{n \times d} \sim XOR\text{-}GMM\text{-}SC(n, d, \mu, \nu, \sigma^2)$. Then we have the following: For any $\epsilon > 0$, if the distance between the means is $|\mu - \nu|_2 = \Omega(max(\sigma(\log n)^{\frac{1}{2}+\epsilon}, \sigma^2 |logit(w_0)|))$, assume for simplicity's sake $K_i > 0$, we have:*

$$accuracy = P(right\ classification) = w_0 \cdot \left( 1 - 2\Phi_c \left( \frac{K}{2} + \frac{\sigma^2 \ln(\frac{w_0}{w_1})}{\sqrt{2}} \right)^2 \right) +$$

$$w_1 \cdot \left( 1 - 2\Phi_c \left( \frac{K}{2} - \frac{\sigma^2 \ln(\frac{w_0}{w_1})}{\sqrt{2}} \right)^2 \right) \pm O(n^{-\epsilon/2})$$

$$precision = \frac{w_0 \cdot \left( 1 - 2\Phi_c \left( \frac{K}{2} + \frac{\sigma^2 \ln(\frac{w_0}{w_1})}{\sqrt{2}} \right)^2 \right) \pm O(n^{-\epsilon/2})}{w_0 \cdot \left( 1 - 2\Phi_c \left( \frac{K}{2} + \frac{\sigma^2 \ln(\frac{w_0}{w_1})}{\sqrt{2}} \right)^2 \right) + w_1 \cdot \left( 2\Phi_c \left( \frac{K}{2} - \frac{\sigma^2 \ln(\frac{w_0}{w_1})}{\sqrt{2}} \right)^2 \right) \pm O(n^{-\epsilon/2})}$$

$$recall = 1 - 2\Phi_c \left( \frac{K}{2} + \frac{\sigma^2 \ln(\frac{w_0}{w_1})}{\sqrt{2}} \right)^2 \pm O(n^{-\epsilon/2})$$

$$f\text{-}score =$$

$$\frac{2w_0 \cdot \left( 1 - 2\Phi_c \left( \frac{K}{2} + \frac{\sigma^2 \ln(\frac{w_0}{w_1})}{\sqrt{2}} \right)^2 \right)^2 \pm O(n^{-\epsilon/2})}{2w_0 \cdot \left( 1 - 2\Phi_c \left( \frac{K}{2} + \frac{\sigma^2 \ln(\frac{w_0}{w_1})}{\sqrt{2}} \right)^2 \right)^2 + w_1 \left( 1 - 2\Phi_c \left( \frac{K}{2} + \frac{\sigma^2 \ln(\frac{w_0}{w_1})}{\sqrt{2}} \right)^2 \right) \left( 2\Phi_c \left( \frac{K}{2} - \frac{\sigma^2 \ln(\frac{w_0}{w_1})}{\sqrt{2}} \right)^2 \right) \pm O(n^{-\epsilon/2})}$$

*Proof.* Let's calculate the precision, the recall and the f-score. First, we will calculate true positive, false positive, false negative:

$$true\ positive = tp = P(positive)P(true|positive) = w_0 \cdot (1 - 2\Phi_c(\frac{K}{2} + \frac{\sigma^2 \ln(\frac{w_0}{w_1})}{\sqrt{2}})^2)$$

$$fp = P(negative)P(false|negative) = w_1 \cdot (2\Phi_c(\frac{K}{2} - \frac{\sigma^2 \ln(\frac{w_0}{w_1})}{\sqrt{2}})^2)$$

$$fn = P(positive)P(false|positive) = w_0 \cdot (2\Phi_c(\frac{K}{2} + \frac{\sigma^2 \ln(\frac{w_0}{w_1})}{\sqrt{2}})^2)$$

Using Similar arguments to Lemma 13, we can see that w.h.p these are the metrics across all of the data with a factor of $\pm O(n^{-\epsilon/2})$.

$$precision = \frac{tp}{tp + fp} =$$

$$\frac{w_0 \cdot \left(1 - 2\Phi_c\left(\frac{K}{2} + \frac{\sigma^2 \ln(\frac{w_0}{w_1})}{\sqrt{2}}\right)^2\right) \pm O(n^{-\epsilon/2})}{w_0 \cdot \left(1 - 2\Phi_c\left(\frac{K}{2} + \frac{\sigma^2 \ln(\frac{w_0}{w_1})}{\sqrt{2}}\right)^2\right) + w_1 \cdot \left(2\Phi_c\left(\frac{K}{2} - \frac{\sigma^2 \ln(\frac{w_0}{w_1})}{\sqrt{2}}\right)^2\right) \pm O(n^{-\epsilon/2})}$$

$$recall = \frac{tp}{tp + fn} = 1 - 2\Phi_c\left(\frac{K}{2} + \frac{\sigma^2 \ln(\frac{w_0}{w_1})}{\sqrt{2}}\right)^2 \pm O(n^{-\epsilon/2})$$

$$f\text{-}score = 2\frac{precision \cdot recall}{precision + recall} =$$

$$\frac{2w_0 \cdot \left(1 - 2\Phi_c\left(\frac{K}{2} + \frac{\sigma^2 \ln(\frac{w_0}{w_1})}{\sqrt{2}}\right)^2\right)^2 \pm O(n^{-\epsilon/2})}{2w_0 \cdot \left(1 - 2\Phi_c\left(\frac{K}{2} + \frac{\sigma^2 \ln(\frac{w_0}{w_1})}{\sqrt{2}}\right)^2\right)^2 + w_1 \left(1 - 2\Phi_c\left(\frac{K}{2} + \frac{\sigma^2 \ln(\frac{w_0}{w_1})}{\sqrt{2}}\right)^2\right)\left(2\Phi_c\left(\frac{K}{2} - \frac{\sigma^2 \ln(\frac{w_0}{w_1})}{\sqrt{2}}\right)^2\right) \pm O(n^{-\epsilon/2})}$$

$\square$

### D.3 Proof of Theorem 4 part three

**Fact 2.** *For any $x \in [0, 1]$, $\frac{x}{2} \leq \log(1 + x) \leq x$.*

**Theorem** (Restatement of part three of Theorem 4). *Let $X \in \mathbb{R}^{n \times d} \sim XOR\text{-}GMM\text{-}SC(n, d, \mu, \nu, \sigma^2)$. For any $\epsilon > 0$, if the distance between the means is $|\mu - \nu|_2 = \Omega(max(\sigma(\log n)^{\frac{1}{2} + \epsilon}, \sigma^2 |logit(w_0)|))$, then for any $c > 0$, with probability at least $1 - O(n^{-c})$, there exists a two-layer that perfectly classify the data, and obtain a cross-entropy loss given by*

$$\ell_\theta(X) = C\exp(-\frac{R}{\sqrt{2}}\|\mu - \nu\|_2(1 \pm \sqrt{c}/(\log n)^\epsilon)),$$

*where $C \in [\frac{1}{2}, 1]$ is an absolute constant and $R$ is the optimality constraint from.*

*Proof.* Consider the two-layer MLPs described in 3, for which we have $\hat{y}_i = \varphi(R(|\langle X_i, \hat{\nu}\rangle| - |\langle X_i, \hat{\mu}\rangle| - \sigma^2 \ln(\frac{w_0}{w_1})))$. We now look at the loss for a single data point $X_i$,

$$\ell_i(X, \theta) = -y_i \log(\hat{y}_i) - (1 - y_i)\log(1 - \hat{y}_i)$$
$$= \log\left(1 + \exp\left((1 - 2y_i)R(|\langle X_i, \hat{\nu}\rangle| - |\langle X_i, \hat{\mu}\rangle| - \sigma^2 \ln(\frac{w_0}{w_1}))\right)\right).$$

From Theorem 1 part 2 in Baranwal et al. (2022), we know that for $\|\mu - \nu\| = \Omega(\sigma(\log n)^{\frac{1}{2} + \epsilon})$, w.h.p we have:

$$(1 - 2y_i)R(|\langle X_i, \hat{\nu}\rangle| - |\langle X_i, \hat{\mu}\rangle|) = -R\gamma'(1 \pm o_n(1))$$

But in our case we have a bias of $\sigma^2 logit(w_0)$, thus, the loss is:

$$\ell_i(X, \theta) = log(1 + exp(-R\gamma'(1 + o_n(1)))) + (2y_i - 1)\sigma^2 logit(w_0))$$

this implies that we also need to require $\gamma = \Omega(\sigma^2|logit(w_0)|)$.

So all in all, $\gamma = \Omega\left(max\left(\sigma(\log n)^{\frac{1}{2}+\epsilon}, \sigma^2\left|logit(w_0)\right|\right)\right)$, and the loss becomes:

$$\ell_i(X, \theta) = log(1 + exp(-R\gamma'(1 + o_n(1)))) + (2y_i - 1)\sigma^2 logit(w_0)) =$$
$$\log(1 + \exp(-\Omega(1)R\gamma'(1 + o_n(1))))$$

Now, the total loss is then given by

$$\ell_\theta(X) = \frac{1}{n}\sum \ell_i(X, \theta) = \log(1 + exp(-\Omega(1)R\gamma'(1 + o_n(1)))).$$

Next, 2 implies that for $t < 0$, $\frac{e^t}{2} \le \log(1 + e^t) \le e^t$, hence, we have that there exists a constant $C \in [\frac{1}{2}, 1]$ such that

$$\ell_\theta(X) = C\exp(-\Omega(1)R\gamma'(1 + o_n(1))).$$

Note that by scaling the optimality constraint $R$, the loss can go arbitrarily close to 0.

$\square$

### D.4 PROOF OF THEOREM 5

**Lemma 14.** *Let $h(x) = |\langle x, \hat{\nu}\rangle| - |\langle x, \hat{\mu}\rangle|$ for any $x \in \mathbb{R}^d$. Consider the two-layer networks in Proposition 3 where the weight parameter of the first layer, $W^{(1)}$, is scaled by a factor of $\varepsilon = sgn(w_0 p - w_1 q)$. If a graph convolution is added to these networks in either the second or the third layer then for a sample $(A, X) \sim XOR - CSBM - I(n, d, \mu, \nu, \sigma^2, p, q)$, the output of the networks for a point $i \in [n]$ is*

$$\hat{y}_i = \varphi(g_i^{(2)}(X)) = \varphi\left(R\varepsilon\left(\frac{1}{deg(i)}\sum_{j\in[n]} a_{ij}h(X_j) - \sigma^2 log(\frac{w_0}{w_1})\right)\right).$$

*Proof.* The networks with scaled parameters are given as follows. For the two-layer network, when a graph convolution is applied at the second layer of this two-layer MLP, the output of the last layer for data $(A, X)$ is $g_i^{(2)}(X) = D^{-1}A[XW^{(1)}]_+W^{(2)}$. Then we have

$$g_i^{(2)}(X) = \frac{R\varepsilon}{deg(i)}\sum_{j\in[n]} a_{ij}\left(|\langle X_j, \hat{\nu}\rangle| - |\langle X_j, \hat{\mu}\rangle| - \sigma^2 log(\frac{w_0}{w_1})\right) =$$
$$R\varepsilon\left(\frac{1}{deg(i)}\sum_{j\in[n]} a_{ij}h(X_j) - \sigma^2 log(\frac{w_0}{w_1})\right) =$$
$$f_i^{(2)}(X) - R\varepsilon\sigma^2 log(\frac{w_0}{w_1})$$

where $f_i^{(2)}$ is defined as in Baranwal et al. (2022) as $f_i^{(2)}(X) = \frac{R\varepsilon}{deg(i)}\sum_{j\in[n]} a_{ij}h(X_j)$

$\square$

**Theorem** (Restatement of Theorem 5). *Let $(A, X) \sim XOR\text{-}CSBM\text{-}I(n, d, \mu, \nu, \sigma^2, p, q)$. If the intra-class and inter-class edge probabilities are $p, q = \Omega(\frac{\log^2 n}{n})$, the distance between the means is $||\mu - \nu||_2 = max(\Omega(\frac{\sigma \log n}{\sqrt{n(p+q)}}), \sigma^2|logit(w_0)|)$, and $sgn(w_0 p - w_1 q) = sgn(w_1 p - w_0 q)$, then*

*for any $c > 0$, with probability at least $1 - O(n^{-c})$, the networks equipped with a graph convolution in the second layer perfectly classify the data, and obtain the following loss:*

$$\ell_\theta(A, X) \leq C' \exp\left(-R\|\|\mu - \nu\|\|_2 \frac{max(|w_0 p - w_1 q|, |w_1 p - w_0 q|)}{w_0 p + w_1 q}\left(1 \pm \sqrt{\frac{c}{\log n}}\right)\right),$$

*where $C > 0$ and $C' \in [\frac{1}{2}, 1]$ are constants.*

*Proof.* Notice that by Lemma 14, we have $g_i^{(2)}(X) = f_i^{(2)}(X) + bias$. Thus, we can reuse from Baranwal et al. (2022) arguments used to characterize $f_i^{(2)}(X)$. Specifically:

1. $\frac{1}{R}f_i^{(2)}(X)$ is Lipschitz with constant $\sqrt{\frac{2}{deg(i)}} \leftrightarrow \frac{1}{R}g_i^{(2)}(X)$ is Lipschitz with constant $\sqrt{\frac{2}{deg(i)}}$.

2. Gaussian concentration -

$$P(\frac{1}{R}|f_i^{(2)}(X) - \mathbb{E}[f_i^{(L)}(X)]| > \delta \mid A) \leq 2\exp(-\frac{\delta^2 deg(i)}{4\sigma^2}) \leftrightarrow$$
$$P(\frac{1}{R}|g_i^{(2)}(X) - \mathbb{E}[g_i^{(L)}(X)]| > \delta \mid A) \leq 2\exp(-\frac{\delta^2 deg(i)}{4\sigma^2})$$

Let $\varepsilon = sgn(p - w_0(p + q)) = sgn(w_0(p + q) - p)$.

$$f_i^{(2)}(X) = \mathbb{E}(f_i^{(2)}(X)) \pm O(R\sigma\sqrt{\frac{c \log n}{n(p+q)}})$$
$$= \frac{R\varepsilon}{deg(i)} \sum_{j \in [n]} a_{ij} \mathbb{E}(h(X_j)) \pm o_n(R\sigma)$$
$$= \frac{R\varepsilon\zeta(\gamma', \sigma)}{deg(i)}(\sum_{j \in C_1} a_{ij} - \sum_{j \in C_0} a_{ij}) \pm o_n(R\sigma) \text{ (using Lemma A.4 in (Baranwal et al., 2022))}$$
.

Now let's look at $\frac{\varepsilon}{deg(i)} \cdot (\sum_{j \in C_1} a_{ij} - \sum_{j \in C_0} a_{ij})$. We know from 2 that:

$$\varepsilon \cdot (\sum_{j \in C_1} a_{ij} - \sum_{j \in C_0} a_{ij}) = \varepsilon \cdot \begin{cases} \frac{w_1 q - w_0 p}{w_0 p + w_1 q}(1 \pm o_n(1)) & if\ i \in |\mu| \\ \frac{w_1 p - w_0 q}{w_1 p + w_0 q}(1 \pm o_n(1)) & if\ i \in |\nu| \end{cases}$$

if we set $\varepsilon = sgn(w_1 p - w_0 q) = -sgn(w_1 q - w_0 p)$ (possible because of our assumption), we get:

$$sgn\left(\varepsilon \cdot (\sum_{j \in C_1} a_{ij} - \sum_{j \in C_0} a_{ij})\right) = \begin{cases} -1 & if\ i \in |\mu| \\ 1 & if\ i \in |\nu| \end{cases}$$

Thus, we have that $f_i^{(2)}(X)$ is positive when $i \in |\nu|$ and negative otherwise, as desired. And the full is expression:

$$f_i^{(2)}(X) = \begin{cases} -R\zeta(\gamma', \sigma)\frac{|w_0 p - w_1 q|}{w_1 p + w_0 q}(1 \pm o_n(1)) \pm o_n(R\sigma)\ if\ i \in |\mu| \\ R\zeta(\gamma', \sigma)\frac{|w_1 p - w_0 q|}{w_1 p + w_0 q}(1 \pm o_n(1)) \pm o_n(R\sigma)\ if\ i \in |\nu| \end{cases}$$

And subsequently

$$g_i^{(2)}(X) = f_i^{(2)}(X) - R\varepsilon\sigma^2 \log(\frac{w_0}{w_1}) =$$

$$\begin{cases} -R\zeta(\gamma', \sigma)\frac{|w_0 p - w_1 q|}{w_1 p + w_0 q}(1 \pm o_n(1)) \pm o_n(R\sigma)\ if\ i \in |\mu| \\ R\zeta(\gamma', \sigma)\frac{|w_1 p - w_0 q|}{w_1 p + w_0 q}(1 \pm o_n(1)) \pm o_n(R\sigma)\ if\ i \in |\nu| \end{cases} - R\varepsilon\sigma^2 \log(\frac{w_0}{w_1})$$

We need $\zeta(\gamma', \sigma) = \Omega(o_n(R\sigma))$ and $\zeta(\gamma', \sigma) = \Omega(\sigma^2 \log(\frac{w_0}{w_1}))$. Aside from the bias term in $g_i^{(2)}(X)$, we know that $\gamma = \Omega(\sigma\frac{\sqrt{\log n}}{\sqrt[4]{n(p+q)}})$ satisfies the first condition.

If $\sigma^2 \log(\frac{w_0}{w_1}) = o_n(1)$, then this value of $\gamma$ also satisfies the second condition.

Otherwise, note that $\zeta(\gamma', \sigma) = O(\gamma')$, thus, we need $\gamma' = \Omega(\sigma^2 \log(\frac{w_0}{w_1}))$. Denote:

$$\Gamma_0(p,q) = \frac{|w_0 p - w_1 q|}{w_1 p + w_0 q}$$
$$\Gamma_1(p,q) = \frac{|w_1 p - w_0 q|}{w_1 p + w_0 q}$$

So we have for some constant $C > 0$:

$$g_i^{(2)}(X) = (2\epsilon_i - 1)CR\zeta(\gamma', \sigma)\Gamma_{\epsilon_i}(p,q)(1 \pm o_n(1))$$

Recall that the loss for node $i$ is given by

$$\ell_\theta^{(i)}(A, X) = \log(1 + e^{(1-2\epsilon_i)g_i^{(L)}(X)}) = \log(1 + \exp(-\frac{CR\gamma^2}{\sigma}\Gamma(p,q)(1 \pm o_n(1)))).$$

Next, Fact 2 implies that for any $t < 0$, $\frac{e^t}{2} \le \log(1 + e^t) \le e^t$, hence, we have for some $C' \in [\frac{1}{2}, 1]$ that

$$\ell_\theta^{(i)}(A, X) = C' \exp(-CR\zeta(\gamma', \sigma)\Gamma_{\epsilon_i}(p,q)(1 \pm o_n(1))).$$

The total loss is given by $\frac{1}{n}\sum_{i \in [n]} \ell_\theta^{(i)}(A, X)$. Thus

$$\ell_\theta(A, X) \le$$
$$max\left(C' \exp(-CR\zeta(\gamma', \sigma)\Gamma_0(p,q)(1 \pm o_n(1))), C' \exp(-CR\zeta(\gamma', \sigma)\Gamma_1(p,q)(1 \pm o_n(1)))\right) =$$
$$C' \exp(-CR\zeta(\gamma', \sigma)(1 \pm o_n(1)) \cdot max(\Gamma_0(p,q), \Gamma_1(p,q)))$$

We can observe the loss decreases as $\gamma$ (distance between the means) increases, and increases if $\sigma^2$ (variance of the data) increases.

$\square$

# E  ADDITIONAL EXPERIMENTS AND ANALYSIS

For the Amazon-reviews data, we first fine tuned a BERT model (Devlin et al., 2019). Then extracted the last-layer embeddings, and treated these as the data in the process of training the MLP and the GNN.

We also evaluated the synthetic data model for the Walmart-Amazon dataset discussed in Section 2.

## E.1  PLOTS

Before presenting the plots, we note that it may seem as if for certain cases, the improvements crosses below 0 or above 1 which is obviously not possible. Let's explain how it may occur.
Since we deal with decimals, we get that the standard deviation is greater the the variance, when most times it's the other way around. When we look at the $mean + sd$ or at the $mean - sd$, we get weird results.
Let's look at concrete examples:

**Example 2.**

$$x = \begin{cases} 0.1 & w.p\, 0.5 \\ 0.9 & w.p\, 0.5 \end{cases}$$

*Then we have:*

$$\mathbb{E}(x) = 0.5$$
$$Var(x) = 0.4$$
$$sd(x) = \sqrt{Var(x)} \approx 0.63$$
$$\mathbb{E}(x) + sd(x) \approx 1.13$$

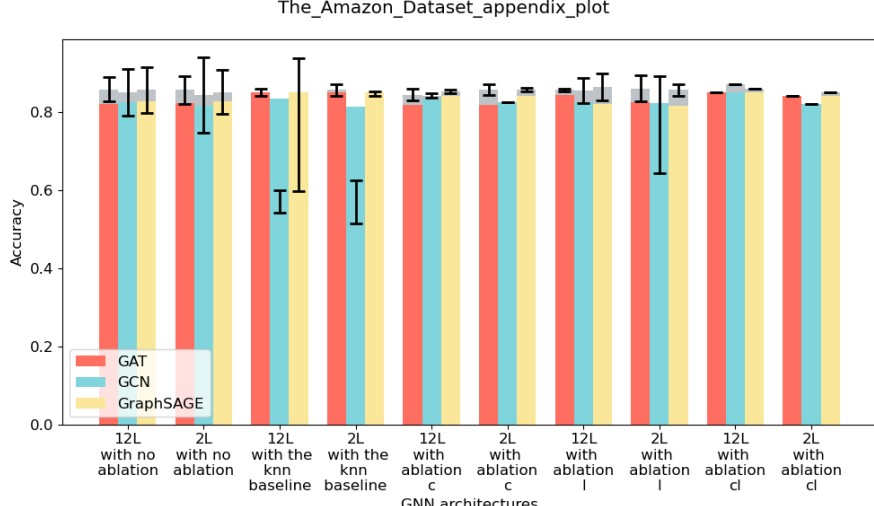

Figure 15: Improvement in the Amazon dataset, across all ablations and baselines.

**Example 3.**

$$x = \begin{cases} 0.1 & w.p\, 0.5 \\ 0.2 & w.p\, 0.5 \end{cases}$$

*Then we have:*

$$\mathbb{E}(x) = 0.15$$
$$Var(x) = 0.05$$
$$sd(x) = \sqrt{Var(x)} \approx 0.22$$
$$\mathbb{E}(x) - sd(x) \approx -0.07$$

We've already shown the plots for the Walmart-Amazon dataset in the main part, we show here the full results for these datasets in addition to the full tables of the other datasets. We present the results in Tables 16, 15, 17 and 18. We can observe the full SHIKI model consistently achieves the highest and most consistent improvement.

E.2   TABLES

We've already shown the tables for the Walmart-Amazon dataset in the main part, we show here the full results for these datasets in addition to the full tables of the other datasets. We present the results in Tables 4, 5, 6,  7, 8 and 9. In each row (representing a GNN architecture), we highlight the best edge creation method in bold based on the mean and standard deviation of the improvement. We can observe that in most cases, the SHIKI model either matches or outperforms the other leading edge creation methods.

E.3   PARAMETERS' EFFECT

Figures  19, 20, 21, 22 display the accuracy (or $f\text{-}score$) of the SHIKI model as a function of its parameters: $p$, $q$, $\tau$, and $percent$.

In most cases, increasing $q$ and $\tau$ boosts performance, whereas increasing $percent$ generally decreases it (interestingly enough aside for the $XOR\text{-}GMM$ model). Changing $p$, shows no consistent effect on improvement.

This suggests that to effectively utilize SHIKI, it is important to ensure confidence in the edges. Additionally, the parameter $q$ indicates that we don't need to rely solely on MLP predictions, and allow for prediction correction by linking nodes that appear to belong to different classes.

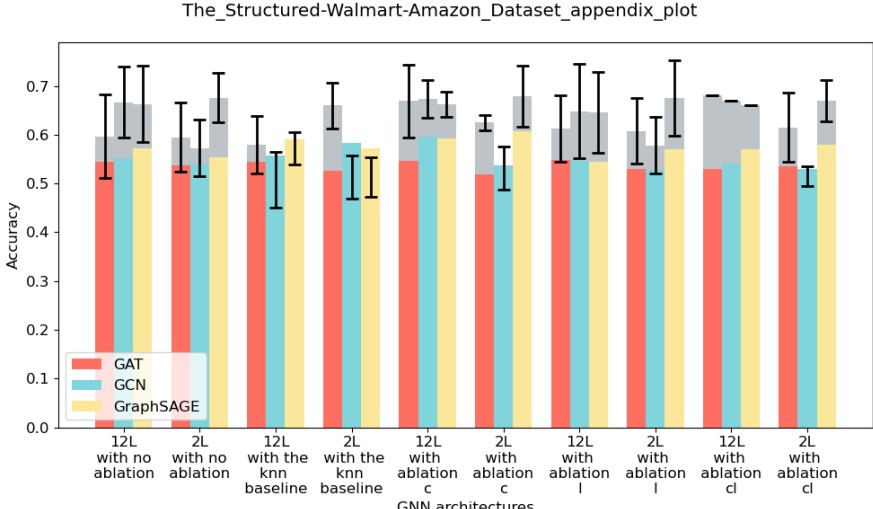

Figure 16: Improvement in the Walmart-Amazon dataset, across all ablations and baselines.

| | SHIKI | knn | No confident nodes | No labels | No confident nodes and labels |
|---|---|---|---|---|---|
| **GCN** | 1235226.0, 0.551 + **0.116 ± 0.072** | 31355.0, 0.583 + -0.07 ± 0.044 | 1677220.25, 0.597 + 0.077 ± 0.039 | 939273.0, 0.546 + 0.102 ± 0.097 | 1934846.0, 0.54 + 0.13 ± 0 |
| **GraphSAGE** | 543543.083, 0.573 + **0.09 ± 0.078** | 31355.0, 0.573 + -0.06 ± 0.04 | 1299022.0, 0.593 + 0.07 ± 0.026 | 794145.2, 0.544 + **0.102 ± 0.083** | 916166.0, 0.57 + 0.09 ± 0 |
| **GAT** | 830688.6, 0.544 + 0.053 ± 0.086 | 31355.0, 0.527 + 0.133 ± 0.047 | 2237028.75, 0.547 + **0.122 ± 0.075** | 872902.4, 0.548 + 0.064 ± 0.068 | 1973241.0, 0.53 + 0.15 ± 0 |
| **GCN2** | 1179941.125, 0.541 + **0.032 ± 0.058** | 31355.0, 0.557 + -0.05 ± 0.057 | 1813669.875, 0.537 + -0.005 ± 0.044 | 921473.4, 0.528 + **0.05 ± 0.058** | 1491614.5, 0.53 + -0.015 ± 0.021 |
| **GraphSAGE2** | 898606.35, 0.554 + **0.122 ± 0.051** | 31355.0, 0.59 + -0.018 ± 0.033 | 1479780.25, 0.608 + 0.071 ± 0.062 | 704871.5, 0.571 + 0.104 ± 0.077 | 918501.0, 0.58 + 0.09 ± 0.042 |
| **GAT2** | 1288287.275, 0.538 + 0.057 ± 0.071 | 31355.0, 0.545 + 0.035 ± 0.059 | 2711898.75, 0.519 + **0.106 ± 0.016** | 926080.4, 0.53 + 0.078 ± 0.068 | 1455389.5, 0.535 + 0.08 ± 0.071 |

Table 4: Mean improvement of our method with different strategies on multiple GNN types on the Walmart-Amazon dataset.

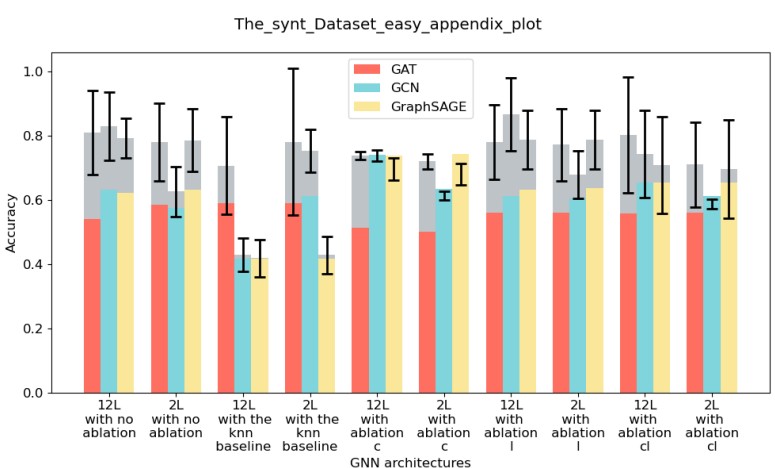

(a) Improvement in the $XOR$-$GMM$ dataset in the easy regime.

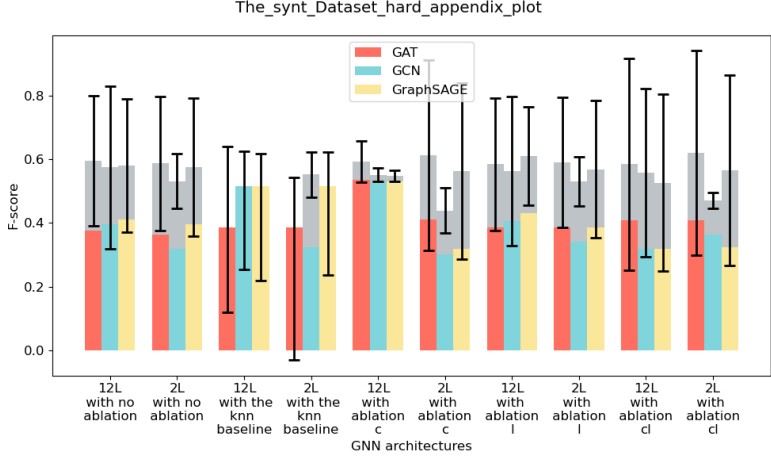

(b) Improvement in the $XOR$-$GMM$ dataset in the hard regime.

Figure 17: Improvement in the $XOR$-$GMM$ dataset, across all ablations and baselines.

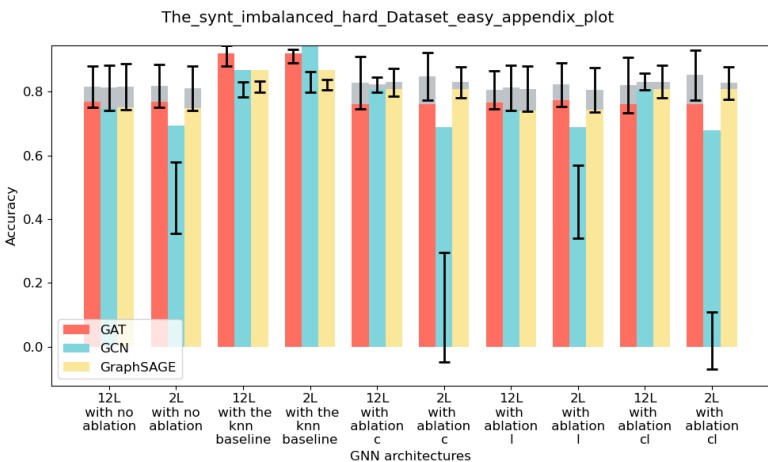

(a) Improvement in the imbalanced $XOR\text{-}GMM$ dataset in the hard regime.

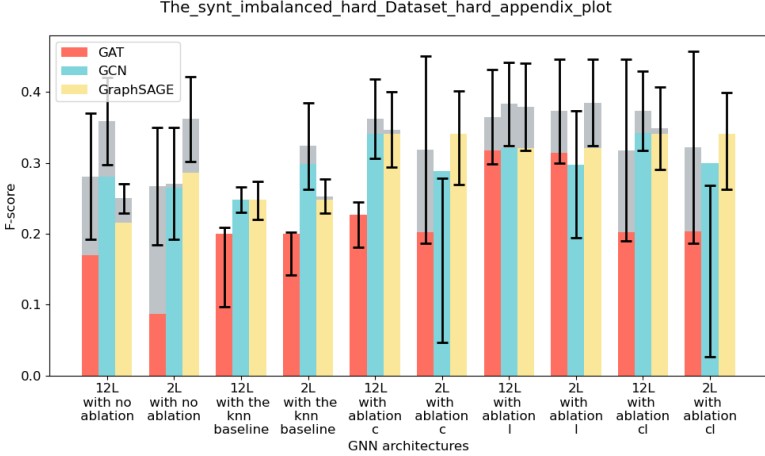

(b) Improvement in the imbalanced $XOR\text{-}GMM$ dataset in the easy regime.

Figure 18: Improvement in the imbalanced $XOR\text{-}GMM$ dataset, across all ablations and baselines.

| | SHIKI | knn | No confident nodes | No labels | No confident nodes and labels |
|---|---|---|---|---|---|
| GCN | 75876.3, 0.824 + 0.026 ± 0.059 | 12203.667, 0.833 + -0.263 ± 0.029 | 179333.75, 0.835 + 0.007 ± 0.005 | 48919.4, 0.824 + **0.03 ± 0.032** | 191680.0, 0.85 + 0.02 ± 0 |
| GraphSAGE | 72891.4, 0.828 + **0.028 ± 0.059** | 12203.667, 0.85 + -0.083 ± 0.171 | 184433.5, 0.84 + 0.012 ± 0.005 | 41641.0, 0.82 + **0.044 ± 0.034** | 191664.0, 0.85 + 0.01 ± 0 |
| GAT | 51860.55, 0.82 + **0.038 ± 0.031** | 12203.667, 0.85 + 0.0 ± 0.01 | 251228.5, 0.817 + 0.027 ± 0.015 | 38868.6, 0.844 + 0.012 ± 0.004 | 191364.0, 0.85 + 0 ± 0 |
| GCN2 | 71398.4, 0.815 + **0.028 ± 0.096** | 12203.667, 0.813 + -0.243 ± 0.055 | 288346.25, 0.825 + 0 ± 0.0 | 58066.8, 0.822 + -0.054 ± 0.124 | 95775.0, 0.82 + 0 ± 0 |
| GraphSAGE2 | 89003.05, 0.827 + **0.024 ± 0.057** | 12203.667, 0.85 + -0.003 ± 0.006 | 252282.25, 0.84 + 0.017 ± 0.005 | 67937.4, 0.816 + **0.04 ± 0.014** | 191700.0, 0.84 + 0.01 ± 0 |
| GAT2 | 65301.15, 0.822 + **0.034 ± 0.035** | 12203.667, 0.85 + 0.007 ± 0.015 | 211304.75, 0.817 + **0.04 ± 0.014** | 24583.0, 0.826 + **0.034 ± 0.033** | 191342.0, 0.84 + 0 ± 0 |

Table 5: Mean improvement of our method with different strategies on multiple GNN types on the Amazon dataset

| | SHIKI | knn | No confident nodes | No labels | No confident nodes and labels |
|---|---|---|---|---|---|
| GCN | 41384.722, 0.245 + **0.083 ± 0.062** | 9137.0, 0.248 + -0.0 ± 0.018 | 115761.094, 0.341 + 0.021 ± 0.056 | 35485.275, 0.321 + 0.062 ± 0.059 | 88497.5, 0.342 + 0.032 ± 0.056 |
| GraphSAGE | 6055.0, 0.216 + 0.034 ± 0.021 | 9137.0, 0.248 + -0.001 ± 0.027 | 140844.156, 0.341 + 0.006 ± 0.053 | 30933.75, 0.321 + **0.058 ± 0.062** | 113537.125, 0.341 + 0.008 ± 0.058 |
| GAT | 25474.308, 0.17 + **0.111 ± 0.089** | 9137.0, 0.2 + -0.047 ± 0.056 | 82940.867, 0.227 + -0.014 ± 0.032 | 29827.2, 0.317 + 0.048 ± 0.067 | 82486.25, 0.202 + **0.116 ± 0.128** |
| GCN2 | 46354.421, 0.215 + **0.026 ± 0.063** | 9137.0, 0.298 + **0.026 ± 0.061** | 180490.875, 0.288 + -0.126 ± 0.116 | 23777.675, 0.297 + -0.013 ± 0.09 | 106424.875, 0.3 + -0.153 ± 0.121 |
| GraphSAGE2 | 40277.2, 0.253 + **0.084 ± 0.062** | 9137.0, 0.248 + 0.005 ± 0.024 | 123188.406, 0.341 + -0.006 ± 0.066 | 29393.15, 0.321 + 0.064 ± 0.061 | 88346.0, 0.341 + -0.01 ± 0.068 |
| GAT2 | 5058.5, 0.087 + **0.18 ± 0.083** | 9137.0, 0.2 + -0.028 ± 0.03 | 119323.219, 0.202 + 0.117 ± 0.132 | 33376.25, 0.314 + 0.059 ± 0.073 | 71477.75, 0.203 + 0.119 ± 0.135 |

Table 6: Mean improvement of our method with different strategies on multiple GNN types on the hard imbalanced dataset.

| | SHIKI | knn | No confident nodes | No labels | No confident nodes and labels |
|---|---|---|---|---|---|
| GCN | 94389.383, 0.748 + **0.064 ± 0.071** | 9137.0, 0.869 + -0.063 ± 0.024 | 167987.271, 0.808 + 0.014 ± 0.024 | 68531.3, 0.744 + **0.068 ± 0.07** | 101141.167, 0.808 + 0.023 ± 0.026 |
| GraphSAGE | 82936.25, 0.752 + **0.063 ± 0.072** | 9137.0, 0.869 + -0.053 ± 0.017 | 145206.021, 0.808 + 0.022 ± 0.044 | 59555.767, 0.741 + **0.068 ± 0.071** | 100986.917, 0.808 + 0.023 ± 0.051 |
| GAT | 84824.742, 0.769 + 0.046 ± 0.065 | 9137.0, 0.921 + -0.008 ± 0.032 | 143709.688, 0.761 + **0.066 ± 0.082** | 59939.4, 0.766 + 0.039 ± 0.06 | 107861.833, 0.761 + 0.059 ± 0.087 |
| GCN2 | 103420.883, 0.693 + -0.226 ± 0.113 | 9137.0, 0.945 + -0.114 ± 0.033 | 193252.646, 0.688 + -0.565 ± 0.172 | 65390.317, 0.688 + -0.234 ± 0.114 | 82802.417, 0.679 + -0.66 ± 0.09 |
| GraphSAGE2 | 84008.742, 0.749 + **0.062 ± 0.07** | 9137.0, 0.869 + -0.048 ± 0.016 | 156263.542, 0.808 + 0.022 ± 0.049 | 46877.733, 0.744 + **0.062 ± 0.069** | 82050.25, 0.809 + 0.018 ± 0.051 |
| GAT2 | 95431.062, 0.769 + 0.05 ± 0.067 | 9137.0, 0.921 + -0.01 ± 0.022 | 141618.062, 0.761 + 0.087 ± 0.075 | 66775.6, 0.772 + 0.05 ± 0.069 | 102172.583, 0.761 + **0.091 ± 0.078** |

Table 7: Mean improvement of our method with different strategies on multiple GNN types on the easy imbalanced dataset.

| | SHIKI | knn | No confident nodes | No labels | No confident nodes and labels |
|---|---|---|---|---|---|
| GCN | 39181.518, 0.632 + **0.198 ± 0.107** | 9137.0, 0.417 + 0.012 ± 0.051 | 125314.515, 0.733 + 0.005 ± 0.049 | 26094.227, 0.613 + **0.254 ± 0.114** | 93707.25, 0.655 + 0.088 ± 0.136 |
| GraphSAGE | 18400.368, 0.623 + **0.169 ± 0.062** | 9137.0, 0.417 + 0.001 ± 0.057 | 121442.031, 0.741 + -0.044 ± 0.033 | 13425.333, 0.631 + 0.156 ± 0.092 | 102421.417, 0.655 + 0.053 ± 0.15 |
| GAT | 38680.627, 0.541 + **0.269 ± 0.132** | 9137.0, 0.59 + 0.117 ± 0.152 | 125610.0, 0.513 + 0.225 ± 0.013 | 25222.017, 0.559 + 0.221 ± 0.116 | 102932.583, 0.558 + 0.245 ± 0.181 |
| GCN2 | 9776.23, 0.575 + 0.051 ± 0.078 | 9137.0, 0.611 + **0.141 ± 0.067** | 205763.455, 0.628 + -0.019 ± 0.013 | 8155.967, 0.605 + 0.073 ± 0.074 | 133461.417, 0.612 + -0.025 ± 0.015 |
| GraphSAGE2 | 18267.591, 0.632 + **0.154 ± 0.097** | 9137.0, 0.417 + 0.011 ± 0.058 | 103895.444, 0.713 + -0.029 ± 0.101 | 13109.153, 0.636 + **0.151 ± 0.091** | 71875.5, 0.655 + 0.041 ± 0.154 |
| GAT2 | 35683.015, 0.583 + **0.2 ± 0.124** | 9137.0, 0.59 + **0.191 ± 0.228** | 134354.829, 0.502 + **0.22 ± 0.024** | 22888.683, 0.559 + **0.213 ± 0.113** | 88188.167, 0.559 + 0.151 ± 0.133 |

Table 8: Mean improvement of our method with different strategies on multiple GNN types on the easy synthetic dataset.

|  | SHIKI | knn | No confident nodes | No labels | No confident nodes and labels |
|---|---|---|---|---|---|
| GCN | 74693.3, 0.395 + **0.18 ± 0.256** | 9137.0, 0.515 + -0.075 ± 0.186 | 185907.0, 0.532 + 0.02 ± 0.022 | 35461.65, 0.406 + 0.158 ± 0.235 | 111521.5, 0.319 + **0.24 ± 0.264** |
| GraphSAGE | 54943.636, 0.412 + 0.168 ± 0.21 | 9137.0, 0.515 + -0.096 ± 0.199 | 170437.156, 0.532 + 0.017 ± 0.018 | 30158.96, 0.43 + 0.181 ± 0.155 | 116324.5, 0.319 + **0.208 ± 0.279** |
| GAT | 43363.5, 0.377 + 0.219 ± 0.205 | 9137.0, 0.385 + -0.006 ± 0.261 | 189625.312, 0.537 + 0.057 ± 0.065 | 28228.7, 0.386 + 0.199 ± 0.208 | 137788.0, 0.409 + 0.176 ± 0.333 |
| GCN2 | 37179.719, 0.319 + **0.213 ± 0.087** | 9137.0, 0.323 + **0.23 ± 0.071** | 143112.812, 0.302 + 0.137 ± 0.071 | 17521.375, 0.342 + 0.189 ± 0.077 | 104540.125, 0.363 + 0.109 ± 0.025 |
| GraphSAGE2 | 54268.725, 0.395 + 0.181 ± 0.218 | 9137.0, 0.516 + -0.086 ± 0.193 | 116882.531, 0.32 + **0.243 ± 0.277** | 44714.9, 0.387 + 0.182 ± 0.216 | 73274.5, 0.323 + **0.243 ± 0.299** |
| GAT2 | 43145.069, 0.363 + **0.225 ± 0.211** | 9137.0, 0.385 + -0.129 ± 0.287 | 192782.938, 0.41 + 0.204 ± 0.3 | 35336.525, 0.387 + 0.204 ± 0.204 | 142004.0, 0.409 + 0.211 ± 0.322 |

Table 9: Mean improvement of our method with different strategies on multiple GNN types on the hard synthetic dataset.

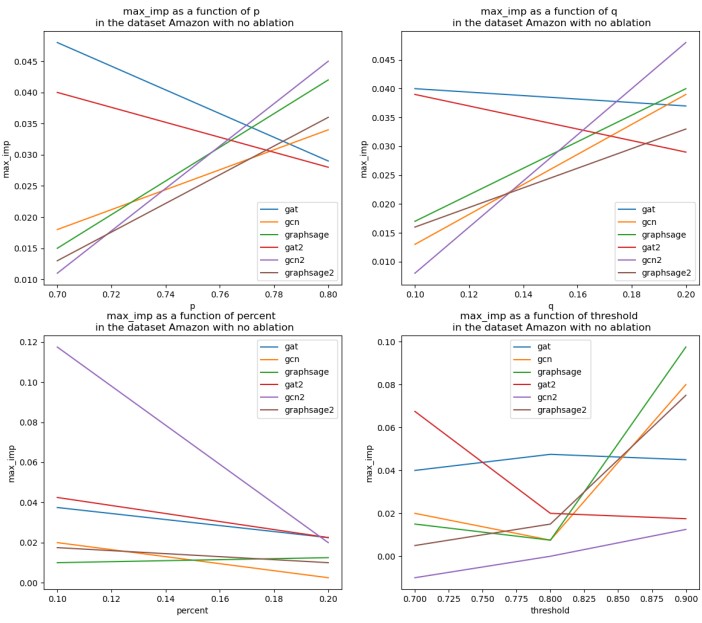

Figure 19: Parameters' effect in the SHIKI model for the Amazon dataset.

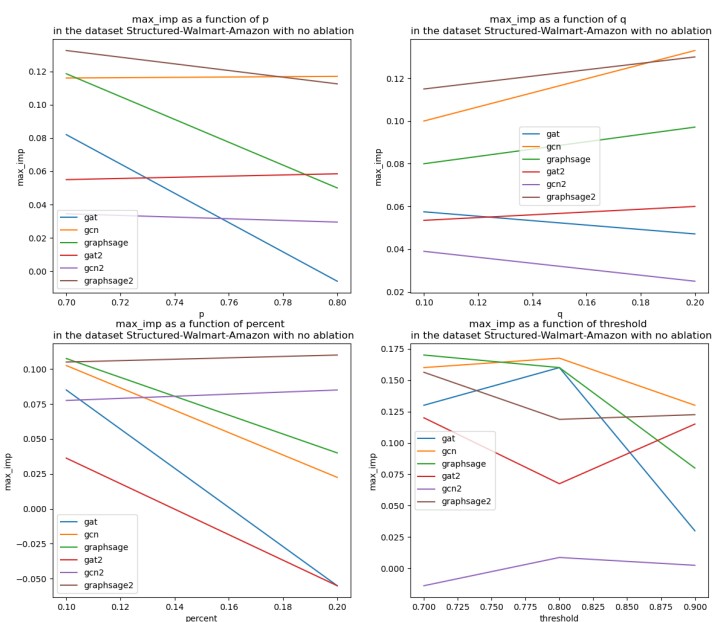

Figure 20: Parameters' effect in the SHIKI model for the Walmart-Amazon dataset.

### E.4 NUMBER OF EDGES

Here, we discuss the number of edges in each edge creation method. As we can see from the tables, the number of edges in the SHIKI model is much bigger than in the KNN baseline.

In easier datasets such as the Walmart-Amazon dataset, the number of edges in SHIKI is about 40 times larger than in KNN.

In tougher datasets, naturally, the percentage of confident nodes is smaller than in easier datasets, thus we can potentially have fewer nodes. Even in this case, the number of edges in SHIKI is about 4 times larger than in KNN.

This may indicate that in order to effectively utilize artificial edges, we need to create many edges.

### E.5 GNN ARCHITECTURES

Examining the figures, we observe SHIKI's improvement across various GNN architectures. In many cases, the differences between them are minimal, and each architecture has instances where it excels. This demonstrates that the SHIKI model performs well and consistently across all GNN architectures.

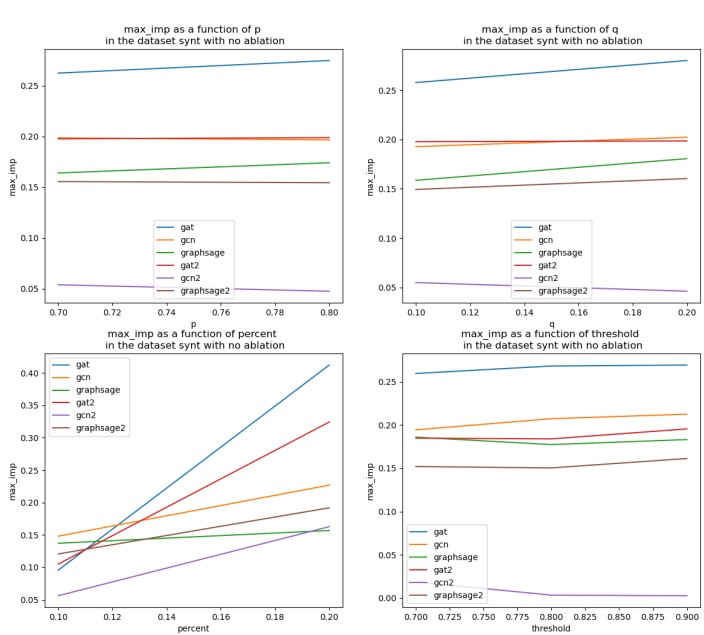

(a) Parameters' effect in the SHIKI model for the easy $XOR$-$GMM$ synthetic dataset.

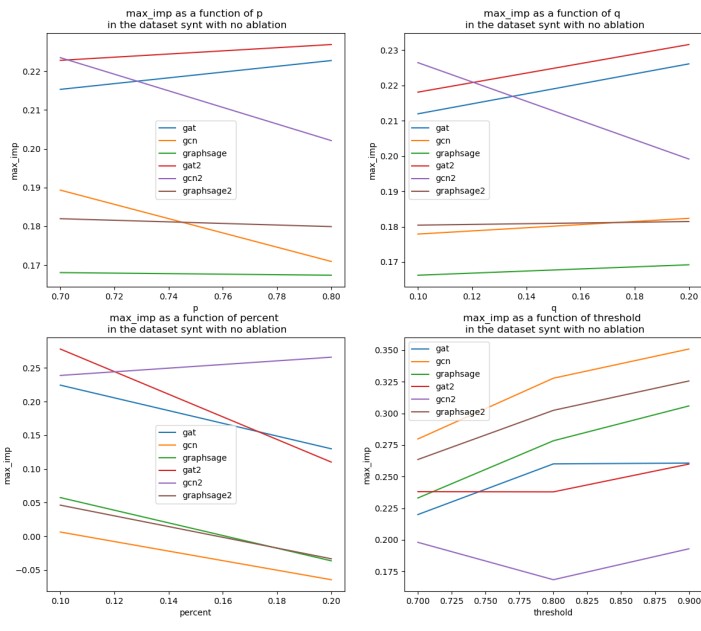

(b) Parameters' effect in the SHIKI model for the hard $XOR$-$GMM$ synthetic dataset.

Figure 21: Parameters' effect in the SHIKI model for the $XOR$-$GMM$ synthetic dataset.

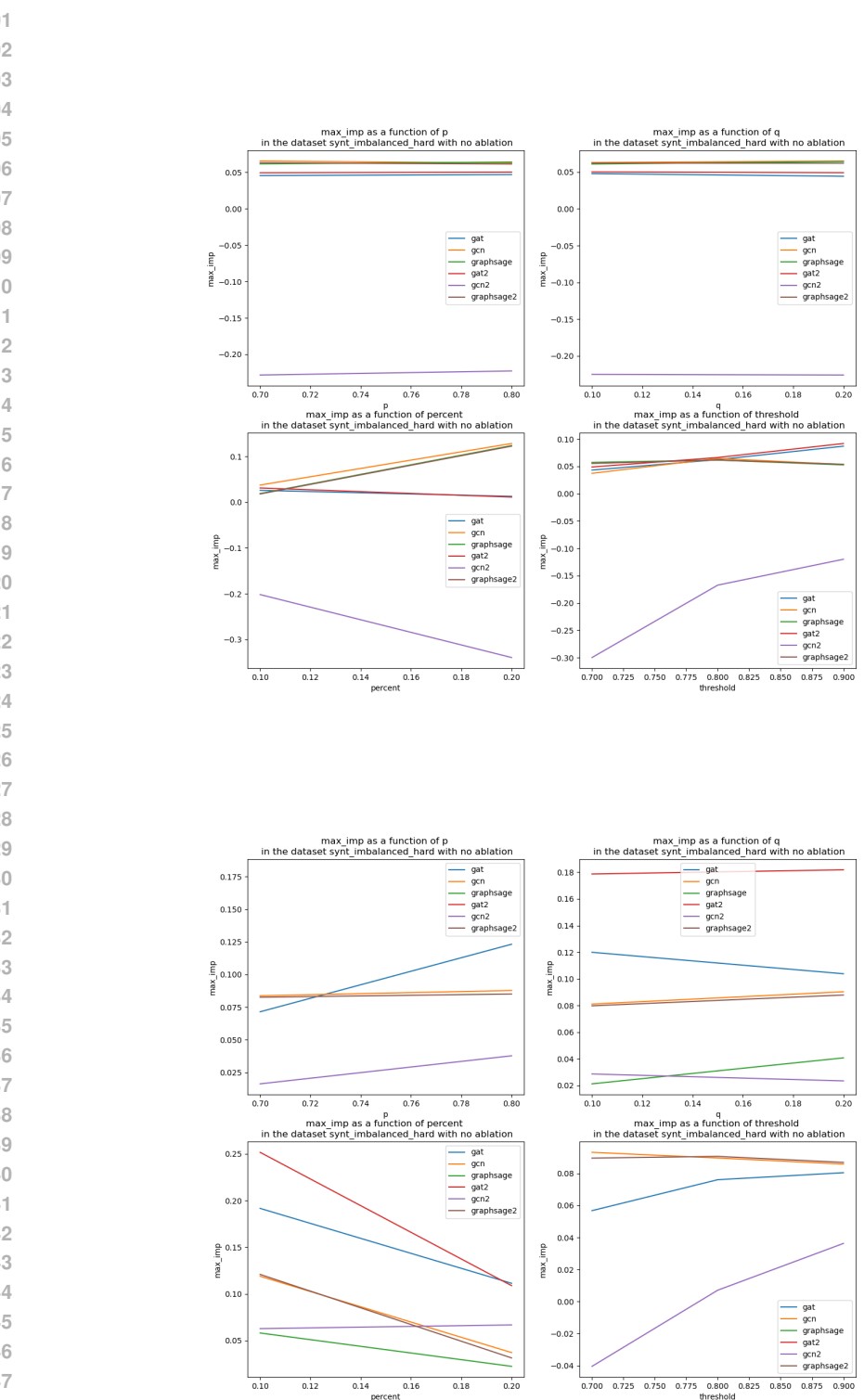

Figure 22: Parameters' effect in the SHIKI model for the synthetic imbalanced dataset.

