# OpenReview forum: "SHIKI: Self-Supervised Heuristic for Improving MLPs' Knowledge by Integrating GNNs"
_ICLR.cc/2025/Conference — ICLR 2025 Conference Withdrawn Submission_

### Official Review · Reviewer_vded · 2024-10-23

**Soundness:** 1
**Presentation:** 1
**Contribution:** 1
**Rating:** 3
**Confidence:** 4

**Summary:**

This paper explores the potential of graph convolution for addressing unstructured classification problems. By artificially constructing a graph, it can leverage the inductive bias induced by labels to aid in classification. To better address this issue, the authors propose a method for filtering using MLP confidence, achieving improved performance.

**Strengths:**

This paper studies an interesting and meaningful problem.

**Weaknesses:**

1. The authors omit many technical details, such as edge creation, and instead direct readers to references that explore different problems. The introduction is too brief and fails to motivate the problem and introduce the proposed solution effectively. The authors should revise the introduction to include a concise solution overview.
2. The authors don't motivate the use of the XOR GMM theoretical model. I speculate it's meant to represent data that is difficult to be separated linearly.  The authors then modify this model based on two specific real-world datasets, making the analysis seem tailored to these datasets and lacking generalizability. Moreover, LLM embeddings are a special feature type, and tabular data may not exhibit the same observations. The authors should revise this section to consider more general scenarios.
3.  The writing lacks rigor in its use of mathematical formulas. Chapter 3 contains numerous instances of symbol confusion. Why is the content on line 212 defined as "improvement in terms of expectation?"
4.  The graphical model in Table 1 is not a GCN.
5.   In line 256, I don't understand what is meant by "want p and q different." Shouldn't we assume the underlying generative model is fixed for a given dataset? Why is it possible to choose p and q?
6.  The theoretical analysis primarily relies on frameworks from other papers, and the proposed MLP confidence-based selection lacks theoretical support. Intuitively, this method isn't sound. Why should we trust the confidence of a weaker model? As mentioned before, this approach probably requires strong features, such as LLM embeddings.
7. The evaluation is not sound. Evaluating the method on two uncommon datasets doesn't convincingly demonstrate its effectiveness. Additionally, the presentation of tables and figures needs improvement.
8.  The related work section omits some highly relevant research. For example, [1] and [2].

[1] Errica, Federico. "On class distributions induced by nearest neighbor graphs for node classification of tabular data." Advances in Neural Information Processing Systems 36 (2024).
[2] Zeng, Hanqing, et al. "Mixture of Weak & Strong Experts on Graphs." arXiv preprint arXiv:2311.05185 (2023).

**Questions:**

1. What's the point of the visualization in Section 2, I don't see how it reflects the general case for classification
2. see weakness 3
3. see weakness 5

---

### Official Review · Reviewer_y3d9 · 2024-11-01

**Soundness:** 2
**Presentation:** 2
**Contribution:** 2
**Rating:** 5
**Confidence:** 4

**Summary:**

The paper introduces a novel self-supervised heuristic for adding edges to non-graphical data, aiming to enhance the knowledge representation of Multilayer Perceptrons (MLPs). The authors' key contribution is the development of a learning pipeline for edge creation, addressing a gap in previous works where edges were typically generated using only prior knowledge.

**Strengths:**

1. The authors' learning pipeline for edge creation is an original contribution, addressing a significant gap in previous works where edge creation relied solely on prior knowledge.
2. The problem the model attempts to solve is interesting and potentially impactful, with real-world applications in various domains where non-graphical data is prevalent.
3. The authors provide some theoretical justification to support their arguments

**Weaknesses:**

1. The paper suffers from poor writing and organizational issues, which significantly hinder its readability and comprehension. For example, heuristic instead of huristic in the section title. L266 states: "We conclude this section with some results on the improvement that can be gained by using SHIKI." This information seems misplaced and should be in the experimental section rather than the proposed pipeline section.
2. The presentation of results in Tables 2 and 3 is unclear and confusing: Multiple metrics (#edges, mean MLP accuracy, mean improvement, and improvement standard deviation) are reported in the same column, making it difficult to interpret the results effectively.
3. The paper's contribution appears incremental: The work primarily builds upon previous research, focusing only on modifying the edge creation part using a learning pipeline.
4. The proposed pipeline seems overly complicated: It requires MLPs to generate potential candidate edges and still relies on GNNs to capture structural information. Would it be possible to just use an MLP to capture the MLP knowledge? The over-complicated pipeline also limits its scalability and computational efficiency.

**Questions:**

See weakness

---

### Official Review · Reviewer_wZJL · 2024-11-02

**Soundness:** 2
**Presentation:** 1
**Contribution:** 2
**Rating:** 3
**Confidence:** 4

**Summary:**

The paper provides a new pipline for building graphs on edge-less dataset and then training GNNs on top of it.

**Strengths:**

1. The SHIKI method introduces a unique heuristic-based approach for creating edges in non-graphical data, enhancing the application of GNNs to new domains. This is particularly beneficial for tasks where inherent graph structures do not exist.

**Weaknesses:**

1. The presentation of the paper is not very concise and clear. For example, Table 2 shows a comparison experiment (which should be main results) as well as ablation studies. And Table 3 shows the same experiment but on a different dataset. Besides, the table format is not very clear for reader to capture key numbers, for example, there are too many values within one entry. It would be better if the authors could revise it by separating different experiments while merging different datasets' results.
2. Problem 2 should not be placed in the main context since it is for future work and is not discussed in this paper.
3. The experiment results are not sufficient to demonstrate the effectiveness of the proposed baseline. For example, in table 2 and table 3, a baseline results of using sole MLP should be reported. And more edge construction methods should be used for comparison, for example, [1][2].

[1] Zhao, Jianan, et al. "Self-supervised graph structure refinement for graph neural networks." Proceedings of the Sixteenth ACM International Conference on Web Search and Data Mining. 2023.
[2] https://github.com/zepengzhang/awesome-graph-structure-learning

**Questions:**

1. What are the meaning of different numbers within the same entry in Table 2 and 3?
2. What are indications of Theorem 1, since it is hard to find the correlation with the proposed pipeline. Besides, the notations used in Theorem 1 is not well introduced in the context.

---

### Official Review · Reviewer_MfzB · 2024-11-04

**Soundness:** 2
**Presentation:** 1
**Contribution:** 1
**Rating:** 3
**Confidence:** 4

**Summary:**

The paper introduces SHIKI, a self-supervised approach designed to improve Multi-Layer Perceptrons (MLPs) by integrating Graph Neural Networks (GNNs). The SHIKI framework addresses the challenge of creating meaningful edges for non-graph-structured data to enhance the learning process. This is achieved by first training an MLP to generate weak labels, identifying confident nodes, and then connecting these nodes to form a synthetic graph that supports GNN training. The method is evaluated on two real-world datasets and one synthetic dataset.

**Strengths:**

- The idea of using GNN to enhance tabular learning is a promising research direction.

**Weaknesses:**

- The paper writing is unclear. It is difficult to follow the contributions, theories, method, and experimental setup. This affects the paper’s readability and makes it challenging to understand the value of the proposed method, especially for a top conference audience.
- The method relies on MLP-generated weak labels and confidence scores to create edges, which is a simplistic approach. This edge construction technique may lead to unreliable connections, especially in datasets with noisy or complex relationships. More sophisticated methods are available for edge construction, and the approach presented here lacks innovation.
- The method is limited to two real-world datasets and one synthetic dataset. This narrow selection fails to demonstrate the generalizability and robustness of the SHIKI framework. A stronger evaluation would include multiple real-world datasets across different domains.
- More baselines for tabular data (e.g. xgboost, TabTransformer) should be compared. The paper only compares with a basic KNN baseline.

**Questions:**

Please see the weakness part.

---

### Note · Authors · 2024-11-25

I have read and agree with the venue's withdrawal policy on behalf of myself and my co-authors.